# Faster Fixed-Point Methods for Multichain MDPs

**Matthew Zurek**
Department of Computer Sciences
University of Wisconsin–Madison
matthew.zurek@wisc.edu

**Yudong Chen**
Department of Computer Sciences
University of Wisconsin–Madison
yudongchen@cs.wisc.edu

## Abstract

We study value-iteration (VI) algorithms for solving general (a.k.a. multichain) Markov decision processes (MDPs) under the average-reward criterion, a fundamental but theoretically challenging setting. Beyond the difficulties inherent to all average-reward problems posed by the lack of contractivity and non-uniqueness of solutions to the Bellman operator, in the multichain setting an optimal policy must solve the navigation subproblem of steering towards the best connected component, in addition to optimizing long-run performance within each component. We develop algorithms which better solve this navigational subproblem in order to achieve faster convergence for multichain MDPs, obtaining improved rates of convergence and sharper measures of complexity relative to prior work. Many key components of our results are of potential independent interest, including novel connections between average-reward and discounted problems, optimal fixed-point methods for discounted VI which extend to general Banach spaces, new sublinear convergence rates for the discounted value error, and refined suboptimality decompositions for multichain MDPs. Overall our results yield faster convergence rates for discounted and average-reward problems and expand the theoretical foundations of VI approaches.

## 1 Introduction

Markov decision processes (MDPs) are the canonical framework for modeling sequential decision-making problems, and sit at the core of reinforcement learning (RL), operations research, and control theory. Planning algorithms for solving MDPs therefore play a fundamental role in several fields. Among planning techniques for MDPs, value-iteration (VI) style methods, which are based upon solving the Bellman equation, are among the simplest and most fundamental, and also serve as key primitives within or templates for countless modern RL algorithms.

In this paper we study MDPs with the average-reward criterion, where the objective is to optimize long-run/steady-state performance. Despite its foundational importance for infinite-horizon problems, the average-reward setting is less well understood from a theoretical perspective due to its complexity. Compared to the discounted setting, where the Bellman operator is strongly contractive, the average-reward Bellman operator is merely nonexpansive and has non-unique solutions. In particular we focus on the unrestricted setting of general (a.k.a. multichain) MDPs, which poses particular analytical challenges, due in part to the facts that all optimal policies may induce multiple recurrent classes and so the optimal average-reward may depend on the initial state, and that multiple Bellman/optimality equations with different behaviors are needed. In more intuitive terms, one key challenge is that relative to communicating MDPs (where all states are accessible from one another via some sequence of actions), multichain MDPs may feature multiple inescapable regions of states that differ in the maximum performance achievable within said regions, and so algorithms for solving multichain MDPs must solve the "naviagtion/transient" subproblem of reaching the best possible such region, in addition to the "recurrent" subproblem of optimizing long-run performance within a region.

39th Conference on Neural Information Processing Systems (NeurIPS 2025).

Some recent work has obtained nonasymptotic convergence guarantees for VI methods in multichain average-reward MDPs, addressing some of these complexities by making connections to the discounted setting or employing Halpern iteration, a fixed-point method with nonasymptotic convergence properties even for nonexpansive operators. However, past work fails to adequately capture and adapt to the difficulty of the navigation subproblem: previous algorithms have performance bounds which degrade with excessively large complexity parameters, which are large even for easier communicating MDPs or only measure the time needed to solve the navigation problem exactly and yield vacuous guarantees before this point. Overall, algorithms from prior work fail to achieve optimal performance for general average-reward MDPs.

In this paper we address these issues, developing faster algorithms in terms of sharper measures of the complexity of multichain MDPs. To achieve this main goal, we develop many intermediate results of independent interest. These include new relationships between average-reward and discounted RL objectives, optimal algorithms for discounted value iteration (and general fixed-point problems in Banach spaces), algorithms with new sublinear $O(1/n)$ rates for reducing the discounted value error, and new suboptimality formulas for multichain MDPs which highlight the roles of both Bellman optimality equations. Beyond our algorithmic improvements, this collection of results advances the theoretical foundations of VI methods and multichain MDPs.

## 1.1  Problem setup

A Markov decision process is a tuple $(\mathcal{S}, \mathcal{A}, P, r)$ where $\mathcal{S}, \mathcal{A}$ denote the state and action spaces, respectively, which are assumed to be finite, $P : \mathcal{S} \times \mathcal{A} \to \Delta(\mathcal{S})$ is the transition kernel where $\Delta(\mathcal{S})$ denotes the probability simplex over $\mathcal{S}$, and $r \in [0,1]^{\mathcal{S} \times \mathcal{A}}$ is the reward function. A (randomized Markovian) policy is a mapping $\pi : \mathcal{S} \to \Delta(\mathcal{A})$, and we let $\Pi^{\mathrm{MR}}$ denote the set of all such policies. A deterministic policy $\pi$ has $\pi(s)$ with all probability mass on one action for all $s \in \mathcal{S}$, in which case we also treat $\pi$ as a mapping $\mathcal{S} \to \mathcal{A}$, and we denote the set of all deterministic policies by $\Pi^{\mathrm{MD}}$. For any initial state $s_0 \in \mathcal{S}$ and any policy $\pi$, we let $\mathbb{E}^{\pi}_{s_0}$ denote the expectation with respect to the induced distribution over trajectories $(s_0, A_0, S_1, A_1, \dots)$ where $A_t \sim \pi(S_t)$ and $S_{t+1} \sim P(\cdot \mid S_t, A_t)$. Let $R_t = r(S_t, A_t)$. Fixing some $\gamma \in [0, 1)$, the discounted value function $V^{\pi}_{\gamma} \in \mathbb{R}^{\mathcal{S}}$ of a policy $\pi$ is $V^{\pi}_{\gamma}(s) = \mathbb{E}^{\pi}_s[\sum_{t=0}^{\infty} \gamma^t R_t]$. Define the optimal value function $V^{\star}_{\gamma} = \sup_{\pi \in \Pi^{\mathrm{MR}}} V^{\pi}_{\gamma}$ (where the supremum is taken elementwise). The gain $\rho^{\pi} \in [0,1]^{\mathcal{S}}$ of a policy $\pi$ is $\rho^{\pi}(s) = \lim_{T \to \infty} \frac{1}{T} \mathbb{E}^{\pi}_s[\sum_{t=0}^{T-1} R_t]$. The optimal gain $\rho^{\star} \in [0,1]^{\mathcal{S}}$ is defined as $\rho^{\star} = \sup_{\pi \in \Pi^{\mathrm{MR}}} \rho^{\pi}$. The bias function $h^{\pi} \in \mathbb{R}^{\mathcal{S}}$ of a policy $\pi$ is $h^{\pi}(s) = \text{C-lim}_{T \to \infty} \mathbb{E}^{\pi}_s[\sum_{t=0}^{T-1} (R_t - \rho^{\pi}(S_t))]$, where C-lim denotes the Cesaro limit. A policy $\pi$ is Blackwell-optimal if there exists some $\overline{\gamma} < 1$ such that for all $\gamma \geq \overline{\gamma}$ we have $V^{\pi}_{\gamma} = V^{\star}_{\gamma}$; at least one such policy always exists and we denote it $\pi^{\star}$. Any policy $\pi$ induces a Markov chain over $\mathcal{S}$, and we let $P_{\pi}$ denote its transition matrix. We let $H_{P_{\pi}}$ denote the Drazin inverse of $I - P_{\pi}$. We collect some of its properties in Appendix C and also refer to Puterman [1994, Appendix A]. The suboptimality of $\pi$ is $\|\rho^{\pi} - \rho^{\star}\|_{\infty}$. An MDP is communicating if for any pair $s, s' \in \mathcal{S}$, $s'$ is accessible from $s$, meaning there exists some $\pi$ and some $k \geq 1$ such that $\mathbb{E}^{\pi}_s \mathbb{I}(S_k = s') > 0$. An MDP is weakly communicating if it consists of a set of states $\mathcal{S}_c$ such that $s'$ is accessible from $s$ for all $s, s' \in \mathcal{S}_c$, and a set of states $\mathcal{S}_t = \mathcal{S} \setminus \mathcal{S}_c$ which are transient under all policies. An MDP is general (which we use interchangeably with multichain) if there are no restrictions. We refer to [Puterman, 1994, Chapter 8.3] for more on MDP classifications. In weakly communicating MDPs $\rho^{\star}$ is equal across all states (a constant vector).

**Fixed-point methods**  For a general operator $\mathcal{L} : X \to X$ where $X$ is a Banach space with norm $\|\cdot\|$, we say that $\mathcal{L}$ is $\alpha$-Lipschitz if for all $x, x' \in X$, $\|\mathcal{L}(x) - \mathcal{L}(x')\| \leq \alpha \|x - x'\|$. $\mathcal{L}$ is ($\alpha$-)contractive if $\alpha < 1$, and $\mathcal{L}$ is nonexpansive if $\alpha = 1$. For an initial point $x_0 \in X$, Picard iteration generates the sequence $(x_t)_{t=0}^{\infty}$ by $x_{t+1} = \mathcal{L}(x_t)$, and Halpern iteration generates the sequence $x_{t+1} = (1 - \beta_{t+1})x_0 + \beta_{t+1}\mathcal{L}(x_t)$ for some sequence $(\beta_t)_{t=1}^{\infty}$ where each $\beta_t \in [0, 1]$.

**Bellman operators**  For any policy $\pi$ the policy projection matrix $M^{\pi}$ is an $\mathcal{S}$-by-$(\mathcal{S} \times \mathcal{A})$ matrix such that for any $Q \in \mathbb{R}^{\mathcal{S} \times \mathcal{A}}$ and $s \in \mathcal{S}$, $(M^{\pi}Q)(s) = \sum_{a \in \mathcal{A}} \pi(a \mid s)Q(s, a)$. The maximization operator $M : \mathbb{R}^{\mathcal{S} \times \mathcal{A}} \to \mathbb{R}^{\mathcal{S}}$ has $(MQ)(s) = \max_{a \in \mathcal{A}} Q(s, a)$. We often treat $P$ as a $(\mathcal{S} \times \mathcal{A})$-by-$\mathcal{S}$ matrix with $P_{sa,s'} = P(s' \mid s, a)$, and with this definition we have $P_{\pi} = M^{\pi}P$. For any $V \in \mathbb{R}^{\mathcal{S}}$ and any $\gamma \in [0, 1)$, the $\gamma$-discounted Bellman operator $\mathcal{T}_{\gamma} : \mathbb{R}^{\mathcal{S}} \to \mathbb{R}^{\mathcal{S}}$ is defined as $\mathcal{T}_{\gamma}(V) = M(r + \gamma PV)$. The average-reward Bellman operator is $\mathcal{T} := \mathcal{T}_1$, that is, $\mathcal{T}(V) =$

$M(r + PV)$. For a vector $V \in \mathbb{R}^{\mathcal{S}}$, the discounted and average-reward fixed-point errors (a.k.a. Bellman errors) are $\|\mathcal{T}_\gamma(V) - V\|_\infty$ and $\|\mathcal{T}(V) - V - \rho^\star\|_\infty$, respectively, and the discounted value error is $\|V - V_\gamma^\star\|_\infty$. For a policy $\pi$, the discounted and average-reward Bellman evaluation operators are $\mathcal{T}_\gamma^\pi(V) = M^\pi(r + \gamma PV)$ and $\mathcal{T}^\pi(V) = M^\pi(r + PV)$, respectively. $\mathcal{T}_\gamma, \mathcal{T}_\gamma^\pi$ are $\gamma$-contractive, and $\mathcal{T}, \mathcal{T}^\pi$ are nonexpansive, all with respect to $\|\cdot\|_\infty$. $V_\gamma^\star$ is the unique fixed-point of $\mathcal{T}_\gamma$.

**Bellman equations**  Let $\rho, h \in \mathbb{R}^{\mathcal{S}}$. The (standard/unmodified) *multichain optimality conditions* are

$$\max_{a \in \mathcal{A}} P_{sa}\rho = \rho(s) \ \forall s \in \mathcal{S}, \tag{1a}$$

$$\max_{a \in \mathcal{A}: P_{sa}\rho = \rho(s)} r(s, a) + P_{sa}h = \rho(s) + h(s) \ \forall s \in \mathcal{S}. \tag{1b}$$

The necessity for two optimality equations is unique to the multichain setting, in particular the fact that optimality requires solving both the transient/navigational subproblem, of steering towards the optimal recurrent class (expressed in equation (1a)), as well as the recurrent subproblem of attaining long-run optimality within each such class. The complications introduced by the restricted maximum of (1b) motivate the introduction of the *modified multichain optimality conditions*

$$\max_{a \in \mathcal{A}} P_{sa}\rho = \rho(s) \ \forall s \in \mathcal{S}, \tag{2a}$$

$$\max_{a \in \mathcal{A}} r(s, a) + P_{sa}h = \rho(s) + h(s) \ \forall s \in \mathcal{S}. \tag{2b}$$

These can be written in vectorized form as $M(P\rho) = \rho$ and $\mathcal{T}(h) = M(r + Ph) = \rho + h$. Solutions always exist to both equations: for any Blackwell-optimal policy $\pi^\star$, $(\rho^\star, h^{\pi^\star})$ satisfies the unmodified Bellman equations, and there exists some $M > 0$ sufficiently large so that $(\rho^\star, h^{\pi^\star} + M\rho^\star)$ satisfies both the modified and unmodified Bellman equations [Puterman, 1994, Proposition 9.1.1]. All solutions $(\rho, h)$ to the unmodified equations (1a) and (1b) have $\rho = \rho^\star$. However, a solution $(\rho, h)$ to the modified equations (2a) and (2b) do not necessarily have $\rho = \rho^\star$ (see [Bertsekas, 2018, Example 5.1.1]). A sufficient condition for $\rho = \rho^\star$ is that there exists a single policy $\pi$ satisfying both maximums simultaneously, meaning $M^\pi(P\rho) = M(P\rho)$ and $M^\pi(r + Ph) = M(r + Ph)$ [Puterman, 1994, Theorem 9.1.2ab]. Potentially of independent interest, in Lemma D.1 we show that a solution $(\rho, h)$ of the modified optimality equations admits such a simultaneously maximizing policy if and only if $(\rho, h)$ also satisfies the unmodified optimality equations.

**Complexity parameters**  For any policy $\pi \in \Pi^{\mathrm{MR}}$ we define $\mathcal{R}^\pi$ to be the set of states which are recurrent in the Markov chain $P_\pi$ and we let $\mathcal{U}^\pi = \mathcal{S} \setminus \mathcal{R}^\pi$ be the set of transient states. We define the *transient time* parameter $\mathsf{B}^\pi$ of policy $\pi$ to be the maximum (over all starting states) expected amount of time spent by $\pi$ in transient states, that is $\max_{s_0} \mathbb{E}_{s_0}^\pi \tau_{\mathcal{R}^\pi}$, where $\tau_{\mathcal{R}^\pi} = \inf\{t \geq 0 : S_t \in \mathcal{R}^\pi\}$ is the hitting time of the set $\mathcal{R}^\pi$. We define the *bounded transient time parameter* of the MDP $P$ to be $\mathsf{B} = \max_{\pi \in \Pi^{\mathrm{MD}}} \mathsf{B}^\pi$ [Zurek and Chen, 2025b]. We define the *minimum gain-optimality gap* $\Delta = \min\{\rho^\star(s) - P_{sa}\rho^\star : \rho^\star(s) - P_{sa}\rho^\star > 0, s \in \mathcal{S}, a \in \mathcal{A}\}$ [Lee and Ryu, 2024]. For $v \in \mathbb{R}^{\mathcal{S}}$ we define the span seminorm $\|v\|_{\mathrm{sp}} = \max_{s \in \mathcal{S}} v(s) - \min_{s \in \mathcal{S}} v(s)$.

## 1.2  Prior work and their limitations

In this subsection we provide a detailed description of results from recent work which are directly comparable with our main theorems. However, since the full set of our results intersects with several different areas, we provide more extensive related work in Appendix B, including references to VI analyses for multichain MDPs which give only asymptotic guarantees, and more information on Halpern iteration. Here we focus on the two most relevant prior works, which are the first to obtain nonasymptotic guarantees for general average-reward MDPs with VI approaches.

Zurek and Chen [2025b] focus on statistical complexity of average-reward RL but develop a reduction from average-reward (gain) suboptimality to discounted suboptimality, which implies the following:

**Proposition 1.1.** *Suppose that $n \geq 4$. Set $\gamma$ so that $\frac{1}{1-\gamma} = \frac{n}{2\log(n)}$. Then for the policy $\widehat{\pi}$ such that $\mathcal{T}_\gamma^{\widehat{\pi}}(V_n) = \mathcal{T}_\gamma(V_n)$ where $V_n = \mathcal{T}_\gamma^{(n)}(\mathbf{0})$, we have*

$$\|\rho^{\widehat{\pi}} - \rho^\star\|_\infty \leq 2\frac{3\mathsf{B} + 3\|h^{\pi^\star}\|_{\mathrm{sp}} + 2}{n}\log(n).$$

See Appendix E for its proof, which follows easily from Zurek and Chen [2025b]. A key shortcoming is that B is nonzero and potentially very large even in MDPs where $\rho^\star$ is constant, thus failing to adapt to situations when the navigation problem is trivial. Also the rate scales worse than $O(1/n)$.

Lee and Ryu [2024, Theorem 2], based on Halpern iteration, obtains the following guarantees:

**Proposition 1.2.** *Let $(\rho^\star, h)$ be a solution to both the modified and unmodified Bellman equations.*[1] *There exists an algorithm that, for any input $V_0 \in \mathbb{R}^\mathcal{S}$, if $n > K$ where $K = \frac{3\|r\|_\infty + 12\|V_0 - h\|_\infty + 3\|\rho^\star\|_\infty}{\Delta}$ after $n$ iterations produces output $V_n \in \mathbb{R}^\mathcal{S}$ such that*

$$\|\rho^{\widehat{\pi}} - \rho^\star\|_\infty \leq \|\mathcal{T}(V_n) - V_n - \rho^\star\|_\infty \leq \frac{8}{n+1}\|V_0 - h\|_\infty + \frac{K}{n+1}\|\rho^\star\|_\infty$$

*where $\widehat{\pi}$ satisfies $\mathcal{T}^{\widehat{\pi}}(V_n) = \mathcal{T}(V_n)$.*

Note the $\|r\|_\infty, \|\rho^\star\|_\infty$ terms are generally $\Theta(1)$ for the scaling used in this paper. The leading term involves the quantity $K$ which introduces a dependence on the potentially very large quantity $\frac{\|V_0 - h\|_\infty}{\Delta}$, which is essentially the number of iterations required to estimate $\rho^\star$ accurately enough to perfectly solve the navigation problem. Overall, (for a no-prior-knowledge initialization $V_0 = \mathbf{0}$) these two results yield incomparable rates $O((\mathsf{B} + \|h^{\pi^\star}\|_{\mathrm{sp}})\frac{\log n}{n})$ and $O(\|h\|_\infty (1 + \frac{1}{\Delta})\frac{1}{n})$, and both seem unable to adequately address the complexity of solving the navigation problem, especially when the iteration budget is not large enough to solve it perfectly. (Also as we show in Theorem D.4, $\|h\|_\infty$ may be much larger than $\|h^{\pi^\star}\|_{\mathrm{sp}}$ and introduce an additional $1/\Delta$ dependence.)

## 1.3 Our contributions

Here we summarize some of our main contributions. Lemma 2.1 and Corollary 2.2 relate policy suboptimality to both Bellman equations, and motivate the introduction of a refined complexity measure (3), whose properties we analyze. We identify new conditions that enable us to develop convergent VI methods, for both $\mathcal{T}^\pi$ (Corollary 3.2) and $\mathcal{T}$ (Algorithm 1 and Theorems 3.4 and 3.5), leading to optimal nonasymptotic rates and dependencies on the refined complexity measure. We develop new average-to-discounted reductions (Lemma 4.1). Theorem 4.2 demonstrates the optimality of Algorithm 2 for general contractive fixed-point problems in Banach spaces, leading to improvements for the discounted operator $\mathcal{T}_\gamma$. Lemma 4.3 shows that the discounted value error $\|V_\gamma^\star - V_t\|_\infty$ can be reduced at a sublinear $O(1/t)$ rate, even when $\gamma$ is close to 1, rather than $O(\gamma^t)$. These are combined to develop an algorithm based on discounted VI for multichain MDPs (Theorem 4.5) with different and improved convergence properties.

## 2 Sensitivity analysis for multichain MDPs

In order to achieve our goal of developing algorithms for multichain MDPs that have the sharpest dependence on the navigation subproblem's difficulty and give nonvacuous performance without solving it perfectly, we first analyze how the degree of error in an approximate solution to the Bellman optimality equations relates to the suboptimality of a policy constructed from this solution.

**Lemma 2.1.** *For any policy $\pi$ and any $h \in \mathbb{R}^\mathcal{S}$ we have*

$$\rho^\pi - \rho^\star = H_{P_\pi}(P_\pi \rho^\star - \rho^\star) + P_\pi^\infty(r_\pi + P_\pi h - \rho^\star - h).$$

The first and second terms on the right-hand side above are related to the first and second modified Bellman equations (2a) and (2b), respectively. If $\|P_\pi \rho^\star - \rho^\star\|_\infty = 0$, which is always the case when $\rho^\star$ is constant (such as when $P$ is weakly communicating), then the first term vanishes and we essentially recover a standard result [Puterman, 1994, Theorem 8.5.5]. However, in general this demonstrates the importance of satisfying the first Bellman equation (6). The main theorem of Lee and Ryu [2024] for general MDPs only provides a guarantee once the number of iterations is sufficiently large that the output policy $\pi$ would satisfy $\|P_\pi \rho^\star - \rho^\star\|_\infty = 0$, which incurs a dependence on the potentially large parameter $1/\Delta$ and hides the role of the first Bellman equation.

---

[1]The statement of [Lee and Ryu, 2024, Theorem 2] only requires $h$ to be a modified Bellman equation solution, but their definition requires the existence of a simultaneously argmaxing policy, which we show in Lemma D.1 is equivalent to $h$ also satisfying the unmodified equations.

Generally the vector $(P_\pi \rho^\star - \rho^\star)$ has many zero entries for reasons explained shortly, implying some entries of the matrix $H_{P_\pi}$ (the Drazin inverse of $I - P_\pi$) are irrelevant. (See Lemma F.3 for more on $H_{P_\pi}$'s entries.) This directly motivates the definition of our sharp complexity parameter $\mathsf{T}_{\mathrm{drop}}^\pi$:

$$\mathsf{T}_{\mathrm{drop}}^\pi = \max_{s \in \mathcal{S}} \mathbb{E}_s^\pi \left[ \sum_{t=0}^{\infty} \mathbb{I}\left( P_{S_t, A_t} \rho^\star < \rho^\star(S_t) \right) \right]. \tag{3}$$

This is equivalently the expected-total-reward value function of policy $\pi$ for the indicator reward function $\bar{r}(s,a) = \mathbb{I}\{P_{sa}\rho^\star < \rho^\star(s)\}$. This connection is used in an essential way for certain proofs. In words, $\mathsf{T}_{\mathrm{drop}}^\pi$ measures the maximum (over starting states) amount of time spent by $\pi$ taking "gain-dropping" actions. We also define $\mathsf{T}_{\mathrm{drop}} = \max_{\pi \in \Pi^{\mathrm{MD}}} \mathsf{T}_{\mathrm{drop}}^\pi$ (note that $\mathsf{T}_{\mathrm{drop}}$ could equivalently be defined as the optimal expected-total-reward value function for the reward $\bar{r}$), which similarly is the maximum amount of time spent taking "gain-dropping" actions by any policy, from any starting state. (By standard results [Puterman, 1994, Chapter 7], even allowing history-dependent randomized policies, the maximum is attained by some $\pi \in \Pi^{\mathrm{MD}}$.) We call $\mathsf{T}_{\mathrm{drop}}$ the *gain-dropping time*. These parameters are essentially defined in the sharpest way so that following bound holds:

**Corollary 2.2.** *For any policy $\pi$ and any vector $h \in \mathbb{R}^\mathcal{S}$,*

$$\|\rho^\pi - \rho^\star\|_\infty \leq \mathsf{T}_{\mathrm{drop}}^\pi \|P_\pi \rho^\star - \rho^\star\|_\infty + \|\mathcal{T}^\pi(h) - h - \rho^\star\|_\infty.$$

While the condition $P_{sa}\rho^\star < \rho^\star(s)$ in (3) appears very quantitative, it is actually closely related to the MDP's topological structure: such "non-gain-preserving" state-action pairs $(s,a)$ are necessarily on the "boundary" of some strongly-connected MDP component in the sense that they have nonzero probability of transitioning to some $s'$ such that there is no policy which can eventually reach $s$ from $s'$ with probability 1 (since if there were, we would have $\rho^\star(s') = \rho^\star(s)$). Since many state-action pairs cannot be on such a boundary (including all state-action pairs which are recurrent under any policy), we must have $P_{sa}\rho^\star = \rho^\star(s)$ for some state-action pairs. (Note we always have $\rho^\star(s) \geq P_{sa}\rho^\star$.)

As shown in Lemma F.1, $\mathsf{T}_{\mathrm{drop}}$ is always finite in finite MDPs. We always have $\mathsf{T}_{\mathrm{drop}} \leq \mathsf{B}$ and $\mathsf{T}_{\mathrm{drop}} \leq \frac{1}{\Delta}$, as shown in Lemmas F.5 and F.6, respectively, the latter of these facts due to an interesting Markov-inequality-like argument. $\mathsf{T}_{\mathrm{drop}} = 0$ if $\rho^\star$ is constant, whereas $\mathsf{B}$ may still be arbitrarily large in this case. We also show examples where $\mathsf{T}_{\mathrm{drop}}$ is arbitrarily smaller than $1/\Delta$ in Theorem D.4.

## 3 Value iteration for multichain average-reward MDPs

### 3.1 Coupling-based analysis

A key requirement for the convergence of Halpern iteration is that the operator must possess some fixed point. This is not the case for average-reward MDPs, where $\mathcal{T}$ instead satisfies $\mathcal{T}(h) = h + \rho^\star$ for $h$ satisfying (2b), which implies $\mathcal{T}$ has no fixed point (unless $\rho^\star = \mathbf{0}$), but the *shifted operator* $\overline{\mathcal{T}}(x) := \mathcal{T}(x) - \rho^\star$ *does* have a fixed point. The gain $\rho^\star$ is generally unknown so we cannot explicitly apply the shifted operator $\overline{\mathcal{T}}$. However, as has been observed in prior work (e.g. Bravo and Contreras [2024]), when $\rho^\star$ is a multiple of the all-one vector $\mathbf{1}$ then it commutes with $\mathcal{T}$ in the sense that $\mathcal{T}(x + \alpha \rho^\star) = \mathcal{T}(x) + \alpha \rho^\star$ for all $\alpha \in \mathbb{R}$ (since it is a standard fact that $\mathcal{T}$ commutes with $\mathbf{1}$ in this sense). This commutativity property is sufficient for obtaining the same guarantees as if the shift $\rho^\star$ were known, since then the unshifted iterates generated with $\mathcal{T}$ can be related to the hypothetical sequence of iterates which would have been generated with the appropriately shifted operator $\overline{\mathcal{T}}$.

We observe this style of coupling argument can be abstracted to general fixed-point problems involving "unshifted" operators without fixed points. In particular we can address the *policy evaluation* setting, where for some policy $\pi$ we seek $h$ such that $\mathcal{T}^\pi(h) \approx h + \rho^\pi$, where $\mathcal{T}^\pi$ is the Bellman *consistency/evaluation* operator. Suboptimal policies $\pi$ generally have non-constant gain $\rho^\pi$, even for MDPs where $\rho^\star$ is state-independent (a multiple of $\mathbf{1}$), such as weakly communicating MDPs. Thus prior arguments (such as those within Bravo and Contreras [2024]) are insufficient to analyze Halpern iteration applied to $\mathcal{T}^\pi$, but since this operator satisfies the key commutativity property $\mathcal{T}^\pi(h + \alpha \rho^\pi) = \mathcal{T}^\pi(h) + \alpha \rho^\pi$, a coupling-based analysis still works.

**Lemma 3.1.** *Fix $\rho, x_0 \in X$ and suppose $\mathcal{L} : X \to X$ satisfies $\mathcal{L}(x + \alpha \rho) = \mathcal{L}(x) + \alpha \rho$ for all $x \in X$ and $\alpha \in \mathbb{R}$. Define the shifted operator $\overline{\mathcal{L}} : X \to X$ by $\overline{\mathcal{L}}(x) = \mathcal{L}(x) - \rho$. Fix some sequence*

$(\beta_t)_{t=1}^{\infty} \in \mathbb{R}^{\mathbb{N}}$. *Then defining the sequences* $(x_t)_{t=0}^{\infty}$ *and* $(y_t)_{t=0}^{\infty}$ *by* $y_0 = x_0$ *and*

$$x_{t+1} = (1 - \beta_{t+1})x_0 + \beta_{t+1}\mathcal{L}(x_t) \qquad and \qquad y_{t+1} = (1 - \beta_{t+1})y_0 + \beta_{t+1}\overline{\mathcal{L}}(y_t),$$

*and letting* $\Lambda_t = \sum_{i=1}^{t} \prod_{j=i}^{t} \beta_j$, *we have for all* $t \geq 0$ *that*

$$x_t = y_t + \Lambda_t \rho.$$

Combining with an analysis of Halpern iteration specialized to affine operators [Contreras and Cominetti, 2022, Theorem 4.6] for an improved constant, we obtain the following result for finding fixed points of $\mathcal{T}^\pi - \rho^\pi$, the (shifted) Bellman *policy evaluation* operator.

**Corollary 3.2.** *Fix* $h_0 \in \mathbb{R}^{\mathcal{S}}$, *fix some policy* $\pi \in \Pi^{\mathrm{MR}}$, *and let* $\mathcal{T}^\pi$ *be the Bellman consistency/evaluation operator for policy* $\pi$. *Then generating the sequence* $(h_t)_{t=0}^{\infty}$ *by* $h_{t+1} = (1 - \beta_{t+1})h_0 + \beta_{t+1}\mathcal{T}^\pi(h_t)$ *where* $\beta_t = 1 - \frac{1}{t+1}$, *we have for all* $t \geq 0$ *that*

$$\|\mathcal{T}^\pi(h_t) - h_t - \rho^\pi\|_\infty \leq \frac{2}{t+1}\|h_0 - h^\pi\|_\infty.$$

This exactly matches the lower bound [Lee and Ryu, 2024, Theorem 4], meaning that this rate for the fixed-point error of $\mathcal{T}^\pi - \rho^\pi$ is unimprovable for any value-iteration-style method satisfying a certain natural subspace condition (see Lee and Ryu [2024] for details).[2]

## 3.2 Leveraging restricted commutativity

However, the commutativity property $\mathcal{L}(x + \alpha\rho) = \mathcal{L}(x) + \alpha\rho$ used in Lemma 3.1 does not generally hold for the Bellman optimality operator $\mathcal{T}$ in general MDPs. However, $\mathcal{T}$ satisfies a certain *restricted commutativity* property: for $h$ satisfying *both* the modified (2b) and unmodified (1b) Bellman equations, we have $\mathcal{T}(h + \alpha\rho^\star) = \mathcal{T}(h) + \alpha\rho^\star = h + (\alpha+1)\rho^\star$ for all $\alpha \in \mathbb{R}$. (See Lemma D.2.) Our next results show that this condition is actually sufficient to develop value-iteration-based algorithms for solving general MDPs, by using Algorithm 1.

---

**Algorithm 1** Approximately Shifted Halpern Iteration

**input:** Number of iterations $n$ for each phase, initial point $h_0$
1: Let $x_0 = h_0$                                                         ▷ gain estimation and warm-start phase
2: **for** $t = 0, \ldots, n-1$ **do**
3:      $x_{t+1} = \mathcal{T}(x_t)$
4: **end for**
5: Let $z_0 = x_n$, form gain estimate $\widehat{\rho} = \frac{1}{n}(x_n - x_0)$, form $\widehat{\mathcal{T}}$ as $\widehat{\mathcal{T}}(z) := \mathcal{T}(z) - \widehat{\rho}$
6: **for** $t = 0, \ldots, n-1$ **do**                    ▷ Halpern iteration phase; begins with $x_n$
7:      $z_{t+1} = (1 - \beta_{t+1})z_0 + \beta_{t+1}\widehat{\mathcal{T}}(z_t)$ where $\beta_t = 1 - \frac{2}{t+2}$
8: **end for**
9: **return** policy $\widehat{\pi}$ such that $\widehat{\pi}(s) = \mathrm{argmax}_{a \in \mathcal{A}} r(s,a) + P_{sa}z_n$

---

Algorithm 1 has two phases, for a total of $2n$ iterations. First, $n$ Picard steps are performed to obtain $x_n = \mathcal{T}^{(n)}(h_0)$, and this $x_n$ is used both to form a gain estimate $\widehat{\rho} = \frac{1}{n}(x_0 - h_0)$ and to initialize the next phase of the algorithm. The second phase of the algorithm is a standard Halpern iteration, but with an approximately shifted operator $\widehat{\mathcal{T}} := \mathcal{T} - \widehat{\rho}$ formed using the gain estimate. The quality of the gain estimate $\widehat{\rho}$ is ensured by the following fact:[3]

**Lemma 3.3.** *Suppose* $h$ *satisfies both the modified* (2b) *and unmodified* (1b) *Bellman equations. Then for any* $h_0 \in \mathbb{R}^{\mathcal{S}}$ *and any integer* $n \geq 0$, *we have* $\left\|\mathcal{T}^{(n)}(h_0) - n\rho^\star - h\right\|_\infty \leq \|h_0 - h\|_\infty$.

Therefore $x_n$ is close to $n\rho^\star$, and thus $\widehat{\rho}$ approaches $\rho^\star$ as $n$ increases. This fact alone is sufficient for the success of the second phase of the algorithm in finding a point $z_n$ with small fixed-point residual

---

[2]While the lower bound [Lee and Ryu, 2024, Theorem 4] is stated for the *optimality* operator $\mathcal{T} - \rho^\star$, the hard instance has $|\mathcal{A}| = 1$ and so this operator is equal to $\mathcal{T}^\pi - \rho^\pi$ for $\pi$ being the unique possible policy.

[3]While a similar statement appears as [Puterman, 1994, Theorem 9.4.1], due to some ambiguity in the $h$ for which it holds, we provide a proof and discuss the details in Appendix D.

$\|\mathcal{T}(z_n) - \rho^\star - z_n\|_\infty$. However as shown in Corollary 2.2, finding such a $z_n$, and taking the greedy policy $\widehat{\pi}$ such that $\mathcal{T}^{\widehat{\pi}}(z_n) = \mathcal{T}(z_n)$, is insufficient for achieving small suboptimality $\|\rho^{\widehat{\pi}} - \rho^\star\|_\infty$; we must also control $\|P_{\widehat{\pi}}\rho^\star - \rho^\star\|_\infty$, which is related to the navigation subproblem. This is why it is essential to initialize the second phase with $x_n$: since $x_n$ is closely aligned with $n\rho^\star$, it is possible to show that all subsequent iterates remain aligned with $n\rho^\star$, and hence $\widehat{\pi}$ will be chosen in a way which approximately maximizes $P_{\widehat{\pi}}\rho^\star$ (and the elementwise maximum of this quantity over all policies is $\rho^\star$). In summary, the first phase of the above algorithm, and in particular the initialization which is close in direction to $n\rho^\star$, is essential to how our algorithm solves the navigation subproblem. These ideas lead to the following theorem.

**Theorem 3.4.** *Let $h$ be any vector satisfying both the modified* (2b) *and unmodified* (1b) *equations (with $\rho = \rho^\star$). For all $n \geq 1$, Algorithm 1 with inputs $n, h_0$ returns $z_n$ and a policy $\widehat{\pi}$ such that*

$$\left\|\rho^\star - \rho^{\widehat{\pi}}\right\|_\infty \leq \frac{\frac{10}{3}\mathsf{T}_{\mathrm{drop}} + 13 + \frac{35}{n} + \frac{20}{n^2}}{n}\|h_0 - h\|_\infty \tag{4}$$

*and*

$$\|\mathcal{T}(z_n) - \rho^\star - z_n\|_\infty \leq \frac{13 + \frac{35}{n} + \frac{20}{n^2}}{n}\|h_0 - h\|_\infty. \tag{5}$$

The bound (5) outperforms the fixed-point error convergence rate from [Lee and Ryu, 2024, Theorem 2] by a factor of $O(1 + 1/\Delta)$, and matches (up to constants) the lower bound of $\Omega(\|h_0 - h\|_\infty/n)$ shown by [Lee and Ryu, 2024, Theorem 3] for the fixed-point residual in the weakly communicating setting. Therefore, Theorem 3.4 demonstrates a surprising finding that general MDPs are no harder than weakly communicating MDPs in terms of fixed-point error, disproving a conjecture in Lee and Ryu [2024]. This result holds for all $n \geq 1$, whereas [Lee and Ryu, 2024, Theorem 2] only holds for $n \geq C\frac{1+\|h_0-h\|_\infty}{\Delta}$ for some constant $C$. The bound (4) also obtains an improved rate of $O(\frac{\mathsf{T}_{\mathrm{drop}}+1}{n}\|h_0 - h\|_\infty)$ for the policy suboptimality, whereas [Lee and Ryu, 2024, Theorem 2] achieves a rate of $O(\frac{1+1/\Delta}{n}\|h_0 - h\|_\infty)$ (which is worse since $\mathsf{T}_{\mathrm{drop}} \leq 1/\Delta$ as shown in Lemma F.6). Furthermore, under a large-$n$ setting similar to that of [Lee and Ryu, 2024, Theorem 2], we obtain a refined guarantee for the policy suboptimality without any algorithmic changes:

**Theorem 3.5.** *Let $h$ be any vector satisfying both the modified* (2b) *and unmodified* (1b) *equations (with $\rho = \rho^\star$). For all $n \geq \frac{4\|h_0-h\|_\infty}{\Delta}$, Algorithm 1 with inputs $n, h_0$ returns a policy $\widehat{\pi}$ such that*

$$\left\|\rho^\star - \rho^{\widehat{\pi}}\right\|_\infty \leq \frac{13 + \frac{35}{n} + \frac{20}{n^2}}{n}\|h_0 - h\|_\infty.$$

Since Theorem 3.5 matches the best-known convergence rate for policy suboptimality in weakly communicating MDPs (e.g. [Lee and Ryu, 2024, Corollary 2]), this result suggests that general MDPs are also no more difficult than weakly communicating MDPs in terms of policy suboptimality, for a regime of sufficiently large $n$. We note however that to the best of our knowledge the only existing lower bound [Lee and Ryu, 2024, Theorem 3] applies specifically to fixed-point error, and showing a similar lower bound for the policy suboptimality is an interesting open question. Understanding whether there is a gap between the optimal convergence rates for policy suboptimality and for fixed-point error outside of this large-$n$ regime is another open question.

Finally we further discuss the restricted commutativity property. Lemma D.2 shows the property $\mathcal{T}(h + \alpha\rho^\star) = h + (\alpha + 1)\rho^\star$ (for all $\alpha \in \mathbb{R}$) holds if and only if $h$ solves both the modified and unmodified Bellman equations. Actually most of the proof can be performed in the abstract setting of a general nonexpansive "unshifted" operator $\mathcal{L}$ where $\mathcal{L}(x^\star + \alpha\rho) = x^\star + (\alpha + 1)\rho$ for some $\rho, x^\star \in X$ and all $\alpha \in \mathbb{R}$ (which implies $x^\star$ is a fixed point of the "shifted" operator $\mathcal{L} - \rho$). See Theorem H.5, which shows that we can find $z_n$ with small shifted fixed-point error $\|\mathcal{L}(z_n) - \rho - z_n\|$; however, there does not seem to be a generic analogue of bounding $\|P_{\widehat{\pi}}\rho^\star - \rho^\star\|_\infty$.

## 4 Discounted value iteration

### 4.1 Discounted reduction

Now we develop algorithms for solving general average-reward MDPs via a discounted VI approach. We first develop a new reduction lemma, bounding the average-reward suboptimality $\|\rho^\pi - \rho^\star\|_\infty$ in terms of the discounted fixed-point error $\|\mathcal{T}_\gamma(V) - V\|_\infty$ and our complexity parameters.

**Lemma 4.1.** *Suppose for some $V \in \mathbb{R}^{\mathcal{S}}$ that policy $\pi$ is greedy with respect to $r + \gamma PV$, that is, $\mathcal{T}_{\gamma}^{\pi}(V) = \mathcal{T}_{\gamma}(V)$, and that $\gamma \geq \frac{1}{2}$. Then, letting $h$ satisfy the modified and unmodified Bellman equations and letting $M = \min \left\{ \|h\|_{\mathrm{sp}}, \|h^{\pi^{\star}}\|_{\mathrm{sp}} + \mathsf{T}_{\mathrm{drop}} + \|h^{\pi^{\star}}\|_{\mathrm{sp}} \mathsf{T}_{\mathrm{drop}} \right\}$, we have*

$$\|\rho^{\pi} - \rho^{\star}\|_{\infty} \leq (\mathsf{T}_{\mathrm{drop}}^{\pi} + 1) \left[ (1 - \gamma)\left(4 + 7M\right) + 16\|\mathcal{T}_{\gamma}(V) - V\|_{\infty} \right].$$

See the slightly more general Lemma J.6, from which Lemma 4.1 follows immediately. Fixing some iteration budget $n$, Lemma 4.1 suggests that to get a $O(1/n)$ rate, we must set the effective horizon $1/(1 - \gamma)$ to be at least order $\Omega(n)$. Next it remains to control the discounted fixed-point error term $\|\mathcal{T}_{\gamma}(V) - V\|_{\infty}$, which is challenging with this large choice of effective horizon.

### 4.2 Faster fixed-point error convergence

As part of our first step in minimizing the discounted fixed-point error, we develop a result of independent interest: a simple algorithm which achieves an optimal fixed-point error convergence rate (up to constants) for $\gamma < 1$ contractive operators in general normed spaces. The algorithm simply deploys standard Halpern iteration for approximately $\frac{1}{1-\gamma}$ steps, then switches to Picard iteration. While the analysis is trivial, to the best of our knowledge an algorithm obtaining such guarantees was not previously known despite prior work on related problems, and we find it surprising that such a simple algorithm and analysis achieves optimality.

---

**Algorithm 2** Halpern-Then-Picard

**input:** Initial point $x_0$, contraction factor $\gamma < 1$, $\gamma$-contractive operator $\mathcal{L}$
1: Let $E = \left\lfloor \frac{1}{1-\gamma} \right\rfloor - 1$
2: **for** $t = 0, \ldots, E - 1$ **do**                    ▷ Halpern iteration for first $\approx \frac{1}{1-\gamma}$ steps
3:     $x_{t+1} = (1 - \beta_{t+1})x_0 + \beta_{t+1}\mathcal{L}(x_t)$ where $\beta_t = 1 - \frac{2}{t+2}$
4: **end for**
5: **for** $t = E, \ldots$ **do**                    ▷ switch to Picard iteration after $\approx \frac{1}{1-\gamma}$ steps
6:     $x_{t+1} = \mathcal{L}(x_t)$
7: **end for**

---

**Theorem 4.2.** *Let $\mathcal{L} : X \to X$ be a $\gamma$-contractive operator with respect to some norm $\|\cdot\|$, and let $x^{\star}$ be its fixed point. Letting $(x_t)_{t \in \mathbb{N}}$ be the sequence of iterates generated by Algorithm 2, we have*

$$\|\mathcal{L}(x_t) - x_t\| \leq \begin{cases} \frac{4}{t+1}\|x_0 - x^{\star}\| & t \leq E \\ 8(1 - \gamma)\gamma^{t-E}\|x_0 - x^{\star}\| & t > E \end{cases} \leq 8e\frac{\gamma^t}{\sum_{i=0}^{t}\gamma^i}\|x_0 - x^{\star}\|.$$

Park and Ryu [2022] show a lower bound of $(1 + \gamma)\frac{\gamma^t}{\sum_{i=0}^{t}\gamma^i}\|x_0 - x^{\star}\|_{\infty}$ for the fixed-point error (in the Hilbert space setting, meaning the lower bound may be improvable for our Banach space setting), which is matched by Algorithm 2 up to a constant of $8e/(1 + \gamma)$.

We can immediately apply Algorithm 2 with the discounted Bellman operator $\mathcal{T}_{\gamma}$ to minimize the discounted fixed-point error $\|\mathcal{T}_{\gamma}(V) - V\|_{\infty}$. With this particular operator, Lee and Ryu [2023] show a lower bound on the discounted fixed-point error of $\frac{\gamma^t}{\sum_{i=0}^{t}\gamma^i}\|V_0 - V_{\gamma}^{\star}\|_{\infty}$ (for a certain natural class of algorithms), implying the optimality of Algorithm 2 up to a factor of $8e$. Lee and Ryu [2023] also presents an algorithm for minimizing the discounted fixed-point error, and while their algorithm achieves a slightly improved constant relative to Theorem 4.2 (with $\mathcal{L} = \mathcal{T}_{\gamma}$) for certain initializations, their guarantee does not match the $\frac{\gamma^t}{\sum_{i=0}^{t}\gamma^i}\|V_0 - V_{\gamma}^{\star}\|_{\infty}$ lower bound up to constants for all initializations $V_0$ (including the initializations required in the next subsection), and hence our Algorithm 2 is the first to do so. See Appendix B.1 for more on comparing with Lee and Ryu [2023].

### 4.3 Faster value error convergence

Algorithm 2 alone (applied to $\mathcal{T}_{\gamma}$) is insufficient to obtain the desired bound on the discounted fixed-point error $\|\mathcal{T}_{\gamma}(V) - V\|_{\infty}$: the error bound depends on the initial *value error* $\|V_{\gamma}^{\star} - V_0\|_{\infty}$,

and with a generic initialization like $V_0 = \mathbf{0}$, this value error can generally be $\Omega(1/(1-\gamma)) = \Omega(n)$ with our large choice of effective horizon. Hence after $O(n)$ iterations Algorithm 2 would only lead to a bound like $\|\mathcal{T}_\gamma(V) - V\|_\infty \leq O(n/n) = O(1)$, whereas we desire this term to be $O(1/n)$.

To fix this problem, we show that by using *undiscounted* value iterations and a re-normalization step, we can reduce the value error at a sublinear $O(1/t)$ rate:

**Lemma 4.3.** *Fix $\gamma \in (0, 1)$. Let $h$ be a solution to both the modified and unmodified Bellman equations. Then for any integer $t$ such that $0 < t \leq \frac{1}{1-\gamma}$, we have*

$$\left\| \frac{1}{t(1-\gamma)} \mathcal{T}^{(t)}(\mathbf{0}) - V_\gamma^\star \right\|_\infty \leq 2 \frac{\min\left\{ \|h\|_{\mathrm{sp}}, \|h^{\pi^\star}\|_{\mathrm{sp}} + \mathsf{T}_{\mathrm{drop}} + \|h^{\pi^\star}\|_{\mathrm{sp}} \mathsf{T}_{\mathrm{drop}} \right\}}{t} \frac{1}{1-\gamma}.$$

Goyal and Grand-Clément [2023, Theorem 3] show a lower bound that $\|V_t - V^\star\|_\infty \geq \frac{\gamma^t}{1-\gamma}$ for any $V_t$ produced by an algorithm satisfying a certain iterate span condition.[4] This geometric $\gamma^t$ rate is vacuous in situations where $\gamma$ is very close to 1 (including our setting where $\gamma = 1 - 1/n$ and hence $\gamma^n \approx e^{-1} = \Omega(1)$). However, the hard instance in their lower bound depends on the iteration count $t$ and has $\|h\|_{\mathrm{sp}} = \|h^{\pi^\star}\|_{\mathrm{sp}} \geq t$ (and also $\mathsf{T}_{\mathrm{drop}} = 0$). If we instead restrict the complexity of the instance (as measured by the complexity parameters) as the iteration count $t$ is increased, their hard instance becomes inadmissible, and we find that a sublinear $O(1/t)$ rate is possible.

In particular, with our choice of effective horizon $1/(1-\gamma) = n$, we can use $t = n$ steps to produce an initialization which has value error independent of $n$ (but depends on the complexity parameters in Lemma 4.3). Generally when the iteration budget is larger than the effective horizon, we can use this warm-start procedure to reduce the value error rapidly before switching to Algorithm 2 to get an improved guarantee for solving discounted MDPs. This "warm-start" phase bears an interesting similarity to the first phase of Algorithm 1.

---

**Algorithm 3** Warm-Start Halpern-Then-Picard

**input:** Discount factor $\gamma < 1$, number of iterations $n$ such that $n \geq \lfloor \frac{1}{1-\gamma} \rfloor$
1: Let $x_0 = \mathbf{0}$, $E' = \lfloor \frac{1}{1-\gamma} \rfloor$
2: **for** $t = 0, \ldots, E' - 1$ **do**                              ▷ initialization phase
3:      $x_{t+1} = \mathcal{T}(x_t)$
4: **end for**
5: Obtain $V$ as the $(n - E')$th iterate from Algorithm 2 with inputs $x_{E'}, \gamma, \mathcal{T}_\gamma$
6: **return** $V$ and policy $\widehat{\pi}$ such that $\widehat{\pi}(s) = \operatorname{argmax}_{a \in \mathcal{A}} r(s, a) + \gamma P_{sa} V$

---

**Theorem 4.4.** *For all $n \geq E' = \lfloor \frac{1}{1-\gamma} \rfloor$, Algorithm 3 returns $V$ such that*

$$\|\mathcal{T}_\gamma(V) - V\|_\infty \leq \frac{8e}{\gamma} \frac{\gamma^{n-E}}{\sum_{i=0}^{n-E} \gamma^i} M \overset{(\star)}{\leq} \frac{8e}{\gamma} \left( 1 + \frac{e}{\gamma} \right) \frac{\gamma^n}{\sum_{i=0}^{n} \gamma^i} M$$

*where $M = \min\left\{ \|h\|_{\mathrm{sp}}, \|h^{\pi^\star}\|_{\mathrm{sp}} + \mathsf{T}_{\mathrm{drop}} + \|h^{\pi^\star}\|_{\mathrm{sp}} \mathsf{T}_{\mathrm{drop}} \right\}$, $h$ satisfies the modified and unmodified Bellman equations, and $(\star)$ holds if $n \geq 2E' - 1$.*

Algorithm 3 thus uses three phases: $E'$ steps of undiscounted Picard iteration, $E$ steps of discounted Halpern iteration, and then finally discounted Picard iteration. Combining this with our reduction result, Lemma 4.1, leads immediately to our final result on solving multichain average-reward MDPs.

**Theorem 4.5.** *Fix an integer $n \geq 2$. Set $\gamma = 1 - \frac{1}{n}$ and run Algorithm 3 with inputs $\gamma, 2n$. Let $h$ satisfy the modified and unmodified Bellman equations. Then the output policy $\widehat{\pi}$ satisfies*

$$\rho^\star - \rho^{\widehat{\pi}} \leq (\mathsf{T}_{\mathrm{drop}} + 1) \frac{71 \min\left\{ \|h\|_{\mathrm{sp}}, \|h^{\pi^\star}\|_{\mathrm{sp}} + \mathsf{T}_{\mathrm{drop}} + \|h^{\pi^\star}\|_{\mathrm{sp}} \mathsf{T}_{\mathrm{drop}} \right\} + 2}{n - 1}.$$

---

[4][Goyal and Grand-Clément, 2023, Theorem 3] actually states a weaker result, but as we show in Theorem B.2, their analysis can be strengthened to obtain the claimed lower bound.

Theorem 4.5 only makes use of the first two phases of Algorithm 3, but as we show in Theorem J.9, the constant in Theorem 4.5 could be improved by utilizing the third phase to rapidly decrease fixed-point error. One interesting advantage of Theorem 4.5 over Theorem 3.4 is that it obtains one guarantee independent of $\|h\|_{\mathrm{sp}}$, which as we discuss in Appendix D.1, can hide a dependence on $\frac{1}{\Delta}$ even when $\mathsf{T}_{\mathrm{drop}}$ and $\|h^{\pi^\star}\|_{\mathrm{sp}}$, the span of a Blackwell optimal policy, are small.

## 5 Conclusion and limitations

In this paper we designed faster VI algorithms for solving multichain average-reward MDPs, and along the way developed a collection of results advancing the theoretical foundations of VI methods and multichain MDPs. One limitation of our results is that the Bellman operators may not be exactly computable, and even if they are, it may be preferable to leverage stochastic evaluations to speed up computations; we believe these are promising directions for future work.

## Acknowledgments and Disclosure of Funding

Y. Chen and M. Zurek acknowledge support by National Science Foundation grants CCF-2233152 and DMS-2023239.

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

# A  Guide to appendices

In Appendix B we discuss related work in more detail. In Appendix C we introduce more notation used within the proofs. In Appendix D we discuss topics related to the solutions of the modified and unmodified Bellman equations. Appendix E contains a proof of Proposition 1.1 discussed in the introduction. The remaining appendices are mainly devoted to proofs of the results presented in the paper, and largely proceed in the same order. Appendix F contains proofs of the results from Section 2. Appendices G and H prove the results from Subsections 3.1 and 3.2. Theorem 4.2 is proven in Appendix I. Other results related to discounted VI are proven in Appendix J.

# B  Related work

Here we discuss more related work.

**Analysis of value iteration in average-reward MDPs**  Value iteration methods have been extensively studied for average-reward MDPs. We refer to Schweitzer and Federgruen [1977, 1979] and the references within, which develop asymptotic convergence guarantees for value iteration in multichain MDPs.

**Perturbation analysis for Markov chains**  Results of a similar nature to our Theorem 2.1 and Corollary 2.2 have a long history and are referred to as perturbation theory or sensitivity analysis for Markov chains. Some references include Schweitzer [1968], Meyer [1980, 1994], Ipsen and Meyer [1994], Cho and Meyer [2000]. Cho and Meyer [2001] provides a comprehensive comparison of such bounds, all focusing on the case of ergodic Markov chains. Such bounds commonly involve parameters related to the deviation matrix $H_P$ (the Drazin inverse of $I - P$ for a Markov chain with transition matrix $P$). Also see Cao [1999, 2000], Cao and Guo [2004] for similar results focused on MDPs.

**Halpern iteration**  While Halpern iteration has a long history [Halpern, 1967], its nonasymptotic convergence properties have been the subject of intense recent study [Sabach and Shtern, 2017]. Lieder [2021] and Park and Ryu [2022] obtain exactly optimal convergence rates in Hilbert spaces, for nonexpansive and contractive operators. Bravo et al. [2022], Contreras and Cominetti [2022] consider the general normed space setting, of greater relevance to MDPs. In the context of Markov decision processes and RL, Bravo and Contreras [2024] develop a Q-learning algorithm based on Halpern iteration, and Lee and Ryu [2023] and Lee and Ryu [2024] utilize Halpern iteration for solving discounted and average-reward MDPs, respectively.

**Complexity of discounted value iteration**  Many different approaches have been taken in the literature to try to accelerate convergence of value iteration methods, especially when the discount factor is close to 1. Goyal and Grand-Clément [2023] develop faster VI algorithms for solving discounted MDPs using momentum-like techniques, obtaining faster convergence in terms of the value error for reversible MDPs and also proving lower bounds. Lee and Ryu [2023] develop an algorithm for solving discounted MDPs with improved convergence properties for the fixed-point error, using a variant of Halpern iteration, and also develop lower bounds. There exist many alternative approaches, less closely related to the present work, based on changing the operator from the Bellman operator. See [Puterman, 1994, Chapter 6] or [Bertsekas, 2018, Chapter 2] for some of the variants of value iteration.

**Average-reward-to-discounted reductions**  Solving average-reward MDPs by approximating them via discounted problems is a very common approach in the RL literature, especially among works studying statistical (sample) complexity. Jin and Sidford [2021], Wang et al. [2022], Zurek and Chen [2025a] develop such reductions applicable to weakly communicating MDPs or even more restrictive classes of MDPs. Fruit et al. [2018], Zurek and Chen [2024a] consider the related approach of perturbing the transition matrix to ensure contractive properties. Zurek and Chen [2025b] develop the first such reduction for general MDPs, involving the parameter B. Zurek and Chen [2025b] also provides lower bounds on the sample complexity of solving general average-reward MDPs in terms of B. An interesting question is whether it could be replaced by the sharper $T_{\text{drop}}$ within their results. Halpern iteration can be understood as regularizing a nonexpansive operator to make it strongly

contractive, and decaying the regularization strength. From this perspective, there are conceptual similarities to discounted RL, particularly for problems where $\gamma$ is not intrinsic but rather functions as a tuning parameter which trades off long-term performance for computational tractability.

**Other related work**  Schweitzer and Federgruen [1978] study the solutions of the unmodified Bellman equations.

## B.1   Comparison of Algorithm 2 to prior work

Here we give more detail on the comparison of Algorithm 2, when applied to the discounted Bellman operator $\mathcal{T}_\gamma$, with Lee and Ryu [2023], who also consider minimizing the discounted fixed-point error and obtain similar guarantees. The algorithm of Lee and Ryu [2023] can also be seen as interpolating between Halpern and Picard iterations, but in a more continuous manner (by continuously adjusting stepsizes) rather than the discrete transition employed in Algorithm 2. For certain specialized initializations, [Lee and Ryu, 2023, Theorem 2] demonstrates an improved constant relative to that achieved by our Algorithm 2, but the key limitation of [Lee and Ryu, 2023, Theorem 2] is that it has order-wise worse performance for some general initializations. Specifically, ignoring constant factors, their result for general initializations can be written as

$$\|\mathcal{T}_\gamma(V_n) - V_n\|_\infty \leq \frac{\gamma^n}{\sum_{i=0}^n \gamma^i} \max\left\{\|V_0 - V_\gamma^\star\|_\infty, \left\|V_0 - \widehat{V}_\gamma^\star\right\|_\infty\right\}$$

where they define $\widehat{V}_\gamma^\star$ to be the fixed-point of a Bellman anti-optimality operator $\mathcal{T}_{\gamma,\text{anti}}$ defined as

$$\mathcal{T}_{\gamma,\text{anti}}(Q)(s,a) := r(s,a) + \min_{a \in \mathcal{A}} \gamma \sum_{s' \in \mathcal{S}} P(s' \mid s, a) Q(s', a)$$

(see [Lee and Ryu, 2023, Section 4] for how their [Lee and Ryu, 2023, Theorem 2] can be written in terms of $\frac{\gamma^n}{\sum_{i=0}^n \gamma^i}$ up to constant factors). $\mathcal{T}_{\gamma,\text{anti}}$ replaces the $\max$ in the usual discounted operator with a $\min$, and as we now show, this can lead its fixed point $\widehat{V}_\gamma^\star$ being very far from $V_\gamma^\star$, the fixed point of $\mathcal{T}_\gamma$. Consider the following one state, two action MDP:

Figure 1: A one-state, two-action MDP where $\widehat{V}_\gamma^\star$ and $V_\gamma^\star$ are far apart. The two actions are denoted by straight and dashed lines respectively, and are annotated with their associated reward.

It is trivial to check that $V_\gamma^\star = \frac{1}{1-\gamma}$ and $\widehat{V}_\gamma^\star = 0$, so for any $V_0 \in \mathbb{R}$ we have $\max\left\{\|V_0 - V_\gamma^\star\|_\infty, \left\|V_0 - \widehat{V}_\gamma^\star\right\|_\infty\right\} \geq \frac{1}{2}\frac{1}{1-\gamma}$. This issue is prohibitive to good performance in the setting considered in our Section 4, where we are able to construct a $V_0$ with $\|V_0 - V_\gamma^\star\|_\infty$ bounded independently of $\frac{1}{1-\gamma}$.

## B.2   Lower bounds for discounted value error

[Goyal and Grand-Clément, 2023, Theorem 3] shows the following:

**Proposition B.1.** *Fix $\gamma \in (0,1)$ and an integer $n \geq 1$. Then there exists a discounted MDP $(\mathcal{S}, \mathcal{A}, P, r, \gamma)$ such that for any sequence of iterates $(V_t)_{t=0}^\infty$ satisfying $V_0 = \mathbf{0}$ and*

$$V_{s+1} \in \text{span}\{V_0, \ldots, V_s, \mathcal{T}_\gamma(V_0), \ldots, \mathcal{T}_\gamma(V_s)\} \ \ \forall s \geq 0,$$

*we have for any $s \in \{1, \ldots, n-1\}$ that*

$$\left\|V_s - V_\gamma^\star\right\|_\infty \geq \frac{\gamma^s}{1+\gamma}.$$

In fact, by reusing the work done within their proof, it is possible to prove the following stronger result:

**Theorem B.2.** *Fix $\gamma \in (0,1)$ and an integer $n \geq 1$. Then there exists a discounted MDP $(\mathcal{S}, \mathcal{A}, P, r, \gamma)$ such that for any sequence of iterates $(V_t)_{t=0}^\infty$ satisfying $V_0 = \mathbf{0}$ and*

$$V_{s+1} \in span\{V_0, \ldots, V_s, \mathcal{T}_\gamma(V_0), \ldots, \mathcal{T}_\gamma(V_s)\} \quad \forall s \geq 0,$$

*we have for any $s \in \{1, \ldots, n-1\}$ that*

$$\left\|V_s - V_\gamma^\star\right\|_\infty \geq \frac{\gamma^s}{1-\gamma}.$$

*Proof.* We use the same instance as the one constructed in the proof of [Goyal and Grand-Clément, 2023, Theorem 3], which has $\mathcal{S} = \{1, \ldots, n\}$. As argued in this proof, the optimal value function satisfies $V_\gamma^\star(i) = \frac{\gamma^{i-1}}{1-\gamma}$ for all $i = 1, \ldots, n$. They also show that for any $s \in \{0, \ldots, n-1\}$, assuming $V_s$ satisfies the condition in the theorem statement, we have that $V_s(i) = 0$ for any $i \geq s+1$. Therefore for any $s \in \{0, \ldots, n-1\}$,

$$\left\|V_s - V_\gamma^\star\right\|_\infty \geq \left|V_s(s+1) - V_\gamma^\star(s+1)\right| = \left|0 - \frac{\gamma^{s+1-1}}{1-\gamma}\right| = \frac{\gamma^s}{1-\gamma}.$$

$\square$

## C    More notation

$\|W\|_{\infty \to \infty}$ denotes the $\|\cdot\|_\infty \to \|\cdot\|_\infty$ operator norm of a matrix $W$. We note that this is also equal to $\max_i \left\|e_i^\top W\right\|_1$, that is, the maximum $\ell_1$ norm of all rows of $W$.

For any policy $\pi$ we let $r_\pi = M^\pi r_\pi$, or equivalently $r_\pi(s) = \sum_{a \in \mathcal{A}} \pi(a \mid s) r(s,a)$ (for any $s \in \mathcal{S}$). We have $V_\gamma^\pi = (I - \gamma P_\pi)^{-1} r_\pi = \sum_{t=0}^\infty \gamma^t P_\pi^t r_\pi$. We let $\pi_\gamma^\star$ denote a $\gamma$-discounted optimal policy (such that $V_\gamma^{\pi_\gamma^\star} = V_\gamma^\star$). There always exists such a discounted optimal policy which is deterministic.

Fixing a policy $\pi$, the limiting matrix $P_\pi^\infty$ is

$$P_\pi^\infty = \underset{T \to \infty}{\text{C-lim}} \, P_\pi^T = \lim_{T \to \infty} \frac{1}{T} \sum_{k=0}^{T-1} P_\pi^k$$

where C-lim is the Cesaro limit. We have $P_\pi^\infty P_\pi = P_\pi P_\pi^\infty = P_\pi^\infty$. We denote the Drazin inverse of $I - P_\pi$ as $H_{P_\pi}$. It is sometimes referred to as the deviation matrix. We have

$$H_{P_\pi} = \underset{T \to \infty}{\text{C-lim}} \sum_{k=0}^{T-1} (P_\pi^k - P_\pi^\infty).$$

Additionally $H_{P_\pi}$ satisfies

$$(I - P_\pi) H_{P_\pi} = H_{P_\pi}(I - P_\pi) = I - P_\pi^\infty$$
$$H_{P_\pi} P_\pi^\infty = P_\pi^\infty H_{P_\pi} = 0$$

We also have that $\rho^\pi = P_\pi^\infty r_\pi$ and $h^\pi = H_{P_\pi} r_\pi$. We refer to [Puterman, 1994, Appendix A] for more properties of $H_{P_\pi}$

The modified Bellman equations can be written in a vectorized form as

$$M(P\rho) = \rho, \tag{6}$$
$$\mathcal{T}(h) = M(r + Ph) = \rho + h. \tag{7}$$

# D Properties of solutions to Bellman equations

First we show that for some $(\rho, h)$ which satisfy the modified Bellman equations, there exists a simultaneously argmaxing policy if and only if they also satisfy the unmodified Bellman equations.

**Lemma D.1.** *Suppose $(\rho, h)$ satisfies the modified Bellman equations* (6) *and* (7)*. Then there exists a policy $\pi$ attaining the maximums in both equations simultaneously, that is there exists $\pi \in \Pi^{\mathrm{MD}}$ such that*

$$M^\pi(P\rho) = \rho \tag{8}$$
$$M^\pi(r + Ph) = \rho + h. \tag{9}$$

*if and only if $(\rho, h)$ also satisfies the unmodified Bellman equations* (1a) *and* (1b)*.*

*Proof.* For the entire proof we fix $(\rho, h)$ satisfying the modified Bellman equations (6) and (7). First we suppose that there exists some $\pi \in \Pi^{\mathrm{MD}}$ attaining the maximums in both equations simultaneously (satisfying (8) and (9)) and try to show that $(\rho, h)$ satisfy the unmodified Bellman equations (1a) and (1b). It is immediate that (1a) is satisfied since it is the same as equation (2a). To check (1b), note that by (8) we have that $P_{s\pi(s)}\rho = \rho$, and by (9) and (2b) (for the first and second equalities, respectively) we have

$$r(s, \pi(s)) + P_{s\pi(s)}h = \rho(s) + h(s) = \max_{a \in \mathcal{A}} r(s, a) + P_{sa}h.$$

Therefore we have that

$$\max_{a \in \mathcal{A}: P_{sa}\rho = \rho(s)} r(s, a) + P_{sa}h \geq r(s, \pi(s)) + P_{s\pi(s)}h = \rho(s) + h(s)$$

(since $P_{s\pi(s)}\rho = \rho$) and also trivially

$$\max_{a \in \mathcal{A}: P_{sa}\rho = \rho} r(s, a) + P_{sa}h \leq \max_{a \in \mathcal{A}} r(s, a) + P_{sa}h = \rho(s) + h(s),$$

so we must have $\max_{a \in \mathcal{A}: P_{sa}\rho = \rho(s)} r(s, a) + P_{sa}h = \rho(s) + h(s)$ (that is, the second unmodified Bellman equation (1b) is satisfied).

Now we assume that $(\rho, h)$ satisfy the unmodified Bellman equations (1a) and (1b) and try to show that there exists a policy $\pi \in \Pi^{\mathrm{MD}}$ satisfying (8) and (9). We define (for each $s \in \mathcal{S}$) $\pi(s) \in \mathcal{A}$ to be an action attaining the maximum in (1b), that is $P_{s\pi(s)}\rho = \rho(s)$ and

$$r(s, \pi(s)) + P_{s\pi(s)}h = \rho(s) + h(s).$$

But then since $\rho(s) + h(s) = \max_{a \in \mathcal{A}} r(s, a) + P_{sa}h$ by the modified Bellman equation (2b), we clearly have that $\pi$ satisfies (8) and (9). $\qquad\square$

Next we show another equivalent property to $(\rho, h)$ satisfying both the unmodified and modified Bellman equations, the restricted commutativity property, which is essential for Algorithm 1.

**Lemma D.2.** $(\rho^\star, h)$ *satisfies both the modified and unmodified Bellman equations if and only if for any $\alpha \in \mathbb{R}$ we have*

$$\mathcal{T}(h + \alpha\rho^\star) = h + (\alpha + 1)\rho^\star. \tag{10}$$

*Proof.* First suppose that $(\rho^\star, h)$ satisfies both the modified and unmodified Bellman equations. By Lemma D.1, there exists a simultaneously argmaxing policy $\pi$ satisfying (8) and (9). Now letting $\alpha \in \mathbb{R}$ be arbitrary, we have that

$$\mathcal{T}(h + \alpha\rho^\star) \geq \mathcal{T}^\pi(h + \alpha\rho^\star) = r_\pi + P_\pi h + P_\pi \alpha\rho^\star = r_\pi + P_\pi h + \alpha\rho^\star = \mathcal{T}(h) + \alpha\rho^\star$$

using the properties of $\pi$. However we can also bound

$$\mathcal{T}(h + \alpha\rho^\star) = M(r + Ph + P\alpha\rho^\star) \leq M(r + Ph) + M(P\alpha\rho^\star) = \mathcal{T}(h) + \alpha\rho^\star.$$

Therefore we must have that $\mathcal{T}(h + \alpha\rho^\star) = \mathcal{T}(h) + \alpha\rho^\star = h + (\alpha + 1)\rho^\star$ as desired (using that $h$ satisfies the modified Bellman equation in this final equality).

Now we assume (10) holds for all $\alpha \in \mathbb{R}$. Note that we immediately have that $(\rho^\star, h)$ satisfies the modified Bellman equation by taking $\alpha = 0$ (and we always have that $MP\rho^\star = \rho^\star$). Now we choose $\overline{\alpha}$ sufficiently large so that the corresponding deterministic argmaxing policy $\pi$ such that

$$\mathcal{T}(h + \overline{\alpha}\rho^\star) = \mathcal{T}^\pi(h + \overline{\alpha}\rho^\star)$$

must satisfy $P_\pi \rho^\star = \rho^\star$. Specifically $\overline{\alpha} > \frac{1 + \|h\|_{\mathrm{sp}}}{\Delta}$ suffices, because then if $\pi$ did not satisfy $P_\pi \rho^\star = \rho^\star$ (but is deterministic) then we would have $P_\pi \rho^\star \le \rho^\star - \Delta = P_{\pi^\star}\rho^\star - \Delta$ by the definition of $\Delta$, and thus (fixing some arbitrary $s \in \mathcal{S}$) we would have

$$\overline{\alpha} > \frac{1 + \|h\|_{\mathrm{sp}}}{\Delta} \ge \frac{r_\pi(s) - r_{\pi^\star}(s) + e_s^\top(P_\pi - P_{\pi^\star})h}{\Delta}$$

$$\implies \overline{\alpha}e_s^\top(P_{\pi^\star}\rho^\star - P_\pi\rho^\star) = \overline{\alpha}e_s^\top(\rho^\star - P_\pi\rho^\star) \ge \overline{\alpha}\Delta > r_\pi(s) - r_{\pi^\star}(s) + e_s^\top(P_\pi - P_{\pi^\star})h$$

$$\implies \mathcal{T}^\pi(h + \overline{\alpha}\rho^\star)(s) = r_\pi(s) + e_s^\top P_\pi(h + \overline{\alpha}\rho^\star) < r_{\pi^\star}(s) + e_s^\top P_{\pi^\star}(h + \overline{\alpha}\rho^\star) = \mathcal{T}^{\pi^\star}(h + \overline{\alpha}\rho^\star)(s)$$

and the final inequality contradicts the fact that $\mathcal{T}(h + \overline{\alpha}\rho^\star) = \mathcal{T}^\pi(h + \overline{\alpha}\rho^\star)$. Now using (10) and then this fact that $P_\pi\rho^\star = \rho^\star$, and then canceling terms, we have

$$h + (\overline{\alpha} + 1)\rho^\star = \mathcal{T}(h + \overline{\alpha}\rho^\star) = \mathcal{T}^\pi(h + \overline{\alpha}\rho^\star) = r_\pi + P_\pi h + P_\pi\overline{\alpha}\rho^\star = r_\pi + P_\pi h + \overline{\alpha}\rho^\star$$

$$\implies h + \rho^\star = r_\pi + P_\pi h = \mathcal{T}^\pi(h).$$

Using $\alpha = 0$ in (10) we have that $\mathcal{T}(h) = \rho^\star + h$, so $\mathcal{T}^\pi(h) = \mathcal{T}(h)$. Thus we have shown that $\pi$ is a simultaneously argmaxing policy for the modified Bellman equation solution $(\rho^\star, h)$ in the sense of (8) and (9), so by Lemma D.1 $h$ satisfies both the modified and unmodified Bellman equations. $\square$

Next we prove Lemma 3.3, and discuss the ambiguity within [Puterman, 1994, Theorem 9.4.1]. To the best of our understanding, the proof of [Puterman, 1994, Theorem 9.4.1] makes use of a particular solution $h^\star$ to the modified Bellman equations which is constructed in [Puterman, 1994, Proposition 9.1.1] by adding a large multiple of $\rho^\star$ to some $h'$ which satisfies the unmodified Bellman equations. This ensures the existence of a simultaneously argmaxing policy (in the sense of equations (8) and (9)), and such a policy is used in an essential way for the proof. As noted in Subsection 1.1, not all solutions $(\rho, h'')$ of the modified Bellman equations have $\rho = \rho^\star$, with an example provided in [Bertsekas, 2018, Example 5.1.1]. Also mentioned there, a sufficient condition for $\rho = \rho^\star$ is the existence of a simultaneously argmaxing policy. Therefore, not all solutions of the modified Bellman equations admit a simultaneously argmaxing policy (with [Bertsekas, 2018, Example 5.1.1] necessarily being an example, and another example provided below), so the proof of [Puterman, 1994, Theorem 9.4.1] does not hold for all solutions to the modified Bellman equations. We note that the statement of [Puterman, 1994, Theorem 9.4.1] in Lee and Ryu [2024] apparently allows a general solution to the modified Bellman equations, although in Lee and Ryu [2024] the definition of a solution to the modified Bellman equation adds the condition that a simultaneously argmaxing policy must exist.

*Proof of Lemma 3.3.* We have that

$$\left\|\mathcal{T}^{(n)}(h_0) - n\rho^\star - h\right\|_\infty \le \left\|\mathcal{T}^{(n)}(h_0) - \mathcal{T}^{(n)}(h)\right\|_\infty + \left\|\mathcal{T}^{(n)}(h) - n\rho^\star - h\right\|_\infty$$

by the triangle inequality. Since $h$ satisfies both the modified and unmodified Bellman equations, by Lemma D.2 (used $n$ times) we have

$$\left\|\mathcal{T}^{(n)}(h) - n\rho^\star - h\right\|_\infty = \left\|\mathcal{T}^{(n-1)}\left(\mathcal{T}(h)\right) - n\rho^\star - h\right\|_\infty$$

$$= \left\|\mathcal{T}^{(n-1)}\left(h + \rho^\star\right) - n\rho^\star - h\right\|_\infty$$

$$= \left\|\mathcal{T}^{(n-1)}\left(h\right) - (n-1)\rho^\star - h\right\|_\infty$$

$$\vdots$$

$$= \left\|\mathcal{T}^{(0)}\left(h\right) - (0)\rho^\star - h\right\|_\infty$$

$$= \|h - h\|_\infty = 0.$$

By nonexpansiveness of $\mathcal{T}$, it is easy to see that $\left\|\mathcal{T}^{(n)}(h_0) - \mathcal{T}^{(n)}(h)\right\|_\infty \le \|h_0 - h\|_\infty$. Combining all these calculations yields the desired result. $\square$

## D.1 Constructing solutions to the modified Bellman equations

In this subsection we demonstrate how solutions to the modified Bellman equations $h$ may have $\|h\|_{\mathrm{sp}}$ on the order of $\frac{1}{\Delta}$ and arbitrarily large, even when constructed from a vector $h^{\pi^\star}$ solving the unmodified Bellman equations which has $\left\|h^{\pi^\star}\right\|_{\mathrm{sp}} \leq O(1)$.

The following lemma demonstrates this construction. This is a non-asymptotic version of Puterman [1994, Proposition 9.1.1] which was originally shown by Denardo and Fox [1968].

**Lemma D.3.** *Letting* $h = h^{\pi^\star} + \dfrac{\left\|h^{\pi^\star}\right\|_{\mathrm{sp}} + 1}{\Delta} \rho^\star$, $h$ *satisfies the modified Bellman equation* (7) *with* $\rho = \rho^\star$.

*Proof.* Since we have

$$M(r + Ph) \geq M^{\pi^\star}(r + Ph)$$
$$= M^{\pi^\star}\left(r + P\left(h^{\pi^\star} + \frac{\left\|h^{\pi^\star}\right\|_{\mathrm{sp}} + 1}{\Delta}\rho^\star\right)\right)$$
$$= r_{\pi^\star} + P_{\pi^\star}h^{\pi^\star} + \frac{\left\|h^{\pi^\star}\right\|_{\mathrm{sp}} + 1}{\Delta}P_{\pi^\star}\rho^\star$$
$$= \rho^\star + h^{\pi^\star} + \frac{\left\|h^{\pi^\star}\right\|_{\mathrm{sp}} + 1}{\Delta}\rho^\star$$
$$= \rho^\star + h$$

since $P_{\pi^\star}\rho^\star = \rho^\star$ and $r_{\pi^\star} + P_{\pi^\star}h^{\pi^\star} = \rho^\star + h^{\pi^\star}$, it suffices to show that for all deterministic policies $\pi$ that $M^\pi(r + Ph) \leq \rho^\star + h$. If $P_{s\pi(s)}\rho^\star = \rho^\star(s)$, then since $h^{\pi^\star}$ satisfies the unmodified Bellman equation (1b) (with $\rho = \rho^\star$), we have

$$e_s^\top M^\pi(r + Ph) = r(s, \pi(s)) + P_{s\pi(s)}h$$
$$= r(s, \pi(s)) + P_{s\pi(s)}h^{\pi^\star} + \frac{\left\|h^{\pi^\star}\right\|_{\mathrm{sp}} + 1}{\Delta}P_{s\pi(s)}\rho^\star$$
$$= r(s, \pi(s)) + P_{s\pi(s)}h^{\pi^\star} + \frac{\left\|h^{\pi^\star}\right\|_{\mathrm{sp}} + 1}{\Delta}\rho^\star(s)$$
$$\leq \max_{a \in \mathcal{A}} r(s, a) + P_{sa}h^{\pi^\star} + \frac{\left\|h^{\pi^\star}\right\|_{\mathrm{sp}} + 1}{\Delta}\rho^\star(s)$$
$$= \max_{a \in \mathcal{A}: P_{sa}\rho^\star = \rho^\star(s)} r(s, a) + P_{sa}h^{\pi^\star} + \frac{\left\|h^{\pi^\star}\right\|_{\mathrm{sp}} + 1}{\Delta}\rho^\star(s)$$
$$= \rho^\star(s) + h^{\pi^\star}(s) + \frac{\left\|h^{\pi^\star}\right\|_{\mathrm{sp}} + 1}{\Delta}\rho^\star(s)$$
$$= \rho^\star(s) + h(s)$$

as desired. Now we consider the case where $P_{s\pi(s)}\rho^\star < \rho^\star(s)$. Letting $a^\star \in \operatorname{argmax}_{a \in \mathcal{A}: P_{sa}\rho^\star = \rho^\star(s)} r(s, a) + P_{sa}h^{\pi^\star}$, first we calculate that

$$r(s, \pi(s)) + P_{s\pi(s)}h^{\pi^\star} - \max_{a \in \mathcal{A}: P_{sa}\rho^\star = \rho^\star(s)}\left(r(s, a) + P_{sa}h^{\pi^\star}\right)$$
$$= r(s, \pi(s)) - r(s, a^\star) + (P_{s\pi(s)} - P_{sa^\star})h^{\pi^\star}$$
$$\leq 1 + \left\|h^{\pi^\star}\right\|_{\mathrm{sp}}. \tag{11}$$

Also since $P_{s\pi(s)}\rho^\star < \rho^\star(s)$, by definition of $\Delta$ we have $\rho^\star(s) - P_{s\pi(s)}\rho^\star \geq \Delta$, and thus

$$e_s^\top M^\pi(r + Ph) = r(s, \pi(s)) + P_{s\pi(s)}h$$

$$= r(s, \pi(s)) + P_{s\pi(s)}h^{\pi^\star} + \frac{\left\|h^{\pi^\star}\right\|_{\mathrm{sp}} + 1}{\Delta}P_{s\pi(s)}\rho^\star$$

$$\leq r(s, \pi(s)) + P_{s\pi(s)}h^{\pi^\star} + \frac{\left\|h^{\pi^\star}\right\|_{\mathrm{sp}} + 1}{\Delta}(\rho^\star(s) - \Delta)$$

$$\leq \max_{a\in\mathcal{A}:P_{sa}\rho^\star=\rho^\star(s)} r(s, a) + P_{sa}h^{\pi^\star} + 1 + \left\|h^{\pi^\star}\right\|_{\mathrm{sp}} + \frac{\left\|h^{\pi^\star}\right\|_{\mathrm{sp}} + 1}{\Delta}(\rho^\star(s) - \Delta)$$

$$= \rho^\star(s) + h^{\pi^\star}(s) + \frac{\left\|h^{\pi^\star}\right\|_{\mathrm{sp}} + 1}{\Delta}\rho^\star(s)$$

$$= \rho^\star(s) + h(s)$$

making use of (11) in the second inequality. $\square$

Now we show a simple example where $\mathsf{T}_{\mathrm{drop}}, \left\|h^{\pi^\star}\right\|_{\mathrm{sp}} \leq O(1)$ and $\|h\|_{\mathrm{sp}} \approx \frac{1}{\Delta}$ is arbitrarily large.

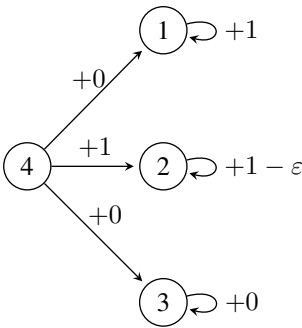

Figure 2: A four-state MDP parameterized by $\varepsilon > 0$. Each arrow represents a (deterministic) action and is annotated with its reward. Only state 4 has multiple actions.

**Theorem D.4.** *Fix $\varepsilon > 0$. Then for the MDP in the above Figure 2,*

1. $\mathsf{T}_{\mathrm{drop}} = 1$.

2. $\left\|h^{\pi^\star}\right\|_{\mathrm{sp}} = 1$.

3. $\frac{1}{\Delta} = \frac{1}{\varepsilon}$.

4. *Letting $\mathcal{H}$ denote the set of all solutions to the modified Bellman equations,*

$$\inf\left\{\left\|h^{\pi^\star} + c\rho^\star\right\|_{\mathrm{sp}} : c \in \mathbb{R}, (\rho^\star, h^{\pi^\star} + c\rho^\star) \in \mathcal{H}\right\} \geq \frac{1}{\varepsilon}.$$

*Proof.* It is immediate to see that

$$\rho^\star = \begin{bmatrix} 1 \\ 1 - \varepsilon \\ 0 \\ 1 \end{bmatrix}.$$

This implies that $\frac{1}{\Delta} = \frac{1}{\varepsilon}$ (by considering the actions in state 4). It is also trivial to see that $\mathsf{T}_{\mathrm{drop}} = 1$, since only state 4 has any "gain-dropping" actions available, and they all lead to states other than state 4 after 1 step. Since necessarily $P_{\pi^\star}^\infty h^{\pi^\star} = \mathbf{0}$, this implies that the absorbing states $s = 1, 2, 3$

must have $h^{\pi^\star}(s) = 0$. Combining with the unmodified Bellman equation (which $h^{\pi^\star}$ must satisfy) implies that

$$h^{\pi^\star} = \begin{bmatrix} 0 \\ 0 \\ 0 \\ -1 \end{bmatrix},$$

so we have $\left\| h^{\pi^\star} \right\|_{\mathrm{sp}} = 1$. Finally, for $(\rho^\star, h^{\pi^\star} + c\rho^\star)$ to satisfy the modified Bellman equation we must have

$$0 + c(1) + h^{\pi^\star}(1) \geq 1 + c(1 - \varepsilon) + h^{\pi^\star}(2)$$
$$\Longleftrightarrow 0 + c(1) + 0 \geq 1 + c(1 - \varepsilon) + 0$$
$$\Longleftrightarrow c \geq \frac{1}{\varepsilon}$$

which implies $h^{\pi^\star}(1) + c\rho^\star \geq \frac{1}{\varepsilon}$ for all solutions of this form to the modified Bellman equations, and furthermore $h^{\pi^\star}(3) + c\rho^\star(3) = 0 + c0 = 0$, which implies the span of $h^{\pi^\star} + c\rho^\star$ is at least $\frac{1}{\varepsilon}$. $\quad\square$

## E    Proof of Proposition 1.1

*Proof of Proposition 1.1.* We have

$$\left\| V^{\widehat{\pi}} - V^\star \right\|_\infty \leq \frac{2}{1 - \gamma} \left\| V_t - V^\star \right\|_\infty \leq \frac{2}{1 - \gamma} \gamma^n \left\| V_0 - V^\star \right\|_\infty \leq \frac{2}{(1 - \gamma)^2} \gamma^n$$

where the inequality $\left\| V^{\widehat{\pi}} - V^\star \right\|_\infty \leq \frac{2}{1-\gamma} \left\| V_t - V^\star \right\|_\infty$ is due to Singh and Yee [1994]. To ensure the above quantity is $\leq 1$, we need

$$\frac{2}{(1 - \gamma)^2} \gamma^n \leq 1$$
$$\Longleftrightarrow \gamma^n \leq \frac{(1 - \gamma)^2}{2}$$
$$\Longleftrightarrow n \log(\gamma) \leq \log\left( \frac{(1 - \gamma)^2}{2} \right)$$
$$\Longleftrightarrow n \geq \frac{\log\left( \frac{(1-\gamma)^2}{2} \right)}{\log \gamma} = \frac{\log\left( \frac{2}{(1-\gamma)^2} \right)}{\log(1/\gamma)}$$
$$\Longleftarrow n \geq \frac{\log\left( \frac{2}{(1-\gamma)^2} \right)}{1 - \gamma}$$

where for the final implication, we use that $\log(1/\gamma) \geq 1 - \gamma$ for any $\gamma$. Finally, with the stated choice of $\gamma$, the RHS above is

$$\frac{n}{2 \log(n)} \log\left( 2\frac{n^2}{4 \log^2(n)} \right) \leq \frac{n}{2 \log(n)} \log\left( \frac{n^2}{\log^2(n)} \right) \leq \frac{n}{2 \log(n)} 2 \log(n) = n$$

as desired. Finally, we can apply [Zurek and Chen, 2025b, Proof of Theorem 6] to give that

$$\rho^\pi \geq \rho^\star - (1 - \gamma) \left( 3\mathsf{B} + 3 \left\| h^{\pi^\star} \right\|_{\mathrm{sp}} + 2 \right) \mathbf{1}.$$

$\square$

## F    Proofs for sensitivity analysis for multichain MDPs

*Proof of Lemma 2.1.* We prove the more general statement that for any policy $\pi$ and any vectors $\rho, h \in \mathbb{R}^{\mathcal{S}}$, we have

$$\rho^\pi - \rho = H_{P_\pi}(P_\pi - I)\rho + P_\pi^\infty(r_\pi + P_\pi h - \rho - h).$$

(Then we can substitute $\rho = \rho^\star$ to obtain the statement of Lemma 2.1.)

Since $H_{P_\pi}(P_\pi - I) = P_\pi^\infty - I$, we have

$$H_{P_\pi}(P_\pi - I)\rho = (P_\pi^\infty - I)\rho.$$

Also since $P_\pi^\infty(P_\pi - I) = \mathbf{0}$ and $\rho^\pi = P_\pi^\infty r_\pi$, we have

$$P_\pi^\infty(r_\pi + P_\pi h - \rho - h) = P_\pi^\infty(r_\pi - \rho) = \rho^\pi - P_\pi^\infty \rho.$$

Combining these two calculations, we have

$$\begin{aligned}
\rho^\pi - \rho &= (\rho^\pi - P_\pi^\infty \rho) + (P_\pi^\infty \rho - \rho) \\
&= P_\pi^\infty(r_\pi + P_\pi h - \rho - h) + H_{P_\pi}(P_\pi - I)\rho.
\end{aligned}$$

$\square$

**Lemma F.1.**      *1. For any policy $\pi$, all states $s$ such that $e_s^\top P_\pi \rho^\star - \rho^\star(s) < 0$ are transient.*

  *2. For any policy $\pi$, $\mathsf{T}_{\mathrm{drop}}^\pi$ is finite.*

  *3. $\mathsf{T}_{\mathrm{drop}}$ is finite.*

*Proof.* For the first statement, suppose some state $s$ is recurrent under the Markov chain $P_\pi$. Then all states in the support of $e_s^\top P_\pi$ must also be recurrent and in the same maximal closed recurrent class. By Zurek and Chen [2024b, Lemma 17], any states $s', s''$ in the same recurrent class have $\rho^\star(s') = \rho^\star(s'')$, so we must have $e_s^\top P_\pi \rho^\star = \rho^\star(s)$. Therefore by contraposition, if $e_s^\top P_\pi \rho^\star - \rho^\star(s) < 0$ then $s$ must be transient.

For the second statement, if for some $s \in \mathcal{S}$ a policy $\pi$ has nonzero probability of taking some action $a$ such that $P_{sa}\rho^\star < \rho^\star$, then we must have $P_\pi \rho^\star < \rho^\star$. Therefore almost surely $\mathbb{I}(P_{S_t, A_t}\rho^\star < \rho^\star(S_t)) \leq \mathbb{I}(e_{S_t}^\top P_\pi \rho^\star < \rho^\star(S_t))$, which implies

$$\mathsf{T}_{\mathrm{drop}}^\pi = \max_{s \in \mathcal{S}} \mathbb{E}_s^\pi \left[ \sum_{t=0}^\infty \mathbb{I}\left(P_{S_t, A_t}\rho^\star < \rho^\star(S_t)\right) \right] \leq \max_{s \in \mathcal{S}} \mathbb{E}_s^\pi \left[ \sum_{t=0}^\infty \mathbb{I}\left(e_{S_t}^\top P_\pi \rho^\star < \rho^\star(S_t)\right) \right].$$

Hence $\mathsf{T}_{\mathrm{drop}}^\pi$ is bounded by the expected amount of time spent by the Markov chain $P_\pi$ in states which are transient under $P_\pi$ (maximized over all starting states), and since the expected amount of time spent in transient states in a finite Markov chain is finite [Durrett, 2019], we must have that $\mathsf{T}_{\mathrm{drop}}^\pi$ is finite.

For the final statement, since there are only a finite number of policies in $\Pi^{\mathrm{MD}}$ and $\mathsf{T}_{\mathrm{drop}}^\pi < \infty$ for each $\pi \in \Pi^{\mathrm{MD}}$, we have that $\mathsf{T}_{\mathrm{drop}} = \max_{\in \Pi^{\mathrm{MD}}} \mathsf{T}_{\mathrm{drop}}^\pi < \infty$. $\square$

**Lemma F.2.** *Suppose that $\pi \in \Pi^{\mathrm{MD}}$ is a deterministic policy. Then*

$$\mathsf{T}_{\mathrm{drop}}^\pi = \max_{s \in \mathcal{S}} \mathbb{E}_s^\pi \left[ \sum_{t=0}^\infty \mathbb{I}\left(e_{S_t}^\top P_\pi \rho^\star < \rho^\star(S_t)\right) \right].$$

*Proof.* Note that for a deterministic policy we have $A_t = \pi(S_t)$, and thus $P_{S_t, A_t}\rho^\star = P_{S_t, \pi(S_t)}\rho^\star = e_{S_t}^\top P_\pi \rho^\star$. Therefore

$$\mathsf{T}_{\mathrm{drop}}^\pi = \max_{s \in \mathcal{S}} \mathbb{E}_s^\pi \left[ \sum_{t=0}^\infty \mathbb{I}\left(P_{S_t, A_t}\rho^\star < \rho^\star(S_t)\right) \right] = \max_{s \in \mathcal{S}} \mathbb{E}_s^\pi \left[ \sum_{t=0}^\infty \mathbb{I}\left(e_{S_t}^\top P_\pi \rho^\star < \rho^\star(S_t)\right) \right].$$

$\square$

**Lemma F.3.** *Fix a deterministic policy $\pi \in \Pi^{\mathrm{MD}}$, let $s, s' \in \mathcal{S}$ be two states, and suppose $s'$ is transient under $P_\pi$. Then*

$$(H_{P_\pi})_{ss'} = \mathbb{E}_s^\pi \left[ \sum_{t=0}^\infty \mathbb{I}(S_t = s') \right], \tag{12}$$

*or in words, $(H_{P_\pi})_{ss'}$ is the expected number of visits of state $s'$ when following policy $\pi$ and starting at state $s$. Therefore*

$$\max_{s \in \mathcal{S}} \sum_{s' : s' \text{ is transient under } P_\pi} (H_{P_\pi})_{ss'} = \mathsf{B}^\pi \tag{13}$$

*and*

$$\max_{s \in \mathcal{S}} \sum_{s' : e_{s'}^\top P_\pi \rho^\star < \rho^\star(s')} (H_{P_\pi})_{ss'} = \mathsf{T}_{\text{drop}}^\pi. \tag{14}$$

*Proof.* Since $s'$ is transient, we have that $e_s^\top P_\pi^\infty e_{s'} = 0$. Thus

$$
\begin{aligned}
(H_{P_\pi})_{ss'} &= \underset{T \to \infty}{\text{C-lim}} \sum_{k=0}^{T-1} e_s^\top (P_\pi^k - P_\pi^\infty) e_{s'} \\
&= \lim_{N \to \infty} \frac{1}{N} \sum_{T=1}^{N} \sum_{k=0}^{T-1} e_s^\top (P_\pi^k - P_\pi^\infty) e_{s'} \\
&= \lim_{N \to \infty} \frac{1}{N} \sum_{T=1}^{N} \sum_{k=0}^{T-1} e_s^\top P_\pi^k e_{s'} \qquad\qquad s' \text{ is transient} \\
&= \lim_{T \to \infty} \sum_{k=0}^{T-1} e_s^\top P_\pi^k e_{s'} \\
&= \mathbb{E}_s^\pi \left[ \sum_{t=0}^{\infty} \mathbb{I}(S_t = s') \right]
\end{aligned}
$$

where the second-last equality follows from elementary analysis arguments, or can be seen using the fact that the Cesaro limit of a convergent sequence is simply its usual limit, and the sequence $\left( \sum_{k=0}^{T-1} e_s^\top P_\pi^k e_{s'} \right)_{T \in \mathbb{N}}$ is convergent since all terms are nonnegative and the infinite sum is finite since $s'$ is transient [Durrett, 2019].

The next two statements follow immediately (noting that by Lemma F.1, all states such that $e_{s'}^\top P_\pi \rho^\star < \rho^\star(s')$ are transient in $P_\pi$, and using the formula for $\mathsf{T}_{\text{drop}}^\pi$ given in Lemma F.2 for $\pi \in \Pi^{\text{MD}}$). $\qquad\square$

*Proof of Corollary 2.2.* Combining Lemma 2.1 with the triangle inequality we have

$$\|\rho^\pi - \rho^\star\|_\infty \le \|H_{P_\pi}(P_\pi \rho^\star - \rho^\star)\|_\infty + \|P_\pi^\infty (r_\pi + P_\pi h - \rho^\star - h)\|_\infty .$$

Since $\|P_\pi^\infty\|_{\infty \to \infty} \le 1$, we can bound

$$
\begin{aligned}
\|P_\pi^\infty (r_\pi + P_\pi h - \rho^\star - h)\|_\infty &\le \|P_\pi^\infty\|_{\infty \to \infty} \|r_\pi + P_\pi h - \rho^\star - h\|_\infty \\
&\le \|r_\pi + P_\pi h - \rho^\star - h\|_\infty \\
&= \|\mathcal{T}^\pi(h) - h - \rho^\star\|_\infty .
\end{aligned}
$$

To bound $\|H_{P_\pi}(P_\pi \rho^\star - \rho^\star)\|_\infty$, using Lemma F.3 we have

$$
\begin{aligned}
\|H_{P_\pi}(P_\pi \rho^\star - \rho^\star)\|_\infty &= \max_{s \in \mathcal{S}} \left| e_s^\top H_{P_\pi}(P_\pi \rho^\star - \rho^\star) \right| \\
&= \max_{s \in \mathcal{S}} \left| \sum_{s' \in \mathcal{S}} (H_{P_\pi})_{ss'} \, e_{s'}^\top (P_\pi \rho^\star - \rho^\star) \right| \\
&\overset{(i)}{=} \max_{s \in \mathcal{S}} \left| \sum_{s' \in \mathcal{S}:\, e_{s'}^\top(P_\pi \rho^\star - \rho^\star) < 0} (H_{P_\pi})_{ss'} \, e_{s'}^\top (P_\pi \rho^\star - \rho^\star) \right| \\
&\overset{(ii)}{=} \max_{s \in \mathcal{S}} \sum_{s' \in \mathcal{S}:\, e_{s'}^\top(P_\pi \rho^\star - \rho^\star) < 0} (H_{P_\pi})_{ss'} \, e_{s'}^\top (\rho^\star - P_\pi \rho^\star) \\
&\leq \max_{s \in \mathcal{S}} \sum_{s' \in \mathcal{S}:\, e_{s'}^\top(P_\pi \rho^\star - \rho^\star) < 0} (H_{P_\pi})_{ss'} \, \|\rho^\star - P_\pi \rho^\star\|_\infty \\
&\overset{(iii)}{=} \mathsf{T}_{\mathrm{drop}}^\pi \, \|\rho^\star - P_\pi \rho^\star\|_\infty .
\end{aligned}
$$

In $(i)$ we used that $e_{s'}^\top(P_\pi \rho^\star - \rho^\star) \neq 0$ is equivalent to $e_{s'}^\top(P_\pi \rho^\star - \rho^\star) < 0$, since $e_{s'}^\top(P_\pi \rho^\star - \rho^\star) > 0$ is impossible. In $(ii)$ we used the fact that $(H_{P_\pi})_{ss'} \geq 0$ for any state $s'$ which is transient under $P_\pi$ (implied by Lemma F.3) and the fact that the states $s'$ such that $e_{s'}^\top P_\pi \rho^\star < \rho^\star(s)$ are transient by Lemma F.1, as well as the fact that $e_{s'}^\top(\rho^\star - P_\pi \rho^\star) \geq 0$ again. In $(iii)$ we again use Lemma F.3. $\quad\square$

**Lemma F.4.** *For any $x, y \in \mathbb{R}^\mathcal{S}$ and policy $\pi$ such that $P_\pi^\infty x = P_\pi^\infty y = \mathbf{0}$, we have that*

$$
x \leq y \implies H_{P_\pi} x \leq H_{P_\pi} y.
$$

*Proof.* To show the desired elementwise inequality, it suffices to fix $s \in \mathcal{S}$ and show $e_s^\top H_{P_\pi} x \leq e_s^\top H_{P_\pi} y$. By an almost identical calculation to that of Lemma F.3 we have that

$$
e_s^\top H_{P_\pi} x = \lim_{T \to \infty} \sum_{k=0}^{T-1} e_s^\top P_\pi^k x \quad \text{and} \quad e_s^\top H_{P_\pi} y = \lim_{T \to \infty} \sum_{k=0}^{T-1} e_s^\top P_\pi^k y.
$$

Since $P_\pi^k$ is a stochastic matrix, $x \leq y$ implies that

$$
\sum_{k=0}^{T-1} e_s^\top P_\pi^k x \leq \sum_{k=0}^{T-1} e_s^\top P_\pi^k y
$$

for any $T$, and thus

$$
e_s^\top H_{P_\pi} x = \lim_{T \to \infty} \sum_{k=0}^{T-1} e_s^\top P_\pi^k x \leq \lim_{T \to \infty} \sum_{k=0}^{T-1} e_s^\top P_\pi^k y = e_s^\top H_{P_\pi} y.
$$

$\quad\square$

**Lemma F.5.** *For any fixed policy $\pi \in \Pi^{\mathrm{MD}}$, we have $\mathsf{T}_{\mathrm{drop}}^\pi \leq \mathsf{B}^\pi$. Consequently $\mathsf{T}_{\mathrm{drop}} \leq \mathsf{B}$.*

*Proof.* First we prove the statement for a fixed policy $\pi$. For any set $X \subseteq \mathcal{S}$ let $e_X \in \mathbb{R}^\mathcal{S}$ be a vector such that

$$
e_X(s) = \begin{cases} 1 & s \in X \\ 0 & \text{otherwise} \end{cases}.
$$

Let $X$ be the set of states which are transient under $P_\pi$ and let $Y$ be the set of states $s$ such that $e_s^\top P_\pi \rho^\star < \rho^\star(s)$. By Lemma F.1 we have $Y \subseteq X$, so elementwise $e_Y \leq e_X$. By Lemma F.3 we have that $\mathsf{B}^\pi = \max_{s \in \mathcal{S}} e_s^\top H_{P_\pi} e_X$ and that $\mathsf{T}_{\mathrm{drop}}^\pi = \max_{s \in \mathcal{S}} e_s^\top H_{P_\pi} e_Y$ (also using Lemma F.2). Then using Lemma F.4, we have that

$$
\mathsf{T}_{\mathrm{drop}}^\pi = \max_{s \in \mathcal{S}} e_s^\top H_{P_\pi} e_Y \leq \max_{s \in \mathcal{S}} e_s^\top H_{P_\pi} e_X = \mathsf{B}^\pi.
$$

Now taking the maximum over all stationary deterministic policies, we have that

$$\mathsf{T}_{\text{drop}} = \sup_{\pi \in \Pi^{\text{MD}}} \mathsf{T}_{\text{drop}}^{\pi} \leq \sup_{\pi \in \Pi^{\text{MD}}} \mathsf{B}^{\pi} = \mathsf{B}.$$

$\square$

**Lemma F.6.** *For any fixed policy* $\pi \in \Pi^{\text{MD}}$, *we have* $\mathsf{T}_{\text{drop}}^{\pi} \leq \frac{1}{\Delta}$. *Consequently* $\mathsf{T}_{\text{drop}} \leq \frac{1}{\Delta}$.

*Proof.* First we fix a policy $\pi \in \Pi^{\text{MD}}$ and show that $\mathsf{T}_{\text{drop}}^{\pi} \leq \frac{1}{\Delta}$. From the proof of Lemma 2.1, we have

$$H_{P_\pi}(\rho^\star - P_\pi \rho^\star) = \rho^\star - P_\pi^\infty \rho^\star$$

and also $\rho^\star - P_\pi^\infty \rho^\star \leq \mathbf{1}$ since $\rho^\star \geq \mathbf{0}$ so $P_\pi^\infty \rho^\star \geq \mathbf{0}$. We define $e_Y$ in the same way as in the proof of Lemma F.5, that is we let $Y$ be the set of states $s$ such that $e_s^\top P_\pi \rho^\star < \rho^\star(s)$ and $e_Y$ be the indicator vector for this set. Then (since $P_\pi \rho^\star \leq \rho^\star$ so $e_s^\top P_\pi \rho^\star > \rho^\star(s)$ for some $s$ is impossible) we have that $e_Y \circ (\rho^\star - P_\pi \rho^\star) = \rho^\star - P_\pi \rho^\star$. Then by the definition of $\Delta$ we have that

$$\rho^\star - P_\pi \rho^\star = e_Y \circ (\rho^\star - P_\pi \rho^\star) \geq \Delta e_Y.$$

Since both sides of this elementwise inequality are only supported on states which are transient under $P_\pi$ (so $P_\pi^\infty(\rho^\star - P_\pi \rho^\star) = \mathbf{0}$ and $P_\pi^\infty e_Y = \mathbf{0}$), by Lemma F.4 we have that

$$H_{P_\pi}(\rho^\star - P_\pi \rho^\star) \geq \Delta H_{P_\pi} e_Y.$$

Combining all these inequalities we have that

$$\Delta H_{P_\pi} e_Y \leq H_{P_\pi}(\rho^\star - P_\pi \rho^\star) = \rho^\star - P_\pi^\infty \rho^\star \leq \mathbf{1}$$

which implies

$$\Delta \mathsf{T}_{\text{drop}}^{\pi} = \Delta \max_{s \in \mathcal{S}} e_s^\top H_{P_\pi} e_Y \leq 1$$

(using that $\mathsf{T}_{\text{drop}}^{\pi} = \max_{s \in \mathcal{S}} e_s^\top H_{P_\pi} e_Y$ as shown in the proof of Lemma F.5). Rearranging we have that $\mathsf{T}_{\text{drop}}^{\pi} \leq \frac{1}{\Delta}$ as desired, and then taking the supremum over all $\pi \in \Pi^{\text{MD}}$ we have that $\mathsf{T}_{\text{drop}} = \sup_{\pi \in \Pi^{\text{MD}}} \mathsf{T}_{\text{drop}}^{\pi} \leq \frac{1}{\Delta}$. $\square$

**Lemma F.7.** *Fix a starting state* $s_0$. *For any infinite sequence of policies* $\pi_0, \pi_1, \ldots$, *letting* $S_0 = s_0, A_t \sim \pi_t(S_t), S_{t+1} \sim P(\cdot \mid S_t, A_t)$ *be its induced distribution over trajectories, and letting* $\mathbb{E}_{s_0}^{(\pi_t)_{t=0}^\infty}$ *denote the corresponding expectation, we have that*

$$\mathbb{E}_{s_0}^{(\pi_t)_{t=0}^\infty} \left[ \sum_{k=0}^\infty \overline{r}(S_k, A_k) \right] \leq \mathsf{T}_{\text{drop}}$$

*where* $\overline{r}(s, a) = \mathbb{I}(P_{sa} \rho^\star < \rho^\star)$.

*Proof.* Note that [Puterman, 1994, Assumption 7.1.1] is satisfied, since the reward function $\overline{r}$ is non-negative. Therefore by [Puterman, 1994, Theorem 7.1.9] there exists a stationary deterministic optimal policy $\widetilde{\pi}$ for the expected total reward problem. Therefore we have

$$\mathbb{E}_{s_0}^{(\pi_t)_{t=0}^\infty} \left[ \sum_{k=0}^\infty \overline{r}(S_k, A_k) \right] \leq \mathbb{E}_{s_0}^{\widetilde{\pi}} \left[ \sum_{k=0}^\infty \overline{r}(S_k, A_k) \right] \leq \mathsf{T}_{\text{drop}}^{\widetilde{\pi}} \leq \mathsf{T}_{\text{drop}}$$

as desired. $\square$

## G   Proofs of coupling-based results

*Proof of Lemma 3.1.*  We prove this result by induction. Note that by convention $\Lambda_0 = 0$ (since it is an empty summation). Therefore the $t = 0$ case holds because we have $x_0 = y_0 = y_0 + 0\rho = y_0 + \Lambda_0\rho$.

Now we suppose that for some $t$ we have $x_t = y_t + \Lambda_t\rho$. Then (also using the definition that $x_0 = y_0$) we can calculate that

$$
\begin{aligned}
x_{t+1} &= (1 - \beta_{t+1})x_0 + \beta_{t+1}\mathcal{L}(x_t) \\
&= (1 - \beta_{t+1})y_0 + \beta_{t+1}\mathcal{L}(y_t + \Lambda_t\rho) \\
&= (1 - \beta_{t+1})y_0 + \beta_{t+1}\mathcal{L}(y_t) + \beta_{t+1}\Lambda_t\rho \\
&= (1 - \beta_{t+1})y_0 + \beta_{t+1}\overline{\mathcal{L}}(y_t) + \beta_{t+1}(\Lambda_t + 1)\rho \\
&= y_{t+1} + \beta_{t+1}(\Lambda_t + 1)\rho.
\end{aligned}
$$

Now it remains to show that $\Lambda_{t+1} = \beta_{t+1}(\Lambda_t + 1)$, which is true because we have

$$
\beta_{t+1}(\Lambda_t + 1) = \beta_{t+1}\left( \sum_{i=1}^{t}\prod_{j=i}^{t}\beta_j + 1 \right) = \sum_{i=1}^{t}\prod_{j=i}^{t+1}\beta_j + \beta_{t+1} = \sum_{i=1}^{t+1}\prod_{j=i}^{t+1}\beta_j = \Lambda_{t+1}.
$$

$\square$

*Proof of Corollary 3.2.*  Fixing some policy $\pi \in \Pi^{\mathrm{MR}}$, we have that $\overline{\mathcal{T}}^\pi(h) := \mathcal{T}^\pi(h) - \rho^\pi = r_\pi + P_\pi h - \rho^\pi$ is an affine operator, and furthermore it has fixed point $h^\pi$ since by the Poisson/Bellman equation we have

$$
\overline{\mathcal{T}}^\pi(h^\pi) = r_\pi + P_\pi h^\pi - \rho^\pi = \rho^\pi + h^\pi - \rho^\pi = h^\pi.
$$

The nonexpansiveness of $\overline{\mathcal{T}}^\pi$ with respect to $\|\cdot\|_\infty$ also follows immediately from that of $\mathcal{T}^\pi$, since

$$
\left\| \overline{\mathcal{T}}^\pi(h) - \overline{\mathcal{T}}^\pi(h') \right\|_\infty = \|\mathcal{T}^\pi(h) - \mathcal{T}^\pi(h')\|_\infty \le \|h - h'\|_\infty.
$$

Therefore we can directly apply [Contreras and Cominetti, 2022, Theorem 4.6] to conclude that, for any initial point $y_0 = h_0$, the Halpern-iteration-generated iterate sequence $(y_t)_{t=0}^\infty$ where

$$
y_{t+1} = (1 - \beta_{t+1})y_0 + \beta_{t+1}\overline{\mathcal{T}}^\pi(y_t)
$$

and $\beta_t = 1 - \frac{1}{t+1}$ satisfies

$$
\left\| \overline{\mathcal{T}}^\pi(y_t) - y_t \right\|_\infty \le \frac{2}{t+1}\|y_0 - h^\pi\|_\infty = \frac{2}{t+1}\|h_0 - h^\pi\|_\infty. \tag{15}
$$

To apply Lemma 3.1, notice that for any $\alpha \in \mathbb{R}$ we have that

$$
\mathcal{T}^\pi(x + \alpha\rho^\pi) = r_\pi + P_\pi(x + \alpha\rho^\pi) = r_\pi + P_\pi x + \alpha P_\pi\rho^\pi = r_\pi + P_\pi x + \alpha\rho^\pi = \mathcal{T}^\pi(x) + \alpha\rho^\pi
$$

due to the key fact that $P_\pi\rho^\pi = \rho^\pi$. Now applying Lemma 3.1, we have that $h_t = y_t + \Lambda_t\rho^\pi$ (where $h_t$ is defined in the statement of Corollary 3.2). Therefore

$$
\begin{aligned}
\|\mathcal{T}^\pi(h_t) - h_t - \rho^\pi\|_\infty &= \|\mathcal{T}^\pi(y_t + \Lambda_t\rho^\pi) - (y_t + \Lambda_t\rho^\pi) - \rho^\pi\|_\infty \\
&= \|\mathcal{T}^\pi(y_t) + \Lambda_t\rho^\pi - (y_t + \Lambda_t\rho^\pi) - \rho^\pi\|_\infty \\
&= \|\mathcal{T}^\pi(y_t) - y_t - \rho^\pi\|_\infty \\
&= \left\| \overline{\mathcal{T}}^\pi(y_t) - y_t \right\|_\infty \\
&\le \frac{2}{t+1}\|h_0 - h^\pi\|_\infty
\end{aligned}
$$

using the commutativity property in the second equality, and applying (15) in the final inequality. $\square$

We note that $h^\pi$ could be replaced with any other fixed point of $\overline{\mathcal{T}}^\pi(h)$ with an unchanged analysis.

# H Proofs of results for fixed-point methods under restricted commutativity

Here we prove the main results of Subsection 3.2.

## H.1 Abstract setting

Recall that a seminorm $\|\cdot\|$ is a function $X \to [0, \infty)$ which satisfies the triangle inequality and positive homogeneity, that is, for all $x, y \in X$ (where $X$ is a vector space) and $\alpha \in \mathbb{R}$,

$$\|x + y\| \leq \|x\| + \|y\|$$

and

$$\|\alpha x\| = |\alpha| \, \|x\| \,.$$

Our results in this section hold for the abstract setting that $\mathcal{L} : X \to X$ is nonexpansive with respect to a seminorm $\|\cdot\|$, meaning

$$\|\mathcal{L}(x) - \mathcal{L}(y)\| \leq \|x - y\|$$

for all $x, y \in X$. We assume that $\mathcal{L}$ satisfies

$$\mathcal{L}(x^\star + \alpha \rho) = x^\star + (\alpha + 1)\rho \tag{16}$$

for some $\rho, x^\star \in X$ and for all $\alpha \in \mathbb{R}$. We also define the shifted operator $\overline{\mathcal{L}}(x) := \mathcal{L}(x) - \rho$. Note the above implies that $x^\star$ is a fixed-point of $\overline{\mathcal{L}}$. Also $\overline{\mathcal{L}}$ is clearly nonexpansive since $\overline{\mathcal{L}}(x) - \overline{\mathcal{L}}(x') = \mathcal{L}(x) - \mathcal{L}(x')$ for any $x, x' \in X$. First we restate Algorithm 1 for this general setting.

---

**Algorithm 4** Approximately Shifted Halpern Iteration (General)

---
**input:** Number of iterations $n$ for each phase, initial point $x_0$
1: **for** $t = 0, \ldots, n - 1$ **do**
2: $\quad x_{t+1} = \mathcal{L}(x_t)$
3: **end for**
4: Let $z_0 = x_n$, form gain estimate $\widehat{\rho} = \frac{1}{n}(x_n - x_0)$ $\quad \triangleright$ Halpern iteration phase; begins with $x_n$
5: Form approximate shifted operator $\widehat{\mathcal{L}}$ as $\widehat{\mathcal{L}}(z) := \mathcal{L}(z) - \widehat{\rho}$
6: **for** $t = 0, \ldots, n - 1$ **do**
7: $\quad z_{t+1} = (1 - \beta_{t+1})z_0 + \beta_{t+1}\widehat{\mathcal{L}}(z_t)$ where $\beta_t = 1 - \frac{2}{t+2}$
8: **end for**
9: **return** $z_n$

---

For the rest of the proofs we treat $n$ and $x_0$, the inputs to Algorithm 4, as fixed. Before beginning the proofs, we define $z^\star = x^\star + n\rho$, and we note that $z^\star$ is also a fixed-point of $\overline{\mathcal{L}}$ (by taking $\alpha = n$ in (16)). For all integers $t \geq 1$ define

$$\Lambda_t = \sum_{i=1}^{t} \prod_{j=i}^{t} \beta_j = \beta_t \beta_{t-1} \cdots \beta_1 + \beta_t \beta_{t-1} \cdots \beta_2 + \cdots + \beta_t \beta_{t-1} + \beta_t. \tag{17}$$

We also assume that $\beta_0 = 0$ and let $\Lambda_0 = 0$.

**Lemma H.1.** *If* $\beta_t = 1 - \frac{2}{t+2}$ *then* $\Lambda_t = \frac{t}{3}$.

*Proof.* We show this by induction. By definition $\Lambda_0 = 0$, so the desired formula holds for $t = 0$. Note that by definition of $\Lambda_t$, for all $t \geq 1$ we have $\Lambda_t = \beta_t(1 + \Lambda_{t-1})$. Therefore supposing the desired formula holds for some $\Lambda_t$ where $t \geq 0$, we have

$$\Lambda_{t+1} = \beta_{t+1}(1 + \Lambda_t) = \left(1 - \frac{2}{t+3}\right)\left(1 + \frac{t}{3}\right) = \frac{t+1}{t+3}\frac{t+3}{3} = \frac{t+1}{3}$$

as desired. $\qquad \square$

**Lemma H.2.**

$$\|\widehat{\rho} - \rho\| \leq \frac{2}{n} \|x_0 - x^\star\|$$

*and*

$$\|z_0 - z^\star\| \leq \|x_0 - x^\star\| .$$

*Proof.* First we show that

$$\left\| \mathcal{L}^{(n)}(x_0) - x_0 - n\rho \right\| \leq 2 \|x_0 - x^\star\| . \tag{18}$$

We calculate that

$$\begin{aligned}
\left\| \mathcal{L}^{(n)}(x_0) - x_0 - n\rho \right\| &\leq \left\| \mathcal{L}^{(n)}(x_0) - x^\star - n\rho \right\| + \|x^\star - x_0\| \\
&= \left\| \mathcal{L}^{(n)}(x_0) - \mathcal{L}^{(n)}(x^\star) \right\| + \|x^\star - x_0\| \\
&\leq \|x^\star - x_0\| + \|x^\star - x_0\| .
\end{aligned}$$

Next, we have

$$\widehat{\rho} = \frac{\mathcal{L}^{(n)}(x_0) - x_0}{n},$$

so using (18),

$$\|\widehat{\rho} - \rho\| = \left\| \frac{\mathcal{L}^{(n)}(x_0) - n\rho - x_0}{n} \right\| \leq \frac{2 \|x_0 - x^\star\|}{n}.$$

Next, since $z^\star = x^\star + n\rho = \mathcal{L}^{(n)}(x^\star)$, we have by nonexpansiveness that

$$\|z_0 - z^\star\| = \left\| \mathcal{L}^{(n)}(x_0) - \mathcal{L}^{(n)}(x^\star) \right\| \leq \|x_0 - x^\star\| .$$

$\square$

Now we show the Halpern iterates remain bounded using the shifted Bellman operator $\widehat{\mathcal{L}}$.

**Lemma H.3.** *For all $t \leq n$,*

$$\|z_t - z^\star\| \leq \frac{5}{3} \|x_0 - x^\star\| .$$

*Proof.* We show the stronger bound

$$\|z_t - z^\star\| \leq \|x_0 - x^\star\| + 2\frac{\Lambda_t}{n} \|x_0 - x^\star\|$$

by induction. This obviously holds for $t = 0$ since $\Lambda_0 = 0$ and by using Lemma H.2. Letting $t$ be arbitrary and assuming the above statement holds for $t - 1$, we have

$$\begin{aligned}
\|z_t - z^\star\| &= \left\| (1 - \beta_t)z_0 + \beta_t \widehat{\mathcal{L}}(z_{t-1}) - z^\star \right\| \\
&\leq \left\| (1 - \beta_t)z_0 + \beta_t \overline{\mathcal{L}}(z_{t-1}) - z^\star \right\| + \beta_t \|\rho - \widehat{\rho}\| \\
&\leq (1 - \beta_t) \|z_0 - z^\star\| + \beta_t \left\| \overline{\mathcal{L}}(z_{t-1}) - z^\star \right\| + \beta_t \|\rho - \widehat{\rho}\| \\
&\leq (1 - \beta_t) \|z_0 - z^\star\| + \beta_t \|z_{t-1} - z^\star\| + \beta_t \|\rho - \widehat{\rho}\| \\
&\leq (1 - \beta_t) \|z_0 - z^\star\| + \beta_t \|z_{t-1} - z^\star\| + \frac{2\beta_t}{n} \|x_0 - x^\star\| \\
&\leq (1 - \beta_t) \|z_0 - z^\star\| + \beta_t \left( \|x_0 - x^\star\| + 2\frac{\Lambda_{t-1}}{n} \|x_0 - x^\star\| \right) + \frac{2\beta_t}{n} \|x_0 - x^\star\| \\
&= (1 - \beta_t + \beta_t) \|z_0 - z^\star\| + 2\frac{\beta_t \Lambda_{t-1} + \beta_t}{n} \|x_0 - x^\star\| \\
&= \|z_0 - z^\star\| + 2\frac{\Lambda_t}{n} \|x_0 - x^\star\| \\
&\leq \|x_0 - x^\star\| + 2\frac{\Lambda_t}{n} \|x_0 - x^\star\|
\end{aligned}$$

where the first two inequalities are by triangle inequality, then we use nonexpansiveness of $\overline{\mathcal{L}}$ and the fact that $\overline{\mathcal{L}}(z^\star) = z^\star$, then we use Lemma H.2 to bound $\|\rho - \widehat{\rho}\|$, then we use the inductive hypothesis, in the penultimate equality we use that $\Lambda_t = \beta_t(1 + \Lambda_{t-1})$, and in the final inequality we use Lemma H.2 to bound $\|z_0 - z^\star\|$.

We conclude by using Lemma H.1 and the fact that $t \leq n$. $\qquad\square$

**Lemma H.4.** *For all $t \leq n$,*

$$\left\|\overline{\mathcal{L}}(z_t) - z_0\right\| \leq \frac{8}{3} \left\|x_0 - x^\star\right\|.$$

*Proof.*

$$
\begin{aligned}
\left\|\overline{\mathcal{L}}(z_t) - z_0\right\| &= \left\|\overline{\mathcal{L}}(z_t) - \overline{\mathcal{L}}(z^\star) + z^\star - z_0\right\| \\
&\leq \left\|\overline{\mathcal{L}}(z_t) - \overline{\mathcal{L}}(z^\star)\right\| + \left\|z^\star - z_0\right\| \\
&\leq \left\|z_t - z^\star\right\| + \left\|z^\star - z_0\right\| \\
&\leq \frac{8}{3} \left\|x_0 - x^\star\right\|
\end{aligned}
$$

using Lemmas H.3 and H.2 in the final step. $\qquad\square$

Now we can prove the main theorem.

**Theorem H.5.**

$$\left\|\overline{\mathcal{L}}(z_n) - z_n\right\| \leq \frac{13 + \frac{35}{n} + \frac{20}{n^2}}{n} \left\|x_0 - x^\star\right\|.$$

*Proof.* For any $t \geq 1$, we have

$$
\begin{aligned}
z_{t+1} - z_t &= (1 - \beta_{t+1})z_0 + \beta_{t+1}\widehat{\mathcal{L}}(z_t) - (1 - \beta_t)z_0 + \beta_t\widehat{\mathcal{L}}(z_{t-1}) \\
&= (\beta_t - \beta_{t+1})z_0 + (\beta_{t+1} - \beta_t)\widehat{\mathcal{L}}(z_t) + \beta_t\left(\widehat{\mathcal{L}}(z_t) - \widehat{\mathcal{L}}(z_{t-1})\right) \\
&= (\beta_{t+1} - \beta_t)\left(\widehat{\mathcal{L}}(z_t) - z_0\right) + \beta_t\left(\widehat{\mathcal{L}}(z_t) - \widehat{\mathcal{L}}(z_{t-1})\right) \\
&= (\beta_{t+1} - \beta_t)\left(\overline{\mathcal{L}}(z_t) - z_0\right) + \beta_t\left(\overline{\mathcal{L}}(z_t) - \overline{\mathcal{L}}(z_{t-1})\right) + (\beta_{t+1} - \beta_t)(\rho - \widehat{\rho}).
\end{aligned}
$$

Therefore

$$
\begin{aligned}
\left\|z_{t+1} - z_t\right\| &\leq (\beta_{t+1} - \beta_t)\left\|\overline{\mathcal{L}}(z_t) - z_0\right\| + \beta_t\left\|\overline{\mathcal{L}}(z_t) - \overline{\mathcal{L}}(z_{t-1})\right\| + (\beta_{t+1} - \beta_t)\left\|\rho - \widehat{\rho}\right\| \\
&\leq (\beta_{t+1} - \beta_t)\frac{8}{3}\left\|x_0 - x^\star\right\| + \beta_t\left\|z_t - z_{t-1}\right\| + (\beta_{t+1} - \beta_t)\frac{2}{n}\left\|x_0 - x^\star\right\| \\
&= \beta_t\left\|z_t - z_{t-1}\right\| + (\beta_{t+1} - \beta_t)\left(\frac{8}{3} + \frac{2}{n}\right)\left\|x_0 - x^\star\right\| \qquad\qquad (19)
\end{aligned}
$$

using the triangle inequality, and then Lemmas H.4 and H.2. We also note that

$$
\begin{aligned}
\left\|z_1 - z_0\right\| &= \left\|(1 - \beta_1)z_0 - \beta_1\widehat{\mathcal{L}}(z_0) - z_0\right\| = \beta_1\left\|z_0 - \widehat{\mathcal{L}}(z_0)\right\| \\
&\leq \beta_1\left\|z_0 - \overline{\mathcal{L}}(z_0)\right\| + \beta_1\left\|\overline{\mathcal{L}}(z_0) - \widehat{\mathcal{L}}(z_0)\right\| = \beta_1\left\|z_0 - \overline{\mathcal{L}}(z_0)\right\| + \beta_1\left\|\rho - \widehat{\rho}\right\| \\
&\leq \beta_1 2\left\|z_0 - z^\star\right\| + \beta_1\left\|\rho - \widehat{\rho}\right\| \\
&\leq \frac{4}{3}\left\|x_0 - x^\star\right\|
\end{aligned}
$$

using Lemma H.2, the fact that $\beta_1 = \frac{1}{3}$, and also the inequality

$$\left\|z_0 - \overline{\mathcal{L}}(z_0)\right\| \leq \left\|z_0 - z^\star\right\| + \left\|\overline{\mathcal{L}}(z^\star) - \overline{\mathcal{L}}(z_0)\right\| \leq 2\left\|z_0 - z^\star\right\|$$

since $z^\star$ is a fixed-point of $\overline{\mathcal{L}}$. Now supposing inductively that $\|z_t - z_{t-1}\| \leq \frac{\frac{16}{3} + \frac{4}{n}}{t+1} \|x_0 - x^\star\|$ (which we have just confirmed for the case that $t = 0$), we can use the bound (19) to obtain that

$$\|z_{t+1} - z_t\| \leq \beta_t \|z_t - z_{t-1}\| + (\beta_{t+1} - \beta_t)\left(\frac{8}{3} + \frac{2}{n}\right)\|x_0 - x^\star\|$$

$$\leq \beta_t \frac{\frac{16}{3} + \frac{4}{n}}{t+1}\|x_0 - x^\star\| + (\beta_{t+1} - \beta_t)\left(\frac{8}{3} + \frac{2}{n}\right)\|x_0 - x^\star\|$$

$$= \left(\frac{\beta_t}{t+1} + \frac{\beta_{t+1} - \beta_t}{2}\right)\left(\frac{16}{3} + \frac{4}{n}\right)\|x_0 - x^\star\|$$

$$\leq \frac{1}{t+2}\left(\frac{16}{3} + \frac{4}{n}\right)\|x_0 - x^\star\|$$

where the final inequality is because

$$\frac{\beta_t}{t+1} + \frac{\beta_{t+1} - \beta_t}{2} = \frac{1 - \frac{2}{t+2}}{t+1} + \frac{1 - \frac{2}{t+3} - 1 + \frac{2}{t+2}}{2}$$

$$= \frac{t}{(t+1)(t+2)} + \frac{1}{(t+2)(t+3)}$$

$$\leq \frac{t+1}{(t+1)(t+2)} = \frac{1}{t+2}.$$

Therefore by induction we have that

$$\|z_t - z_{t-1}\| \leq \frac{\frac{16}{3} + \frac{4}{n}}{t+1}\|x_0 - x^\star\|$$

for all $t \geq 1$.

Now relating $\|z_t - z_{t-1}\|$ to the fixed-point error, we have

$$z_{t+1} - z_t = (1 - \beta_{t+1})z_0 + \beta_{t+1}\widehat{\mathcal{L}}(z_t) - z_t$$

$$= (1 - \beta_{t+1})(z_0 - z_t) + \beta_{t+1}(\widehat{\mathcal{L}}(z_t) - z_t)$$

$$= (1 - \beta_{t+1})(z_0 - z_t) + \beta_{t+1}(\overline{\mathcal{L}}(z_t) - z_t) + \beta_{t+1}(\rho - \widehat{\rho})$$

which implies

$$\beta_{t+1}(\overline{\mathcal{L}}(z_t) - z_t) = (z_{t+1} - z_t) - \beta_{t+1}(\rho - \widehat{\rho}) - (1 - \beta_{t+1})(z_0 - z_t)$$

so by triangle inequality

$$\beta_{t+1}\left\|\overline{\mathcal{L}}(z_t) - z_t\right\| \leq \|z_{t+1} - z_t\| + \beta_{t+1}\|\rho - \widehat{\rho}\| + (1 - \beta_{t+1})\|z_0 - z_t\|$$

$$\leq \|z_{t+1} - z_t\| + \beta_{t+1}\|\rho - \widehat{\rho}\| + (1 - \beta_{t+1})\left(\|z_0 - z^\star\| + \|z_t - z^\star\|\right)$$

$$\leq \frac{\frac{16}{3} + \frac{4}{n}}{t+2}\|x_0 - x^\star\| + \beta_{t+1}\frac{2}{n}\|x_0 - x^\star\| + (1 - \beta_{t+1})\frac{8}{3}\|x_0 - x^\star\|$$

$$= \left(\frac{\frac{16}{3} + \frac{4}{n}}{t+2} + \frac{t+1}{t+3}\frac{2}{n} + \frac{2}{t+3}\frac{8}{3}\right)\|x_0 - x^\star\|.$$

Rearranging and simplifying, we obtain

$$\left\|\overline{\mathcal{L}}(z_t) - z_t\right\| \leq \frac{1}{\beta_{t+1}}\left(\frac{\frac{16}{3} + \frac{4}{n}}{t+2} + \frac{t+1}{t+3}\frac{2}{n} + \frac{2}{t+3}\frac{8}{3}\right)\|x_0 - x^\star\|$$

$$= \frac{t+3}{t+1}\left(\frac{\frac{16}{3} + \frac{4}{n}}{t+2} + \frac{t+1}{t+3}\frac{2}{n} + \frac{2}{t+3}\frac{8}{3}\right)\|x_0 - x^\star\|$$

$$= \frac{2}{n}\|x_0 - x^\star\| + \frac{t+3}{t+1}\left(\frac{\frac{16}{3} + \frac{4}{n}}{t+2} + \frac{1}{t+3}\frac{16}{3}\right)\|x_0 - x^\star\|$$

$$\leq \frac{2}{n}\|x_0 - x^\star\| + \frac{t+3}{t+1}\left(\frac{1}{t+2} + \frac{1}{t+3}\right)\left(\frac{16}{3} + \frac{4}{n}\right)\|x_0 - x^\star\|$$

$$= \frac{2}{n}\|x_0 - x^\star\| + \frac{t+3}{t+1}\frac{2t+5}{(t+2)(t+3)}\left(\frac{16}{3} + \frac{4}{n}\right)\|x_0 - x^\star\|.$$

Now substituting $t = n$, and using that $\frac{n+2.5}{n+2} = \frac{1+\frac{2.5}{n}}{1+\frac{2}{n}} \le 1 + \frac{2.5}{n}$, we have

$$
\begin{aligned}
\left\| \overline{\mathcal{L}}(z_t) - z_t \right\| &\le \frac{2}{n} \|x_0 - x^\star\| + \frac{2 + \frac{5}{n}}{n+1} \left( \frac{16}{3} + \frac{4}{n} \right) \|x_0 - x^\star\| \\
&\le \frac{2 + \frac{32}{3} + \frac{80}{3n} + \frac{8}{n} + \frac{20}{n^2}}{n} \|x_0 - x^\star\| \\
&\le \frac{13 + \frac{35}{n} + \frac{20}{n^2}}{n} \|x_0 - x^\star\| .
\end{aligned}
$$

$\square$

## H.2 Undiscounted value iteration setting

Here we specialize to the case of $\mathcal{L} = \mathcal{T}$ and prove Theorems 3.4 and 3.5.

First we develop a result to control the first term of Corollary 2.2.

**Lemma H.6.** *Let $h$ satisfy both the modified and unmodified Bellman equations. Suppose that $z \in \mathbb{R}^{\mathcal{S}}$ satisfies $\|z - \Gamma \rho^\star - h\|_\infty \le B$ for some $\Gamma, B > 0$. Then if $\pi$ is greedy with respect to $r + Pz$ ($\mathcal{T}^\pi(z) = \mathcal{T}(z)$), we have*

$$
P_\pi \rho^\star \ge \rho^\star - \frac{2B}{\Gamma} \mathbf{1}.
$$

*Proof.* Since $h$ satisfies both the modified and unmodified Bellman equations, by Lemma D.1 there exists a policy $\widetilde{\pi}$ which attains the maximum in both simultaneously, meaning $\mathcal{T}^{\widetilde{\pi}}(h) = \rho^\star + h$ and $P_{\widetilde{\pi}} \rho^\star = \rho^\star$. Then we can calculate

$$
\begin{aligned}
P_\pi \rho^\star &= \frac{1}{\Gamma} P_\pi (\Gamma \rho^\star) \\
&= \frac{1}{\Gamma} P_\pi (\Gamma \rho^\star + h - h) \\
&\ge \frac{1}{\Gamma} P_\pi (z - \|z - \Gamma \rho^\star - h\|_\infty \mathbf{1} - h) \\
&\ge \frac{1}{\Gamma} P_\pi (z - B\mathbf{1} - h) \\
&\ge \frac{1}{\Gamma} (P_\pi (z - B\mathbf{1}) + r_\pi - \rho^\star - h) \\
&= \frac{1}{\Gamma} (r_\pi + P_\pi z - B\mathbf{1} - \rho^\star - h) \\
&\ge \frac{1}{\Gamma} (r_{\pi^\star} + P_{\pi^\star} z - B\mathbf{1} - \rho^\star - h) \\
&\ge \frac{1}{\Gamma} (r_{\pi^\star} + P_{\pi^\star} (\Gamma \rho^\star + h) - P_{\pi^\star} \|z - \Gamma \rho^\star - h\|_\infty \mathbf{1} - B\mathbf{1} - \rho^\star - h) \\
&\ge \frac{1}{\Gamma} (r_{\pi^\star} + P_{\pi^\star} (\Gamma \rho^\star + h) - 2B\mathbf{1} - \rho^\star - h) \\
&= \rho^\star - \frac{2B}{\Gamma} \mathbf{1}
\end{aligned}
$$

where we used the fact that (by the modified Bellman equation (7)) $r_\pi + P_\pi h = \mathcal{T}^\pi(h) \le \mathcal{T}(h) = \rho^\star + h$ which implies $-P_\pi h \ge r_\pi - \rho^\star - h$, and then the fact that since $\pi$ is greedy with respect to $r + Pz$ we have $r_\pi + P_\pi z = \mathcal{T}^\pi(z) = \mathcal{T}(z) \ge \mathcal{T}^{\widetilde{\pi}}(z) = r_{\widetilde{\pi}} + P_{\widetilde{\pi}} z$, and then in the final equality we used that $P_{\widetilde{\pi}} \rho^\star = \rho^\star$ and that $r_{\widetilde{\pi}} + P_{\widetilde{\pi}} h = \rho^\star + h$. $\square$

**Lemma H.7.** *Let $h$ satisfy both the modified and unmodified Bellman equations. The policy $\widehat{\pi}$ output by Algorithm 1 satisfies*

$$
\|P_{\widehat{\pi}} \rho^\star - \rho^\star\|_\infty \le \frac{\frac{10}{3} \|h_0 - h\|_\infty}{n}.
$$

*Proof.* Specializing $\|\cdot\| = \|\cdot\|_\infty$, $\mathcal{L} = \mathcal{T}$, and $x^\star = h$, by Lemma H.3, we have

$$\|z_n - n\rho^\star - h\|_\infty = \|z_n - z^\star\|_\infty \leq \frac{5}{3}\|x_0 - x^\star\|_\infty = \frac{5}{3}\|h_0 - h\|_\infty.$$

By Lemma H.6 this implies that

$$P_{\widehat{\pi}}\rho^\star \geq \rho^\star - \frac{\frac{10}{3}\|h_0 - h\|_\infty}{n}.$$

Also we trivially have $P_{\widehat{\pi}}\rho^\star \leq \rho^\star$. $\qquad\square$

**Lemma H.8.** *Let $h$ satisfy both the modified and unmodified Bellman equations. The vector $z_n$ and policy $\widehat{\pi}$ output by Algorithm 1 satisfy*

$$\left\|\mathcal{T}^{\widehat{\pi}}(z_n) - \rho^\star - z_n\right\|_\infty \leq \frac{13 + \frac{35}{n} + \frac{20}{n^2}}{n}\|h_0 - h\|_\infty.$$

*Proof.* This follows immediately from Theorem H.5 by specializing $\|\cdot\| = \|\cdot\|_\infty$, $\mathcal{L} = \mathcal{T}$, and $x^\star = h$. We note that $\widehat{\pi}$ is defined so that $\mathcal{T}(z_n) = \mathcal{T}^{\widehat{\pi}}(z_n)$. $\qquad\square$

*Proof of Theorem 3.4.* This follows immediately by using Lemmas H.7 and H.8 to bound the two terms in Corollary 2.2. $\qquad\square$

*Proof of Theorem 3.5.* Under the assumption that $n \geq \frac{4\|h_0 - h\|_\infty}{\Delta}$, we have by Lemma H.7 that

$$\|P_{\widehat{\pi}}\rho^\star - \rho^\star\|_\infty \leq \frac{\frac{10}{3}\|h_0 - h\|_\infty}{n} \leq \frac{10}{12}\Delta < \Delta.$$

By the definition of $\Delta$ and since $\widehat{\pi}$ is a deterministic policy, this implies that $\|P_{\widehat{\pi}}\rho^\star - \rho^\star\|_\infty = 0$.

Using this fact, as well as Lemma H.8 to bound the other term in Corollary 2.2, we immediately obtain the desired conclusion. $\qquad\square$

# I  Proof of Theorem 4.2

The following theorem on the performance of Halpern iteration for general normed spaces is due to [Sabach and Shtern, 2017, Lemma 5]. Their result is presented for Euclidean spaces but the proof holds for general norms. See [Contreras and Cominetti, 2022, Remark 2] for more discussion of this bound.

**Theorem I.1.** *Suppose $\mathcal{L}$ is nonexpansive with respect to some norm $\|\cdot\|$. Let $x^\star$ be a fixed point of $\mathcal{L}$, and fix some initial point $x_0$. For all $t = 1, 2, \ldots$, let $x_{t+1} = (1 - \beta_{t+1})x_0 + \beta_{t+1}\mathcal{L}(x_t)$, where $\beta_t = 1 - \frac{2}{t+2}$. Then for all $t \geq 0$ we have*

$$\|\mathcal{L}(x_t) - x_t\| \leq \frac{4}{t+1}\|x_0 - x^\star\|.$$

*Proof of Theorem 4.2.* Let $x^\star$ be the unique fixed point of $\mathcal{L}$. Let $E = \left\lfloor \frac{1}{1-\gamma} \right\rfloor - 1$, which is $\geq 0$ since $\frac{1}{1-\gamma} \geq 1$. Since Algorithm 2 generates iterates $x_0, \ldots, x_E$ in an identical manner to the Halpern iteration described in Theorem I.1, and also if $\mathcal{L}$ is $\gamma$-contractive for $\gamma < 1$ then it is clearly nonexpansive, by Theorem I.1 we have that

$$\|\mathcal{L}(x_t) - x_t\| \leq \frac{4}{t+1}\|x_0 - x^\star\|$$

for all $t = 0, \ldots, E$. In particular

$$\|\mathcal{L}(x_E) - x_E\| \leq \frac{4}{E+1}\|x_0 - x^\star\| \leq \frac{4}{\left\lfloor \frac{1}{1-\gamma} \right\rfloor}\|x_0 - x^\star\|.$$

Now for $t = E+1, E+2, \ldots$, since we have $x_t = \mathcal{L}(x_{t-1})$, it is immediate from $\gamma$-contractivity of $\mathcal{L}$ that

$$\|\mathcal{L}(x_t) - x_t\| = \|\mathcal{L}(x_t) - \mathcal{L}(x_{t-1})\| \le \gamma \|x_t - x_{t-1}\| = \gamma \|\mathcal{L}(x_{t-1}) - x_{t-1}\|.$$

Therefore for all $t = E+1, E+2, \ldots$ we have

$$\|\mathcal{L}(x_t) - x_t\| \le \gamma^{t-E} \|\mathcal{L}(x_E) - x_E\| \le \gamma^{t-E} \frac{4}{\left\lfloor \frac{1}{1-\gamma} \right\rfloor} \|x_0 - x^\star\|.$$

Now we relate these bounds to the quantity $\frac{\gamma^t}{\sum_{i=0}^t \gamma^i}$. First we note the useful fact that

$$\sup_{\gamma \in (0,1)} \gamma^{-\frac{1}{1-\gamma}+1} = e. \tag{20}$$

which we will verify later. Now for the case that $t = 0, \ldots, E$, we can bound

$$\frac{\frac{4}{t+1}}{\frac{\gamma^t}{\sum_{i=0}^t \gamma^i}} = 4 \frac{\sum_{i=0}^t \gamma^i}{t+1} \gamma^{-t} \le 4 \frac{\sum_{i=0}^t 1}{t+1} \gamma^{-E} \le 4e$$

using (20) in the final inequality, since $\gamma^{-\left\lfloor \frac{1}{1-\gamma} \right\rfloor + 1} \le \gamma^{-\frac{1}{1-\gamma}+1}$. Next for the case that $t \ge E+1$, we can bound

$$\frac{\frac{4}{\left\lfloor \frac{1}{1-\gamma} \right\rfloor} \gamma^{t-E}}{\frac{\gamma^t}{\sum_{i=0}^t \gamma^i}} = 4 \left( \frac{1}{\left\lfloor \frac{1}{1-\gamma} \right\rfloor} \sum_{i=0}^t \gamma^i \right) \gamma^{-E} \le 4e \frac{\frac{1}{1-\gamma}}{\left\lfloor \frac{1}{1-\gamma} \right\rfloor}$$

where in the final inequality we used (20) again and that $\sum_{i=0}^t \gamma^i \le \sum_{i=0}^\infty \gamma^i = \frac{1}{1-\gamma}$.

Therefore we have shown that

$$\|\mathcal{L}(x_t) - x_t\| \le \begin{cases} \frac{4}{t+1} \|x_0 - x^\star\| & t \le E \\ 4 \frac{1}{\left\lfloor \frac{1}{1-\gamma} \right\rfloor} \gamma^{t-E} \|x_0 - x^\star\| & t > E \end{cases} \le 4e \frac{\frac{1}{1-\gamma}}{\left\lfloor \frac{1}{1-\gamma} \right\rfloor} \frac{\gamma^t}{\sum_{i=0}^t \gamma^i} \|x_0 - x^\star\|. \tag{21}$$

In the case that $\frac{1}{1-\gamma}$ is an integer, we thus have that

$$\|\mathcal{L}(x_t) - x_t\| \le \begin{cases} \frac{4}{t+1} \|x_0 - x^\star\| & t \le E \\ 4(1-\gamma)\gamma^{t-E} \|x_0 - x^\star\| & t > E \end{cases} \le 4e \frac{\gamma^t}{\sum_{i=0}^t \gamma^i} \|x_0 - x^\star\|. \tag{22}$$

In the case that $\frac{1}{1-\gamma}$ is not an integer, using the fact that $\left\lfloor \frac{1}{1-\gamma} \right\rfloor \ge \frac{1}{2} \frac{1}{1-\gamma}$ since $\frac{1}{1-\gamma} \ge 1$, we obtain

$$\|\mathcal{L}(x_t) - x_t\| \le \begin{cases} \frac{4}{t+1} \|x_0 - x^\star\| & t \le E \\ 8(1-\gamma)\gamma^{t-E} \|x_0 - x^\star\| & t > E \end{cases} \le 8e \frac{\gamma^t}{\sum_{i=0}^t \gamma^i} \|x_0 - x^\star\|.$$

Now it remains to verify (20). Note $\gamma \mapsto \gamma^{-\frac{1}{1-\gamma}+1}$ is a smooth function on $(0,1)$, and it suffices to show that the function $\gamma \mapsto \log(\gamma^{-\frac{1}{1-\gamma}+1})$ is non-decreasing and approaches 1 as $\gamma \to 1$ (since $\log$ is monotone increasing). To show $\gamma \mapsto \log(\gamma^{-\frac{1}{1-\gamma}+1})$ is non-decreasing it suffices to show that its derivative is $\ge 0$. We have

$$\frac{d}{d\gamma} \log(\gamma^{-\frac{1}{1-\gamma}+1}) = \frac{d}{d\gamma} \frac{\gamma}{\gamma-1} \log(\gamma) = \frac{1}{\gamma} \frac{\gamma}{\gamma-1} + \frac{\gamma-1-\gamma}{(\gamma-1)^2} \log(\gamma) = \frac{\gamma-1-\log(\gamma)}{(\gamma-1)^2}$$

and by Taylor's theorem applied to $\log(\gamma)$ about $\gamma = 1$, for any $\gamma \in (0,1)$ there exists some $\xi \in (0,1)$ such that $\log(\gamma) = 0 + 1(\gamma-1) - \xi^{-2}(\gamma-1)^2 < \gamma-1$, and so $\gamma - 1 - \log(\gamma) > 0$ for any $\gamma \in (0,1)$ and thus the function $\log(\gamma^{-\frac{1}{1-\gamma}+1})$ is nondecreasing. Next, to show that the limit of this function as $\gamma \to 1$ is 1, by L'Hopital's rule we have

$$\lim_{\gamma \to 1} \log(\gamma^{-\frac{1}{1-\gamma}+1}) = \lim_{\gamma \to 1} \frac{\gamma}{\gamma-1} \log(\gamma) = \lim_{\gamma \to 1} \frac{\log(\gamma)+1}{1} = 1$$

as desired. Thus $\sup_{\gamma \in (0,1)} \log(\gamma^{-\frac{1}{1-\gamma}+1}) = 1$, and so since $\log$ is monotone increasing we have that $\sup_{\gamma \in (0,1)} \gamma^{-\frac{1}{1-\gamma}+1} = \exp(1) = e$ as desired. $\qquad \square$

## J  Proofs of DMDP Results

### J.1  Gain approximation lemmas

In this section we develop several different bounds on the quantity $\left\|V_\gamma^\star - \frac{1}{1-\gamma}\rho^\star\right\|_\infty$.

**Lemma J.1.** *For any $\gamma \in (0,1)$, we have that*

$$\left\|V_\gamma^\star - \frac{1}{1-\gamma}\rho^\star\right\|_\infty \leq \left\|h^{\pi^*}\right\|_{\mathrm{sp}} + \mathsf{T}_{\mathrm{drop}} + \left\|h^{\pi^*}\right\|_{\mathrm{sp}}\mathsf{T}_{\mathrm{drop}}.$$

*Proof.* By Zurek and Chen [2024b, Lemma 20], $\left\|V_\gamma^{\pi^*} - \frac{1}{1-\gamma}\rho^\star\right\|_\infty \leq \left\|h^{\pi^*}\right\|_{\mathrm{sp}}$, which implies

$$V_\gamma^\star \geq V_\gamma^{\pi^*} \geq \frac{1}{1-\gamma}\rho^\star - \left\|h^{\pi^*}\right\|_{\mathrm{sp}}.$$

Thus it remains to upper-bound $V_\gamma^\star$. Let $\pi_\gamma^\star$ be a deterministic discounted optimal policy (such that $V_\gamma^\star = V_\gamma^{\pi_\gamma^\star}$), which is guaranteed to exist Puterman [1994]. From the unmodified Bellman equation (1b) (with $\rho = \rho^\star$), if $s \in \mathcal{S}$ satisfies $e_s^\top P_{\pi_\gamma^\star}\rho^\star = \rho^\star(s)$, then

$$\rho^\star(s) + h^{\pi^*}(s) = \max_{a \in \mathcal{A}: P_{sa}\rho = \rho(s)} r(s,a) + P_{sa}h^{\pi^*} \geq r(s,\pi_\gamma^\star(s)) + P_{s\pi_\gamma^\star(s)}h^{\pi^*} = r_{\pi_\gamma^\star}(s) + e_s^\top P_{\pi_\gamma^\star}h^{\pi^*}.$$

If instead we have that $e_s^\top P_{\pi_\gamma^\star}\rho^\star < \rho^\star(s)$, then instead we can bound

$$r_{\pi_\gamma^\star}(s) + e_s^\top P_{\pi_\gamma^\star}h^{\pi^*} - r_{\pi^\star}(s) - e_s^\top P_{\pi^\star}h^{\pi^*} \leq \|r\|_{\mathrm{sp}} + e_s^\top(P_{\pi_\gamma^\star} - P_{\pi^\star})h^{\pi^*} \leq 1 + \left\|h^{\pi^*}\right\|_{\mathrm{sp}},$$

and since $r_{\pi^\star}(s) + e_s^\top P_{\pi^\star}h^{\pi^*} = \rho^\star(s) + h^{\pi^*}(s)$, we have that

$$\rho^\star(s) + h^{\pi^*}(s) + 1 + \left\|h^{\pi^*}\right\|_{\mathrm{sp}} \geq r_{\pi_\gamma^\star}(s) + e_s^\top P_{\pi_\gamma^\star}h^{\pi^*}.$$

Thus, similarly to what we have done in above proofs, letting $Y$ be the set of states $s$ such that $e_s^\top P_{\pi_\gamma^\star}\rho^\star < \rho^\star(s)$, and letting $e_Y$ be the indicator vector for this set, combining the two cases we obtain the elementwise inequality

$$r_{\pi_\gamma^\star} + P_{\pi_\gamma^\star}h^{\pi^*} \leq \rho^\star + h^{\pi^*} + \left(1 + \left\|h^{\pi^*}\right\|_{\mathrm{sp}}\right)e_Y. \tag{23}$$

Thus using the fact that all entries of $(I - \gamma P_{\pi_\gamma^\star})^{-1}$ are nonnegative, we have that

$$V_\gamma^\star = (I - \gamma P_{\pi_\gamma^\star})^{-1}r_{\pi_\gamma^\star}$$
$$\leq (I - \gamma P_{\pi_\gamma^\star})^{-1}\rho^\star + (I - \gamma P_{\pi_\gamma^\star})^{-1}(I - P_{\pi_\gamma^\star})h^{\pi^*} + \left(1 + \left\|h^{\pi^*}\right\|_{\mathrm{sp}}\right)(I - \gamma P_{\pi_\gamma^\star})^{-1}e_Y. \tag{24}$$

Now we bound all the terms in (24). First, since $P_{\pi_\gamma^\star}\rho^\star \leq \rho^\star$, we can derive that

$$P_{\pi_\gamma^\star}^k\rho^\star = P_{\pi_\gamma^\star}^{k-1}P_{\pi_\gamma^\star}\rho^\star \leq P_{\pi_\gamma^\star}^{k-1}\rho^\star \leq \cdots \leq \rho^\star$$

for any integer $k \geq 0$, and thus

$$(I - \gamma P_{\pi_\gamma^\star})^{-1}\rho^\star = \sum_{k=0}^\infty \gamma^k P_{\pi_\gamma^\star}^k\rho^\star \leq \sum_{k=0}^\infty \gamma^k\rho^\star = \frac{1}{1-\gamma}\rho^\star.$$

Next, by Lemma K.1 we have that $(I - \gamma P_{\pi_\gamma^\star})^{-1}(I - P_{\pi_\gamma^\star})h^{\pi^*} \leq \left\|h^{\pi^*}\right\|_{\mathrm{sp}}\mathbf{1}$. Finally, as shown in Lemma F.1, $e_Y$ is only nonzero on states which are transient under the Markov chain $P_{\pi_\gamma^\star}$, which implies $P_\pi^\infty e_Y = 0$. Therefore for any $\gamma'$ we have that

$$(I - \gamma' P_{\pi_\gamma^\star})^{-1}e_Y = (I - \gamma' P_{\pi_\gamma^\star})^{-1}e_Y - \frac{1}{1-\gamma'}P_\pi^\infty e_Y$$

and so by Lattimore and Szepesvári [2020, Exercise 38.9] we have that

$$\lim_{\gamma'\uparrow 1}(I - \gamma' P_{\pi_\gamma^\star})^{-1} e_Y = \lim_{\gamma'\uparrow 1}(I - \gamma' P_{\pi_\gamma^\star})^{-1} e_Y - \frac{1}{1-\gamma'} P_\pi^\infty e_Y = H_{P_{\pi_\gamma^\star}} e_Y = \mathsf{T}_{\mathrm{drop}}^{\pi_\gamma^\star}$$

where the final equality is due to Lemma F.3. But we also have that $(I - \gamma' P_{\pi_\gamma^\star})^{-1} e_Y$ is (elementwise) non-decreasing as $\gamma' \uparrow 1$, so this implies that

$$(I - \gamma P_{\pi_\gamma^\star})^{-1} e_Y \leq \lim_{\gamma'\uparrow 1}(I - \gamma' P_{\pi_\gamma^\star})^{-1} e_Y = \mathsf{T}_{\mathrm{drop}}^{\pi_\gamma^\star}.$$

Finally using these three bounds in (24), we obtain that

$$V_\gamma^\star \leq \frac{1}{1-\gamma}\rho^\star + \left\|h^{\pi^\star}\right\|_{\mathrm{sp}} + \left(1 + \left\|h^{\pi^\star}\right\|_{\mathrm{sp}}\right)\mathsf{T}_{\mathrm{drop}}^{\pi_\gamma^\star} \leq \frac{1}{1-\gamma}\rho^\star + \left\|h^{\pi^\star}\right\|_{\mathrm{sp}} + \left(1 + \left\|h^{\pi^\star}\right\|_{\mathrm{sp}}\right)\mathsf{T}_{\mathrm{drop}}.$$

$\square$

**Lemma J.2.** *For any $\gamma \in (0,1)$, letting $h$ be any solution to both the modified and unmodified Bellman equations* (2b) *and* (1b), *we have that*

$$\left\|V_\gamma^\star - \frac{1}{1-\gamma}\rho^\star\right\|_\infty \leq \|h\|_{\mathrm{sp}}.$$

*Proof.* By Lemma D.1, there exists a policy $\pi$ attain both maximums in the modified Bellman equations simultaneously in the sense of (8) and (9), that is, $P_\pi \rho^\star = \rho^\star$ and $r_\pi + P_\pi h = \mathcal{T}^\pi(h) = \mathcal{T}(h) = \rho^\star + h$. Rearranging, this implies that $r_\pi = \rho^\star + (I - P_\pi)h$. Then we have

$$V_\gamma^\star \geq V_\gamma^\pi$$
$$= (I - \gamma P_\pi)^{-1} r_\pi$$
$$= (I - \gamma P_\pi)^{-1}\left(\rho^\star + (I - P_\pi)h\right)$$
$$= (I - \gamma P_\pi)^{-1}\rho^\star + (I - \gamma P_\pi)^{-1}(I - P_\pi)h.$$

Since $P_\pi \rho^\star = \rho^\star$, this implies that $P_\pi^t \rho^\star = \rho^\star$ for all $t \geq 0$, so we have that

$$(I - \gamma P_\pi)^{-1}\rho^\star = \sum_{t=0}^\infty \gamma^t P_\pi^t \rho^\star = \sum_{t=0}^\infty \gamma^t \rho^\star = \frac{1}{1-\gamma}\rho^\star.$$

Also $(I - \gamma P_\pi)^{-1}(I - P_\pi)h \geq -\|h\|_{\mathrm{sp}}\mathbf{1}$ by using Lemma K.1. Therefore we have shown that $V_\gamma^\star \geq \frac{1}{1-\gamma}\rho^\star - \|h\|_{\mathrm{sp}}\mathbf{1}$.

By the second modified Bellman equation (2b) we have that

$$\rho^\star + h = M(r + Ph) \geq M^{\pi_\gamma^\star}(r + Ph) = r_{\pi_\gamma^\star} + P_{\pi_\gamma^\star}h.$$

Rearranging we have that $r_{\pi_\gamma^\star} \leq \rho^\star + (I - P_{\pi_\gamma^\star})h$. Then by monotonicity of $(I - \gamma P_{P_{\pi_\gamma^\star}})^{-1}$ we have that

$$V_\gamma^\star = (I - \gamma P_{\pi_\gamma^\star})^{-1} r_{\pi_\gamma^\star}$$
$$\leq (I - \gamma P_{\pi_\gamma^\star})^{-1}\left(\rho^\star + (I - P_{\pi_\gamma^\star})h\right)$$
$$= (I - \gamma P_{\pi_\gamma^\star})^{-1}\rho^\star + (I - \gamma P_{\pi_\gamma^\star})^{-1}(I - P_{\pi_\gamma^\star})h.$$

Since $P_{\pi_\gamma^\star}\rho^\star \leq \rho^\star$, this implies that $P_{\pi_\gamma^\star}^t \rho^\star \leq \rho^\star$ for all $t \geq 0$, and we have that

$$(I - \gamma P_{\pi_\gamma^\star})^{-1}\rho^\star = \sum_{t=0}^\infty \gamma^t P_{\pi_\gamma^\star}^t \rho^\star \leq \sum_{t=0}^\infty \gamma^t \rho^\star = \frac{1}{1-\gamma}\rho^\star.$$

Again using Lemma K.1 we have that $(I - \gamma P_{\pi_\gamma^\star})^{-1}(I - P_{\pi_\gamma^\star})h \leq \|h\|_{\mathrm{sp}}\mathbf{1}$. Thus we have checked the other direction that $V_\gamma^\star \leq \frac{1}{1-\gamma}\rho^\star + \|h\|_{\mathrm{sp}}\mathbf{1}$, so combining with the above lower-bound and rearranging, we have that

$$\left\|V_\gamma^\star - \frac{1}{1-\gamma}\rho^\star\right\|_\infty \leq \|h\|_{\mathrm{sp}}.$$

$\square$

## J.2 Discounted reduction results

In this subsection we relate the terms appearing in Corollary 2.2 to those which are directly controlled by solving discounted MDPs, that is, the discounted fixed-point error $\|\mathcal{T}_\gamma(V) - V\|_\infty$ and the value error $\|V - V_\gamma^\star\|_\infty$.

**Lemma J.3.** *Suppose for some $V \in \mathbb{R}^S$ that some policy $\pi$ is $\varepsilon$-greedy with respect to $r + \gamma PV$, that is, $\mathcal{T}_\gamma^\pi(V) \geq \mathcal{T}_\gamma(V) - \varepsilon \mathbf{1}$. Then*

$$\|\mathcal{T}(V) - V - \rho^\star\|_\infty \leq \|P_\pi \rho^\star - \rho^\star\|_\infty + \|\mathcal{T}_\gamma(V) - V\|_\infty + (1-\gamma)\left\|V - \frac{1}{1-\gamma}\rho^\star\right\|_\infty + \varepsilon.$$

*In particular if $\pi$ is greedy with respect to $r + \gamma PV$ ($\varepsilon = 0$) then*

$$\|\mathcal{T}(V) - V - \rho^\star\|_\infty \leq \|P_\pi \rho^\star - \rho^\star\|_\infty + \|\mathcal{T}_\gamma(V) - V\|_\infty + (1-\gamma)\left\|V - \frac{1}{1-\gamma}\rho^\star\right\|_\infty.$$

*Proof.* We just need to prove the first statement, from which the second follows immediately. We can lower-bound $\mathcal{T}(V)$ as

$$\begin{aligned}
\mathcal{T}(V) &= M(r + PV) = M\left(r + \gamma PV + (1-\gamma)PV\right) \\
&\geq M\left(r + \gamma PV + (1-\gamma)P\frac{1}{1-\gamma}\rho^\star - (1-\gamma)P\left\|V - \frac{1}{1-\gamma}\rho^\star\right\|_\infty \mathbf{1}\right) \\
&= M\left(r + \gamma PV + P\rho^\star - (1-\gamma)\left\|V - \frac{1}{1-\gamma}\rho^\star\right\|_\infty \mathbf{1}\right) \\
&= M\left(r + \gamma PV + P\rho^\star\right) - (1-\gamma)\left\|V - \frac{1}{1-\gamma}\rho^\star\right\|_\infty \mathbf{1} \\
&\geq M^\pi\left(r + \gamma PV + P\rho^\star\right) - (1-\gamma)\left\|V - \frac{1}{1-\gamma}\rho^\star\right\|_\infty \mathbf{1} \\
&= M^\pi\left(r + \gamma PV\right) + P_\pi \rho^\star - (1-\gamma)\left\|V - \frac{1}{1-\gamma}\rho^\star\right\|_\infty \mathbf{1} \\
&= \mathcal{T}_\gamma^\pi(V) + P_\pi \rho^\star - (1-\gamma)\left\|V - \frac{1}{1-\gamma}\rho^\star\right\|_\infty \mathbf{1} \\
&= \mathcal{T}_\gamma(V) - \varepsilon \mathbf{1} + P_\pi \rho^\star - (1-\gamma)\left\|V - \frac{1}{1-\gamma}\rho^\star\right\|_\infty \mathbf{1}.
\end{aligned}$$

We upper-bound $\mathcal{T}(V)$ as

$$\begin{aligned}
\mathcal{T}(V) &= M(r + PV) = M\left(r + \gamma PV + (1-\gamma)PV\right) \\
&\leq M\left(r + \gamma PV + (1-\gamma)P\frac{1}{1-\gamma}\rho^\star + (1-\gamma)P\left\|V - \frac{1}{1-\gamma}\rho^\star\right\|_\infty \mathbf{1}\right) \\
&= M\left(r + \gamma PV + P\rho^\star + (1-\gamma)\left\|V - \frac{1}{1-\gamma}\rho^\star\right\|_\infty \mathbf{1}\right) \\
&= M\left(r + \gamma PV + P\rho^\star\right) + (1-\gamma)\left\|V - \frac{1}{1-\gamma}\rho^\star\right\|_\infty \mathbf{1} \\
&\leq M\left(r + \gamma PV\right) + M\left(P\rho^\star\right) + (1-\gamma)\left\|V - \frac{1}{1-\gamma}\rho^\star\right\|_\infty \mathbf{1} \\
&= \mathcal{T}_\gamma(V) + \rho^\star + (1-\gamma)\left\|V - \frac{1}{1-\gamma}\rho^\star\right\|_\infty \mathbf{1}.
\end{aligned}$$

Subtracting $V + \rho^\star$ from both inequalities and combining, we obtain

$$\mathcal{T}_\gamma(V) - V - \varepsilon\mathbf{1} + P_\pi\rho^\star - \rho^\star - (1-\gamma)\left\|V - \frac{1}{1-\gamma}\rho^\star\right\|_\infty \mathbf{1}$$
$$\leq \mathcal{T}(V) - V - \rho^\star$$
$$\leq \mathcal{T}_\gamma(V) - V + (1-\gamma)\left\|V - \frac{1}{1-\gamma}\rho^\star\right\|_\infty \mathbf{1}.$$

Therefore

$$\|\mathcal{T}(V) - V - \rho^\star\|_\infty \leq \|P_\pi\rho^\star - \rho^\star\|_\infty + \|\mathcal{T}_\gamma(V) - V\|_\infty + (1-\gamma)\left\|V - \frac{1}{1-\gamma}\rho^\star\right\|_\infty + \varepsilon.$$

$\square$

**Lemma J.4.** *Suppose for some $V \in \mathbb{R}^\mathcal{S}$ that some policy $\pi$ is $\varepsilon$-greedy with respect to $r + \gamma PV$, that is, $\mathcal{T}_\gamma^\pi(V) \geq \mathcal{T}_\gamma(V) - \varepsilon\mathbf{1}$. Then*

$$\rho^\star - P_\pi\rho^\star \leq (1-\gamma)\frac{1 + \varepsilon + \|\mathcal{T}_\gamma(V) - V\|_\infty}{\gamma}\mathbf{1} + 2(1-\gamma)\left\|V - \frac{1}{1-\gamma}\rho^\star\right\|_\infty \mathbf{1}.$$

*Proof.* First we calculate that

$$P_\pi V = \frac{1}{\gamma}V + \left(P_\pi V - \frac{1}{\gamma}V\right) = \frac{1}{\gamma}V + \frac{1}{\gamma}(\gamma P_\pi V - V)$$
$$= \frac{V - r_\pi}{\gamma} + \frac{r_\pi + \gamma P_\pi V - V}{\gamma} = \frac{V - r_\pi}{\gamma} + \frac{\mathcal{T}_\gamma^\pi(V) - V}{\gamma}$$
$$\geq \frac{V - r_\pi}{\gamma} + \frac{\mathcal{T}_\gamma(V) - V}{\gamma} - \frac{\varepsilon}{\gamma}$$

which implies

$$V - P_\pi V \leq V - \frac{V}{\gamma} + \frac{\|r_\pi\|_\infty + \varepsilon}{\gamma}\mathbf{1} + \frac{\|\mathcal{T}_\gamma(V) - V\|_\infty}{\gamma} \leq \frac{1 + \varepsilon + \|\mathcal{T}_\gamma(V) - V\|_\infty}{\gamma}\mathbf{1}.$$

Using this key bound, we have that

$$\rho^\star - P_\pi\rho^\star \leq (1-\gamma)V + (1-\gamma)\left\|\frac{1}{1-\gamma}\rho^\star - V\right\|_\infty \mathbf{1} - P_\pi\rho^\star$$
$$\leq (1-\gamma)V + (1-\gamma)\left\|\frac{1}{1-\gamma}\rho^\star - V\right\|_\infty \mathbf{1} - (1-\gamma)P_\pi V + P_\pi(1-\gamma)\left\|\frac{1}{1-\gamma}\rho^\star - V\right\|_\infty \mathbf{1}$$
$$= (1-\gamma)(V - P_\pi V) + 2(1-\gamma)\left\|\frac{1}{1-\gamma}\rho^\star - V\right\|_\infty \mathbf{1}$$
$$\leq (1-\gamma)\frac{1 + \varepsilon + \|\mathcal{T}_\gamma(V) - V\|_\infty}{\gamma}\mathbf{1} + 2(1-\gamma)\left\|\frac{1}{1-\gamma}\rho^\star - V\right\|_\infty \mathbf{1}.$$

$\square$

**Lemma J.5.** *Suppose for some $V \in \mathbb{R}^\mathcal{S}$ that some policy $\pi$ is $\varepsilon$-greedy with respect to $r + \gamma PV$, that is, $\mathcal{T}_\gamma^\pi(V) \geq \mathcal{T}_\gamma(V) - \varepsilon\mathbf{1}$. Then*

$$\|\mathcal{T}^\pi(V) - V - \rho^\star\|_\infty \leq \|\mathcal{T}(V) - V - \rho^\star\|_\infty + \|P_\pi\rho^\star - \rho^\star\|_\infty + 2(1-\gamma)\left\|V - \frac{1}{1-\gamma}\rho^\star\right\|_\infty + \varepsilon.$$

*Proof.* Note that since $\mathcal{T}^\pi(V) \leq \mathcal{T}(V)$, we immediately have that

$$\mathcal{T}^\pi(V) - V - \rho^\star \leq \mathcal{T}(V) - V - \rho^\star \leq \|\mathcal{T}(V) - V - \rho^\star\|_\infty \mathbf{1}.$$

Thus it remains to lower-bound $\mathcal{T}^\pi(V) - V - \rho^\star$. Let $\widetilde{\pi}$ be some policy such that $M(r + PV) = M^{\widetilde{\pi}}(r + PV)$. Now we calculate that

$$
\begin{aligned}
\mathcal{T}^\pi(V) = M^\pi(r + PV) &= M^\pi(r + \gamma PV) + (1 - \gamma)P_\pi V \\
&\geq M(r + \gamma PV) - \varepsilon\mathbf{1} + (1 - \gamma)P_\pi V \\
&\geq M^{\widetilde{\pi}}(r + \gamma PV) - \varepsilon\mathbf{1} + (1 - \gamma)P_\pi V \\
&= M^{\widetilde{\pi}}(r + PV) - (1 - \gamma)P_{\widetilde{\pi}}V - \varepsilon\mathbf{1} + (1 - \gamma)P_\pi V \\
&= M(r + PV) - \varepsilon\mathbf{1} - (1 - \gamma)\left(P_{\widetilde{\pi}} - P_\pi\right)V \\
&= \mathcal{T}(V) - \varepsilon\mathbf{1} - (P_{\widetilde{\pi}} - P_\pi)\rho^\star - (1 - \gamma)(P_{\widetilde{\pi}} - P_\pi)\left(V - \frac{1}{1 - \gamma}\rho^\star\right) \\
&\geq \mathcal{T}(V) - \varepsilon\mathbf{1} - (P_{\widetilde{\pi}} - P_\pi)\rho^\star - 2(1 - \gamma)\left\|V - \frac{1}{1 - \gamma}\rho^\star\right\|_\infty \mathbf{1} \\
&\geq \mathcal{T}(V) - \varepsilon\mathbf{1} - (\rho^\star - P_\pi\rho^\star) - 2(1 - \gamma)\left\|V - \frac{1}{1 - \gamma}\rho^\star\right\|_\infty \mathbf{1}
\end{aligned}
$$

where in the final inequality we used that $P_{\widetilde{\pi}}\rho^\star \leq \rho^\star$. Subtracting $V + \rho^\star$ from both inequalities we obtain that

$$
\mathcal{T}^\pi(V) - V - \rho^\star \geq \mathcal{T}(V) - V - \rho^\star - \varepsilon\mathbf{1} - (\rho^\star - P_\pi\rho^\star) - 2(1 - \gamma)\left\|V - \frac{1}{1 - \gamma}\rho^\star\right\|_\infty \mathbf{1}.
$$

Combining this with the upper bound, taking $\|\cdot\|_\infty$, and using the triangle inequality, we obtain the desired statement. $\qquad\square$

**Lemma J.6.** *Suppose for some $V \in \mathbb{R}^\mathcal{S}$ that some policy $\pi$ is $\varepsilon$-greedy with respect to $r + \gamma PV$, that is, $\mathcal{T}_\gamma^\pi(V) \geq \mathcal{T}_\gamma(V) - \varepsilon\mathbf{1}$. Then*

$$
\|\rho^\pi - \rho^\star\|_\infty \leq \left(\mathsf{T}_{\mathrm{drop}}^\pi + 1\right)\left(7(1 - \gamma)\left\|V - \frac{1}{1 - \gamma}\rho^\star\right\|_\infty + \frac{2 - \gamma}{\gamma}\|\mathcal{T}_\gamma(V) - V\|_\infty + 2\frac{1 - \gamma}{\gamma} + \frac{2}{\gamma}\varepsilon\right).
$$

*Furthermore, letting $h$ satisfy the modified Bellman equations* (7), *we have*

$$
\begin{aligned}
\|\rho^\pi - \rho^\star\|_\infty \leq \left(\mathsf{T}_{\mathrm{drop}}^\pi + 1\right)&\left(7(1 - \gamma)\min\left\{\|h\|_{\mathrm{sp}}, \left\|h^{\pi^\star}\right\|_{\mathrm{sp}} + \mathsf{T}_{\mathrm{drop}} + \left\|h^{\pi^\star}\right\|_{\mathrm{sp}}\mathsf{T}_{\mathrm{drop}}\right\} + \frac{2 + 6\gamma}{\gamma}\|\mathcal{T}_\gamma(V) - V\|_\infty\right. \\
&\left. + 2\frac{1 - \gamma}{\gamma} + \frac{2}{\gamma}\varepsilon\right).
\end{aligned}
$$

*Additionally assuming $\gamma \geq \frac{1}{2}$, we have*

$$
\begin{aligned}
\|\rho^\pi - \rho^\star\|_\infty \leq \left(\mathsf{T}_{\mathrm{drop}}^\pi + 1\right)&\left((1 - \gamma)\left(4 + 7\min\left\{\|h\|_{\mathrm{sp}}, \left\|h^{\pi^\star}\right\|_{\mathrm{sp}} + \mathsf{T}_{\mathrm{drop}} + \left\|h^{\pi^\star}\right\|_{\mathrm{sp}}\mathsf{T}_{\mathrm{drop}}\right\}\right)\right. \\
&\left. + 16\|\mathcal{T}_\gamma(V) - V\|_\infty + 4\varepsilon\right).
\end{aligned}
$$

*Proof.* By Corollary 2.2 we have

$$
\|\rho^\pi - \rho^\star\|_\infty \leq \mathsf{T}_{\mathrm{drop}}^\pi\|P_\pi\rho^\star - \rho^\star\|_\infty + \|\mathcal{T}^\pi(V) - V - \rho^\star\|_\infty \tag{25}
$$

so it remains to control the terms $\|P_\pi\rho^\star - \rho^\star\|_\infty$ and $\|\mathcal{T}^\pi(V) - V - \rho^\star\|_\infty$. By Lemma J.4 we have

$$
\|\rho^\star - P_\pi\rho^\star\|_\infty \leq (1 - \gamma)\frac{1 + \varepsilon + \|\mathcal{T}_\gamma(V) - V\|_\infty}{\gamma} + 2(1 - \gamma)\left\|V - \frac{1}{1 - \gamma}\rho^\star\right\|_\infty. \tag{26}
$$

Starting with Lemma J.5, we have

$$\|\mathcal{T}^{\pi}(V) - V - \rho^{\star}\|_{\infty} \leq \|\mathcal{T}(V) - V - \rho^{\star}\|_{\infty} + \|P_{\pi}\rho^{\star} - \rho^{\star}\|_{\infty} + 2(1-\gamma)\left\|V - \frac{1}{1-\gamma}\rho^{\star}\right\|_{\infty} + \varepsilon$$

$$\overset{(i)}{\leq} \left(\|P_{\pi}\rho^{\star} - \rho^{\star}\|_{\infty} + \|\mathcal{T}_{\gamma}(V) - V\|_{\infty} + (1-\gamma)\left\|V - \frac{1}{1-\gamma}\rho^{\star}\right\|_{\infty} + \varepsilon\right)$$

$$+ \|P_{\pi}\rho^{\star} - \rho^{\star}\|_{\infty} + 2(1-\gamma)\left\|V - \frac{1}{1-\gamma}\rho^{\star}\right\|_{\infty} + \varepsilon$$

$$= 2\|P_{\pi}\rho^{\star} - \rho^{\star}\|_{\infty} + 3(1-\gamma)\left\|V - \frac{1}{1-\gamma}\rho^{\star}\right\|_{\infty} + \|\mathcal{T}_{\gamma}(V) - V\|_{\infty} + 2\varepsilon$$

$$\overset{(ii)}{\leq} 2\left((1-\gamma)\frac{1+\varepsilon+\|\mathcal{T}_{\gamma}(V) - V\|_{\infty}}{\gamma} + 2(1-\gamma)\left\|V - \frac{1}{1-\gamma}\rho^{\star}\right\|_{\infty}\right)$$

$$+ 3(1-\gamma)\left\|V - \frac{1}{1-\gamma}\rho^{\star}\right\|_{\infty} + \|\mathcal{T}_{\gamma}(V) - V\|_{\infty} + 2\varepsilon$$

$$= 7(1-\gamma)\left\|V - \frac{1}{1-\gamma}\rho^{\star}\right\|_{\infty} + \left(2\frac{1-\gamma}{\gamma} + 1\right)\|\mathcal{T}_{\gamma}(V) - V\|_{\infty}$$

$$+ 2\frac{1-\gamma}{\gamma} + 2\left(\frac{1-\gamma}{\gamma} + 1\right)\varepsilon$$

$$= 7(1-\gamma)\left\|V - \frac{1}{1-\gamma}\rho^{\star}\right\|_{\infty} + \frac{2-\gamma}{\gamma}\|\mathcal{T}_{\gamma}(V) - V\|_{\infty} + 2\frac{1-\gamma}{\gamma} + \frac{2}{\gamma}\varepsilon$$

$$(27)$$

where we used Lemma J.3 in $(i)$ to bound $\|\mathcal{T}(V) - V - \rho^{\star}\|_{\infty}$, and then we used (26) in $(ii)$ to bound $\|P_{\pi}\rho^{\star} - \rho^{\star}\|_{\infty}$.

Now using this bound (27) on $\|\mathcal{T}^{\pi}(V) - V - \rho^{\star}\|_{\infty}$ and the bound (26) on $\|P_{\pi}\rho^{\star} - \rho^{\star}\|_{\infty}$ in (25), we have that

$$\|\rho^{\pi} - \rho^{\star}\|_{\infty} \leq \mathsf{T}_{\text{drop}}^{\pi}\left((1-\gamma)\frac{1+\varepsilon+\|\mathcal{T}_{\gamma}(V) - V\|_{\infty}}{\gamma} + 2(1-\gamma)\left\|V - \frac{1}{1-\gamma}\rho^{\star}\right\|_{\infty}\right)$$

$$+ \left(7(1-\gamma)\left\|V - \frac{1}{1-\gamma}\rho^{\star}\right\|_{\infty} + \frac{2-\gamma}{\gamma}\|\mathcal{T}_{\gamma}(V) - V\|_{\infty} + 2\frac{1-\gamma}{\gamma} + \frac{2}{\gamma}\varepsilon\right)$$

$$\leq \left(\mathsf{T}_{\text{drop}}^{\pi} + 1\right)\left(7(1-\gamma)\left\|V - \frac{1}{1-\gamma}\rho^{\star}\right\|_{\infty} + \frac{2-\gamma}{\gamma}\|\mathcal{T}_{\gamma}(V) - V\|_{\infty} + 2\frac{1-\gamma}{\gamma} + \frac{2}{\gamma}\varepsilon\right).$$

$$(28)$$

This is the first statement of the lemma.

For the second statement of the lemma, we have by the triangle inequality and Lemmas J.1 and J.2 that

$$\left\|V - \frac{1}{1-\gamma}\rho^{\star}\right\|_{\infty} \leq \|V - V_{\gamma}^{\star}\|_{\infty} + \left\|V_{\gamma}^{\star} - \frac{1}{1-\gamma}\rho^{\star}\right\|_{\infty}$$

$$\leq \|V - V_{\gamma}^{\star}\|_{\infty} + \min\left\{\|h\|_{\text{sp}}, \left\|h^{\pi^{\star}}\right\|_{\text{sp}} + \mathsf{T}_{\text{drop}} + \left\|h^{\pi^{\star}}\right\|_{\text{sp}}\mathsf{T}_{\text{drop}}\right\}$$

$$\leq \frac{1}{1-\gamma}\|\mathcal{T}_{\gamma}(V) - V\|_{\infty} + \min\left\{\|h\|_{\text{sp}}, \left\|h^{\pi^{\star}}\right\|_{\text{sp}} + \mathsf{T}_{\text{drop}} + \left\|h^{\pi^{\star}}\right\|_{\text{sp}}\mathsf{T}_{\text{drop}}\right\}$$

$$(29)$$

where for the final inequality we used the standard fact that $\left\|V - V_{\gamma}^{\star}\right\|_{\infty} \leq \frac{1}{1-\gamma}\|\mathcal{T}_{\gamma}(V) - V\|_{\infty}$ (which follows from

$$\left\|V - V_{\gamma}^{\star}\right\|_{\infty} \leq \|V - \mathcal{T}_{\gamma}(V)\|_{\infty} + \left\|\mathcal{T}_{\gamma}(V) - \mathcal{T}_{\gamma}(V_{\gamma}^{\star})\right\|_{\infty} \leq \|V - \mathcal{T}_{\gamma}(V)\|_{\infty} + \gamma\left\|V - V_{\gamma}^{\star}\right\|_{\infty}$$

and then rearranging). Combining (29) with (28), we have that

$$\|\rho^\pi - \rho^\star\|_\infty \le \left(\mathsf{T}_{\text{drop}}^\pi + 1\right)\left(7(1-\gamma)\left\|V - \frac{1}{1-\gamma}\rho^\star\right\|_\infty + \frac{2-\gamma}{\gamma}\|\mathcal{T}_\gamma(V) - V\|_\infty + 2\frac{1-\gamma}{\gamma} + \frac{2}{\gamma}\varepsilon\right)$$

$$\le \left(\mathsf{T}_{\text{drop}}^\pi + 1\right)\left(7(1-\gamma)\left(\frac{1}{1-\gamma}\|\mathcal{T}_\gamma(V) - V\|_\infty + \min\left\{\|h\|_{\text{sp}}, \left\|h^{\pi^\star}\right\|_{\text{sp}} + \mathsf{T}_{\text{drop}} + \left\|h^{\pi^\star}\right\|_{\text{sp}}\mathsf{T}_{\text{drop}}\right\}\right)\right.$$

$$\left. + \frac{2-\gamma}{\gamma}\|\mathcal{T}_\gamma(V) - V\|_\infty + 2\frac{1-\gamma}{\gamma} + \frac{2}{\gamma}\varepsilon\right)$$

$$= \left(\mathsf{T}_{\text{drop}}^\pi + 1\right)\left(7(1-\gamma)\min\left\{\|h\|_{\text{sp}}, \left\|h^{\pi^\star}\right\|_{\text{sp}} + \mathsf{T}_{\text{drop}} + \left\|h^{\pi^\star}\right\|_{\text{sp}}\mathsf{T}_{\text{drop}}\right\} + \frac{2+6\gamma}{\gamma}\|\mathcal{T}_\gamma(V) - V\|_\infty\right.$$

$$\left. + 2\frac{1-\gamma}{\gamma} + \frac{2}{\gamma}\varepsilon\right)$$

as desired.

The final statement follows immediately from the facts that $\gamma \le 1$ and $\frac{1}{\gamma} \le 2$. $\qquad\square$

## J.3   Faster value error convergence

In this subsection we present results which relate the output of undiscounted Picard iterations to the discounted value function $V_\gamma^\star$, with the objective of proving Lemma 4.3.

The following lemma basically follows from [Puterman, 1994, Theorem 9.4.1], although as discussed in Appendix D, there are some ambiguities in its requirements on $h$. For this reason we provide a complete proof.

**Lemma J.7.** *Suppose that $(\rho^\star, h)$ satisfies both the modified and unmodified Bellman equations. Then for any $n \ge 0$, we have*

$$\left\|\mathcal{T}^{(n)}(\mathbf{0}) - n\rho^\star\right\|_\infty \le \|h\|_{\text{sp}}.$$

*Proof.* First we note that elementwise

$$h - (\max_{s\in\mathcal{S}} h(s))\mathbf{1} \le \mathbf{0} \le h - (\min_{s\in\mathcal{S}} h(s))\mathbf{1}.$$

By monotonicity of $\mathcal{T}$ (applied $n$ times for each inequality), we have that

$$\mathcal{T}^{(n)}\left(h - (\max_{s\in\mathcal{S}} h(s))\mathbf{1}\right) \le \mathcal{T}^{(n)}(\mathbf{0}) \le \mathcal{T}^{(n)}\left(h - (\min_{s\in\mathcal{S}} h(s))\mathbf{1}\right). \tag{30}$$

Using the constant shift property of $\mathcal{T}$ (that $\mathcal{T}(x + c\mathbf{1}) = \mathcal{T}(x) + c\mathbf{1}$ for any $x \in \mathbb{R}^\mathcal{S}$ and any $c \in \mathbb{R}$), as well as Lemma D.2 which guarantees that $\mathcal{T}(h + c\rho^\star) = h + (c+1)\rho^\star$, we have that

$$\mathcal{T}^{(n)}(h + c\mathbf{1}) = \mathcal{T}^{(n-1)}(\mathcal{T}(h + c\mathbf{1})) = \mathcal{T}^{(n-1)}(h + \rho^\star + c\mathbf{1})$$

$$= \mathcal{T}^{(n-2)}(\mathcal{T}(h + \rho^\star + c\mathbf{1})) = \mathcal{T}^{(n-2)}(2\rho^\star + c\mathbf{1})$$

$$\vdots$$

$$= \mathcal{T}^{(0)}(n\rho^\star + c\mathbf{1}) = n\rho^\star + c\mathbf{1}.$$

Therefore using this calculation to evaluate the left- and right-hand sides of (30), we have that

$$n\rho^\star + h - (\max_{s\in\mathcal{S}} h(s))\mathbf{1} \le \mathcal{T}^{(n)}(\mathbf{0}) \le n\rho^\star + h - (\min_{s\in\mathcal{S}} h(s))\mathbf{1}.$$

Rearranging we have

$$h - (\max_{s\in\mathcal{S}} h(s))\mathbf{1} \le \mathcal{T}^{(n)}(\mathbf{0}) - n\rho^\star \le h - (\min_{s\in\mathcal{S}} h(s))\mathbf{1}$$

which implies

$$\left\|\mathcal{T}^{(n)}(\mathbf{0}) - n\rho^\star\right\|_\infty \le \max\left\{\max_{s'\in\mathcal{S}} h(s') - (\min_{s\in\mathcal{S}} h(s)), -\left(\min_{s'\in\mathcal{S}} h(s') - (\max_{s\in\mathcal{S}} h(s))\right)\right\} = \|h\|_{\text{sp}}.$$

$$\square$$

**Lemma J.8.** *For any $n \geq 0$, we have*

$$\left\| \mathcal{T}^{(n)}(\mathbf{0}) - n\rho^\star \right\|_\infty \leq \left\| h^{\pi^\star} \right\|_{\mathrm{sp}} + \mathsf{T}_{\mathrm{drop}} + \left\| h^{\pi^\star} \right\|_{\mathrm{sp}} \mathsf{T}_{\mathrm{drop}}.$$

*Proof.* Let $\pi_k$ be some policy attaining the maximum during the $k$th application of $\mathcal{T}$, that is

$$\mathcal{T}^{(k)}(\mathbf{0}) = \mathcal{T}^{\pi_k}\left( \mathcal{T}^{(k-1)}(\mathbf{0}) \right).$$

Then it is straightforward to compute that

$$\mathcal{T}^{(n)}(\mathbf{0}) = r_{\pi_n} + P_{\pi_n} r_{\pi_{n-1}} + P_{\pi_n} P_{\pi_{n-1}} r_{\pi_{n-2}} + \cdots + \left( P_{\pi_n} P_{\pi_{n-1}} \cdots P_{\pi_2} \right) r_{\pi_1}$$

$$= \sum_{t=1}^{n} \left( \prod_{k=n}^{t+1} P_{\pi_k} \right) r_{\pi_t} \tag{31}$$

where we take the product $\prod_{k=n}^{t+1} P_{\pi_k}$ to be equal to the identity matrix if it is empty (if $n < t+1$, which happens only for $t = n$).

Using steps identical to those used to obtain (23) but replacing $\pi_\gamma^\star$ by $\pi_t$, we obtain for any $t$ that

$$r_{\pi_t} \leq \rho^\star + (I - P_{\pi_t})h^{\pi^\star} + \left( 1 + \left\| h^{\pi^\star} \right\|_{\mathrm{sp}} \right) e_{\mathcal{S}_t}$$

where $e_{\mathcal{S}_t}$ is the indicator function for the set of states $\mathcal{S}_t = \{s : e_s^\top P_{\pi_t} \rho^\star < \rho^\star(s)\}$. Recalling the reward function $\bar{r}(s, a) = \mathbb{I}\{P_{sa}\rho^\star < \rho^\star(s)\}$ defined in Section 2, this is equivalent to $e_{\mathcal{S}_t} = M^{\pi_t}\bar{r} = \bar{r}_{\pi_t}$. Combining with (31) (and using monotonicity of $P_\pi$ for any $\pi$) we have that

$$\mathcal{T}^{(n)}(\mathbf{0}) \leq \sum_{t=1}^{n} \left( \prod_{k=n}^{t+1} P_{\pi_k} \right) \left( \rho^\star + (I - P_{\pi_t})h^{\pi^\star} + \left( 1 + \left\| h^{\pi^\star} \right\|_{\mathrm{sp}} \right) \bar{r}_{\pi_t} \right) \tag{32}$$

and now we upper bound each term in (32). For any policy $\pi$ we have $P_\pi \rho^\star = M^\pi P \rho^\star \leq MP\rho^\star = \rho^\star$, and so applying this fact many times we have that

$$\sum_{t=1}^{n} \left( \prod_{k=n}^{t+1} P_{\pi_k} \right) \rho^\star \leq \sum_{t=1}^{n} \rho^\star = n\rho^\star.$$

Next, we have that

$$\sum_{t=1}^{n} \left( \prod_{k=n}^{t+1} P_{\pi_k} \right) (I - P_{\pi_t})h^{\pi^\star} = \sum_{t=1}^{n} \left( \prod_{k=n}^{t+1} P_{\pi_k} \right) h^{\pi^\star} - \sum_{t=1}^{n} \left( \prod_{k=n}^{t} P_{\pi_k} \right) h^{\pi^\star}$$

$$= h^{\pi^\star} - \left( \prod_{k=n}^{1} P_{\pi_k} \right) h^{\pi^\star}.$$

Since $\left( \prod_{k=n}^{1} P_{\pi_k} \right) = P_{\pi_n} \cdots P_{\pi_1}$ is a stochastic matrix, we have that $h^{\pi^\star} - \left( \prod_{k=n}^{1} P_{\pi_k} \right) h^{\pi^\star} \leq \left\| h^{\pi^\star} \right\|_{\mathrm{sp}} \mathbf{1}$. Finally, we have

$$\sum_{t=1}^{n} \left( \prod_{k=n}^{t+1} P_{\pi_k} \right) \bar{r}_{\pi_t} \leq \mathsf{T}_{\mathrm{drop}} \mathbf{1}$$

using Lemma F.7, since each coordinate of the left-hand side is bounded by the expected total reward value function of the nonstationary policy $(\pi_1, \pi_2, \dots)$ (with terms after time $n$ dropped). Combining these three bounds with (32), we obtain that

$$\mathcal{T}^{(n)}(\mathbf{0}) \leq n\rho^\star + \left\| h^{\pi^\star} \right\|_{\mathrm{sp}} \mathbf{1} + \left( 1 + \left\| h^{\pi^\star} \right\|_{\mathrm{sp}} \right) \mathsf{T}_{\mathrm{drop}} \mathbf{1}.$$

Finally it remains to lower-bound $\mathcal{T}^{(n)}(\mathbf{0})$. Since we have $\mathbf{0} \geq h^{\pi^\star} - \max_{s \in \mathcal{S}} h^{\pi^\star}(s)\mathbf{1} =: x$, by monotonicity of $\mathcal{T}$ we have that

$$\mathcal{T}^{(n)}(\mathbf{0}) \geq \mathcal{T}^{(n)}(x).$$

Furthermore since $\mathcal{T}(h) \geq \mathcal{T}^{\pi^\star}(h)$ for any $h$ we have that

$$\mathcal{T}^{(n)}(x) \geq \left(\mathcal{T}^{\pi^\star}\right)^{(n)}(x).$$

Furthermore we have

$$\mathcal{T}^{\pi^\star}(x) = r_{\pi^\star} + P_{\pi^\star}(h^{\pi^\star} - \max_{s \in \mathcal{S}} h^{\pi^\star}(s)\mathbf{1}) = r_{\pi^\star} + P_{\pi^\star}h^{\pi^\star} - \max_{s \in \mathcal{S}} h^{\pi^\star}(s)\mathbf{1}$$

$$= \rho^\star + h^{\pi^\star} - \max_{s \in \mathcal{S}} h^{\pi^\star}(s)\mathbf{1} = \rho^\star + x.$$

Repeating this fact $n$ times we have

$$\left(\mathcal{T}^{\pi^\star}\right)^{(n)}(x) = n\rho^\star + x \geq n\rho^\star - \left\|h^{\pi^\star}\right\|_{\mathrm{sp}}\mathbf{1}.$$

Therefore

$$\mathcal{T}^{(n)}(\mathbf{0}) \geq n\rho^\star - \left\|h^{\pi^\star}\right\|_{\mathrm{sp}}\mathbf{1},$$

and so combining with the upper bound on $\mathcal{T}^{(n)}(\mathbf{0})$ we conclude that

$$\left\|\mathcal{T}^{(n)}(\mathbf{0}) - n\rho^\star\right\|_\infty \leq \left\|h^{\pi^\star}\right\|_{\mathrm{sp}} + \left(1 + \left\|h^{\pi^\star}\right\|_{\mathrm{sp}}\right)\mathsf{T}_{\mathrm{drop}}.$$

$\square$

Now we can combine both of these lemmas along with Lemmas J.2 and J.1 to prove Lemma 4.3.

*Proof of Lemma 4.3.* By triangle inequality we have

$$\left\|\frac{1}{t(1-\gamma)}\mathcal{T}^{(t)}(\mathbf{0}) - V_\gamma^\star\right\|_\infty \leq \left\|\frac{1}{t(1-\gamma)}\mathcal{T}^{(t)}(\mathbf{0}) - \frac{1}{1-\gamma}\rho^\star\right\|_\infty + \left\|\frac{1}{1-\gamma}\rho^\star - V_\gamma^\star\right\|_\infty$$

$$= \frac{1}{t(1-\gamma)}\left\|\mathcal{T}^{(t)}(\mathbf{0}) - t\rho^\star\right\|_\infty + \left\|\frac{1}{1-\gamma}\rho^\star - V_\gamma^\star\right\|_\infty. \qquad (33)$$

Using Lemma J.7 to bound the first term and Lemma J.2 on the second term, we obtain that

$$\left\|\frac{1}{t(1-\gamma)}\mathcal{T}^{(t)}(\mathbf{0}) - V_\gamma^\star\right\|_\infty \leq \frac{1}{t(1-\gamma)}\|h\|_{\mathrm{sp}} + \|h\|_{\mathrm{sp}}$$

$$\leq \frac{2}{t(1-\gamma)}\|h\|_{\mathrm{sp}}$$

using the fact that $t \leq \frac{1}{1-\gamma}$ (so $\frac{1}{t(1-\gamma)} \geq 1$) in the second inequality.

Using identical steps but instead bounding the first and second terms of (33) with Lemmas J.8 and J.1, respectively, we obtain that

$$\left\|\frac{1}{t(1-\gamma)}\mathcal{T}^{(t)}(\mathbf{0}) - V_\gamma^\star\right\|_\infty \leq \frac{2}{t(1-\gamma)}\left(\left\|h^{\pi^\star}\right\|_{\mathrm{sp}} + \mathsf{T}_{\mathrm{drop}} + \left\|h^{\pi^\star}\right\|_{\mathrm{sp}}\mathsf{T}_{\mathrm{drop}}\right).$$

We conclude by taking the minimum of these two bounds on $\left\|\frac{1}{t(1-\gamma)}\mathcal{T}^{(t)}(\mathbf{0}) - V_\gamma^\star\right\|_\infty$. $\square$

## J.4 Discounted VI results

*Proof of Theorem 4.4.* Let $M = \min\left\{\|h\|_{\mathrm{sp}}, \|h^{\pi^\star}\|_{\mathrm{sp}} + \mathsf{T}_{\mathrm{drop}} + \|h^{\pi^\star}\|_{\mathrm{sp}}\mathsf{T}_{\mathrm{drop}}\right\}$. Since $E' = \left\lfloor\frac{1}{1-\gamma}\right\rfloor \geq \frac{1}{1-\gamma}$, we can apply Lemma 4.3 to obtain that

$$\left\|x_E - V_\gamma^\star\right\|_\infty = \left\|\mathcal{T}^{(E)}(\mathbf{0}) - V_\gamma^\star\right\|_\infty$$

$$\leq \frac{2}{E(1-\gamma)}M$$

$$\leq \frac{2}{\left(\frac{1}{1-\gamma} - 1\right)(1-\gamma)}M$$

$$= \frac{2}{\gamma}M.$$

We can immediately combine this with Theorem 4.2 to obtain that

$$\|\mathcal{T}_\gamma(V) - V\|_\infty \le \frac{16e}{\gamma} \frac{\gamma^{n-E'}}{\sum_{i=0}^{n-E'} \gamma^i} M.$$

The second statement of the theorem would follow if we could show the bound

$$\frac{\gamma^{n-E'}}{\sum_{i=0}^{n-E'} \gamma^i} \le \left(1 + \frac{e}{\gamma}\right) \frac{\gamma^n}{\sum_{i=0}^{n} \gamma^i}$$

under the condition that $n \ge 2E' - 1$. By rearranging, this is equivalent to showing that

$$\sum_{i=0}^{n} \gamma^i \le (1+e)\gamma^{E'} \sum_{i=0}^{n-E'} \gamma^i = \left(1 + \frac{e}{\gamma}\right) \sum_{i=E'}^{n} \gamma^i. \tag{34}$$

Writing $\sum_{i=0}^{n} \gamma^i = \sum_{i=0}^{E'-1} \gamma^i + \sum_{i=E'}^{n} \gamma^i$, (34) would follow from showing that

$$\sum_{i=0}^{E'-1} \gamma^i \le e \sum_{i=E'}^{n} \gamma^i. \tag{35}$$

Under the assumption that $n \ge 2E' - 1$, we have that

$$\sum_{i=E'}^{n} \gamma^i = \gamma^{E'} \sum_{i=0}^{n-E'} \gamma^i \ge \gamma^{E'} \sum_{i=0}^{E'-1} \gamma^i$$

(since $n - E' \ge E' - 1$). By rearranging and using the bound $\gamma^{-\frac{1}{1-\gamma}+1} \le e$ from (20), we have that $\gamma^{-E'} \le \gamma^{-\frac{1}{1-\gamma}} \le \frac{e}{\gamma}$, and so

$$\sum_{i=0}^{E'-1} \gamma^i \le \gamma^{-E'} \sum_{i=E'}^{n} \gamma^i \le \frac{e}{\gamma} \sum_{i=E'}^{n} \gamma^i$$

showing (35) as desired. $\qquad\square$

*Proof of Theorem 4.5.* Let $M = \min\left\{\|h\|_{\mathrm{sp}}, \|h^{\pi^\star}\|_{\mathrm{sp}} + \mathsf{T}_{\mathrm{drop}} + \|h^{\pi^\star}\|_{\mathrm{sp}}\mathsf{T}_{\mathrm{drop}}\right\}$. By Lemma 4.3 we have

$$\left\|\mathcal{T}^{(n)}(\mathbf{0}) - V_\gamma^\star\right\|_\infty \le \frac{2M}{n(1-\gamma)} = \frac{2M}{n\frac{1}{n}} = 2M.$$

By applying (22) from the proof of Theorem 4.2 with $\mathcal{L} = \mathcal{T}_\gamma$, $x_0 = \mathcal{T}^{(n)}(\mathbf{0})$, $x^\star = V_\gamma^\star$, and $t = n$, since $\lfloor \frac{1}{1-\gamma} \rfloor - 1 = n - 1$ we obtain that it outputs $V$ such that

$$\|\mathcal{T}_\gamma(V) - V\|_\infty \le 4(1-\gamma)\gamma^{n-(n-1)} \left\|\mathcal{T}^{(n)}(\mathbf{0}) - V_\gamma^\star\right\|_\infty$$

$$\le \frac{4}{n+1} \left\|\mathcal{T}^{(n)}(\mathbf{0}) - V_\gamma^\star\right\|_\infty$$

$$\le \frac{4}{n+1} \cdot 2M$$

where for the second inequality we used that $(1-\gamma)\gamma = \frac{1-\frac{1}{n}}{n} \le \frac{1}{n+1}$, since $\frac{1-\frac{1}{n}}{n} = \frac{1}{n} \frac{(n+1)(1-\frac{1}{n})}{n+1} = \frac{1}{n} \frac{n+1-1-\frac{1}{n}}{n+1}$.

Now combining this bound with Lemma J.6, we have that the policy $\widehat{\pi}$ which is greedy with respect to $r + \gamma PV$ satisfies

$$
\left\| \rho^{\widehat{\pi}} - \rho^\star \right\|_\infty \leq \left( \mathsf{T}^{\widehat{\pi}}_{\mathrm{drop}} + 1 \right) \left( 7(1-\gamma)M + \frac{2+6\gamma}{\gamma} \left\| \mathcal{T}_\gamma(V) - V \right\|_\infty + 2\frac{1-\gamma}{\gamma} \right)
$$

$$
\leq (\mathsf{T}_{\mathrm{drop}} + 1) \left( \frac{7}{n}M + \frac{2+6}{1-\frac{1}{n}}\frac{8M}{n+1} + 2\frac{\frac{1}{n}}{1-\frac{1}{n}} \right)
$$

$$
= (\mathsf{T}_{\mathrm{drop}} + 1) \left( \frac{7}{n}M + \frac{64M}{n-\frac{1}{n}} + \frac{2}{n-1} \right)
$$

$$
\leq (\mathsf{T}_{\mathrm{drop}} + 1) \left( \frac{71M+2}{n-1} \right).
$$

$\square$

Now we demonstrate the effects of using more iterations from the "third phase" of Algorithm 3. Note that the below theorem involves $(2+k)n$ iterations, so to compare its convergence rate to that of Theorem 4.5, it should be multiplied by $(2+k)$, and the guarantee of Theorem 4.5 should be multiplied by 2 (since it uses $2n$ iterations).

**Theorem J.9.** *Fix an integer $n \geq 2$ and a number $k \geq 0$ such that $kn$ is an integer. Set $\gamma = 1 - \frac{1}{n}$ and run Algorithm 4.4 with inputs $\gamma, (2+k)n$. Then*

$$
\left\| \rho^{\widehat{\pi}} - \rho^\star \right\|_\infty \leq (\mathsf{T}_{\mathrm{drop}} + 1) \left( \frac{(7 + e^{-k}64)M + 2}{n-1} \right).
$$

*Proof.* Let $M = \min \left\{ \|h\|_{\mathrm{sp}}, \|h^{\pi^\star}\|_{\mathrm{sp}} + \mathsf{T}_{\mathrm{drop}} + \|h^{\pi^\star}\|_{\mathrm{sp}}\mathsf{T}_{\mathrm{drop}} \right\}$. By identical steps to the previous proof, we have that the vector $V$ output by Algorithm 4.4 satisfies

$$
\| \mathcal{T}_\gamma(V) - V \|_\infty \leq 4(1-\gamma)\gamma^{(k+1)n-(n-1)} \left\| \mathcal{T}^{(n)}(\mathbf{0}) - V^\star_\gamma \right\|_\infty
$$

$$
\leq 4(1-\gamma)\gamma^{(k+1)n-(n-1)}2M
$$

$$
\leq \frac{8}{n}\gamma^{kn}M.
$$

Furthermore we have $\gamma^{kn} = \left(1 - \frac{1}{n}\right)^{kn} \leq e^{-k}$. Combining with Lemma J.6, we have that the policy $\widehat{\pi}$ which is greedy with respect to $r + \gamma PV$ satisfies

$$
\left\| \rho^{\widehat{\pi}} - \rho^\star \right\|_\infty \leq \left( \mathsf{T}^{\widehat{\pi}}_{\mathrm{drop}} + 1 \right) \left( 7(1-\gamma)M + \frac{2+6\gamma}{\gamma} \left\| \mathcal{T}_\gamma(V) - V \right\|_\infty + 2\frac{1-\gamma}{\gamma} \right)
$$

$$
\leq (\mathsf{T}_{\mathrm{drop}} + 1) \left( \frac{7}{n}M + \frac{2+6}{1-\frac{1}{n}}e^{-k}\frac{8M}{n} + 2\frac{\frac{1}{n}}{1-\frac{1}{n}} \right)
$$

$$
= (\mathsf{T}_{\mathrm{drop}} + 1) \left( \frac{7}{n}M + e^{-k}\frac{64M}{n-1} + \frac{2}{n-1} \right)
$$

$$
\leq (\mathsf{T}_{\mathrm{drop}} + 1) \left( \frac{(7+e^{-k}64)M+2}{n-1} \right).
$$

$\square$

We remark that the function $(2+k)(7+64e^{-k})$ is minimized at $k \approx 3.78$ with value $\approx 48.9$, which is a factor of approximately $142/48.9 \approx 2.9$ smaller than the value at $k = 0$.

# K  Auxiliary lemmas

The following calculation, which we use several times and hence include for completeness, is essentially identical to one within the proof of Zurek and Chen [2024b, Lemma 20].

**Lemma K.1.** *For any policy $\pi$ and any $h \in \mathbb{R}^{\mathcal{S}}$, we have that*

$$\left\| (I - \gamma P_\pi)^{-1}(I - P_\pi)h \right\|_\infty \leq \|h\|_{\mathrm{sp}}.$$

*Proof.* Using the Neumann series to expand $(I - \gamma P_\pi)^{-1}$, we have

$$
\begin{aligned}
(I - \gamma P_\pi)^{-1}(I - P_\pi) &= \sum_{t=0}^{\infty} \gamma^t P_\pi^t - \sum_{t=0}^{\infty} \gamma^t P_\pi^{t+1} \\
&= I + \gamma P_\pi \sum_{t=0}^{\infty} \gamma^t P_\pi^t - P_\pi \sum_{t=0}^{\infty} \gamma^t P_\pi^t \\
&= I + (\gamma - 1) P_\pi \sum_{t=0}^{\infty} \gamma^t P_\pi^t \\
&= I - P_\pi (1 - \gamma)(I - \gamma P_\pi)^{-1}.
\end{aligned}
$$

$I, P_\pi$, and $(1 - \gamma)(I - \gamma P_\pi)^{-1}$ are all stochastic matrices (all entries are non-negative and all rows sum to 1), and hence $P_\pi(1 - \gamma)(I - \gamma P_\pi)^{-1}$ is also a stochastic matrix. Then it is immediate for any stochastic matrix $Q$ that elementwise we have $\min_{s \in \mathcal{S}} h(s)\mathbf{1} \leq Qh \leq \max_{s \in \mathcal{S}} h(s)\mathbf{1}$, which implies

$$\left\| (I - \gamma P_\pi)^{-1}(I - P_\pi)h \right\|_\infty = \left\| Ih - P_\pi(1 - \gamma)(I - \gamma P_\pi)^{-1}h \right\|_\infty \leq \|h\|_{\mathrm{sp}}$$

as desired. $\qquad\square$

# L Experiments

While this paper focuses on theoretical complexity analysis, we provide a few preliminary experimental examples.

First we describe, for parameters $k, T \geq 1$ and $\varepsilon > 0$, the parameterized MDP $\mathcal{M}(k, T)$ on which we run our experiments. $\mathcal{M}(k, T)$ has $k + 1$ states, where state 0 is absorbing (has only one action, which leads back to state 0). The remaining $k$ states each have two actions, titled "good" and "bad". The "good" actions all lead deterministically to another of the states $\{1, \dots, k\}$ such that if the "good" action is taken in all such states then it forms a cycle of length $k$. For convenience we let $\pi_c$ denote this policy which takes the "good" action in all states in $\{1, \dots, k\}$. The reward for the "good" action is chosen randomly to be 0 or 0.5 with equal probability. The "bad" action in state $s$ for any $s \in \{1, \dots, k\}$ has reward 1, and leads to state 0 with probability $1/T$ and back to the given state $s$ with probability $1 - 1/T$. Finally, we define the reward of the only action in the absorbing state 0 to be $\rho^{\pi_c}(1) - \varepsilon$ (that is, we compute the reward of the cycle, and then subtract $\varepsilon$). Hence the optimal policy is $\pi_c$ (considered to take the only available action in state 0), which will have $\rho^{\pi_c}(s) = \rho^{\pi_c}(1)$ for all $s \in \{1, \dots, k\}$ and $\rho^{\pi_c}(0) = \rho^{\pi_c}(1) - \varepsilon$. Note all other policies have gains equal to $(\rho^{\pi_c}(1) - \varepsilon)\mathbf{1}$. It is straightforward to compute that $\Delta = \frac{\varepsilon}{T}$ and $\mathsf{T}_{\mathrm{drop}} = T$.

Below we plot the fixed point error $\|\mathcal{T}(h_t) - h_t - \rho^\star\|_\infty$ for the $t$th iterate as generated by Algorithm 1, standard value iteration (VI), and [Lee and Ryu, 2024, Theorem 2] (LR). We intentionally focus on the case where the number of iterations $n$ is smaller than $k$ (the total number of states). We plot both $\varepsilon = 1/2$ and $\varepsilon = 1/20$, keeping $T = 10$, $k = 300$. All experiments were run on a single consumer laptop in less than a minute.

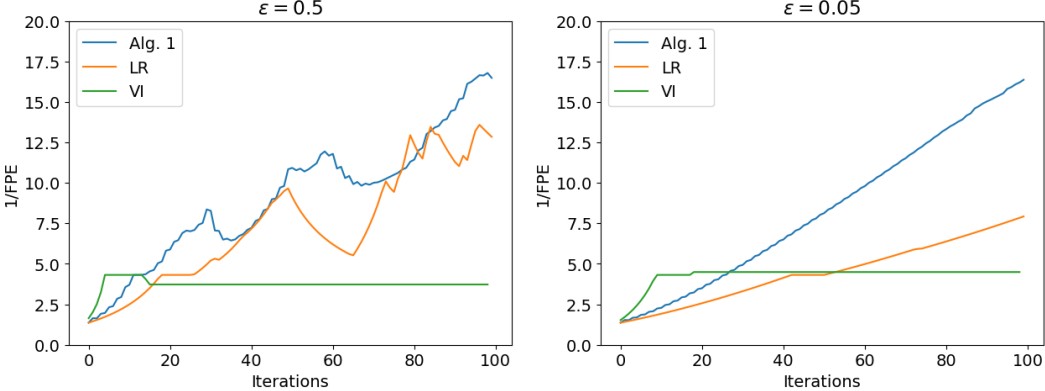

We note that due to periodicity, VI does not converge. In this experiment, decreasing $\varepsilon$ seems not to affect the convergence of Algorithm 1, whereas LR seems to converge more slowly. We emphasize that these experiments only consider the fixed-point error, whereas our sensitivity analysis reveals that it is essential to additionally control $\|P_\pi \rho^\star - \rho^\star\|_\infty$ in order to obtain suboptimality bounds. Overall, much more thorough experimental study on more domains and metrics is needed to better understand the practical performance of our algorithms.

