# OpenReview forum: "Faster Fixed-Point Methods for Multichain MDPs"
_NeurIPS.cc/2025/Conference — NeurIPS 2025 poster_

### Official Review · Reviewer_9RCM · 2025-06-11

**Clarity:** 3
**Significance:** 3
**Originality:** 3
**Rating:** 4
**Confidence:** 4

**Summary:**

This paper presents new value-iteration (VI) algorithms for solving multichain Markov Decision Processes (MDPs) under the average-reward criterion---a setting known for its analytical and computational difficulties. Unlike discounted MDPs, average-reward problems lack contractivity and may yield non-unique Bellman solutions. A central challenge in the multichain setting is the navigation subproblem, where an optimal policy must guide the agent toward the most rewarding recurrent class. The authors address this with novel algorithms that exhibit faster convergence by incorporating sharper complexity measures such as the gain-dropping time \\(T_{\text{drop}}\\). Additional contributions include improved reductions between average-reward and discounted problems, optimal fixed-point methods applicable to Banach spaces, and refined suboptimality bounds. Collectively, these results strengthen the theoretical underpinnings of VI for general MDPs.

**Questions:**

1. How sensitive are the algorithms to inaccuracies in the estimated gain vector \\( \rho^\star \\)? Could a poor estimate significantly affect convergence or policy quality?
2. Many guarantees require that the initial vector \\( h_0 \\) and the target \\( h \\) satisfy both the modified and unmodified Bellman equations. How practical is it to construct such an \\( h \\) in real-world settings?
3. To what extent do the results depend on the specific commutativity conditions of the Bellman operator? Could similar results be generalized to operators lacking this structure?
4. The warm-start procedures (e.g., using \\( T^{(t)}(0) \\)) are critical to performance. Are there other initialization schemes that could yield better empirical performance or simplify implementation?
5. The value of \\( \gamma = 1 - \frac{1}{n} \\) plays a crucial role in performance. Could the algorithm adapt \\( \gamma \\) dynamically during training based on observed convergence behavior?

**Ethical Concerns:**

["NO or VERY MINOR ethics concerns only"]

**Final Justification:**

Having reviewed the authors' responses, I remain satisfied and will uphold my positive recommendation.

**Limitations:**

Yes

**Paper Formatting Concerns:**

No issues

**Quality:**

3

**Strengths And Weaknesses:**

Strengths

1. Introduction of the gain-dropping time \\(T_{\text{drop}}\\) allows more accurate characterization of the navigation difficulty in multichain MDPs.
 2. Proposed methods achieve optimal or near-optimal non-asymptotic convergence rates, surpassing prior works by Lee \& Ryu and Zurek \& Chen.
 3. Results extend to general Banach spaces and unify various aspects of fixed-point theory and MDP planning.
  4. Intermediate results such as discounted-to-average-reward reductions and fixed-point error analyses can benefit a broad range of applications.
   5. The paper presents well-structured algorithms (e.g., Shifted Halpern Iteration, Warm-start Halpern-then-Picard) with precise theoretical guarantees.


Weaknesses

1. The work is entirely theoretical; practical performance on real-world or benchmark problems is not demonstrated.
 2. While asymptotically efficient, the proposed multi-phase algorithms may incur substantial runtime in large-scale problems.
 3. Several results assume access to solutions of both modified and unmodified Bellman equations, which may not be readily obtainable.
 4. The technical presentation is dense, potentially limiting readability for practitioners unfamiliar with fixed-point theory or advanced MDP concepts.

---

> ### Author Rebuttal · Authors · 2025-07-30
>
> We thank you for your positive review. We would like to respond to some of your listed weaknesses and questions.
>
> 1. Weakness 3/Question 2: For the theorems in Section 3 which involve an initialization $h_0$, no assumptions are made on $h_0$. In particular it does not need to satisfy both the modified and unmodified Bellman equations. A vector $h$ satisfying both equations is needed only within the analysis and the statement of the convergence guarantee within the theorems, but is not needed to run the algorithm.
> 2. Question 3: You are correct that our results can be generalized to operators beyond the Bellman operator, as we show in Appendix H.1 (culminating in Theorem H.5). However, this theorem shows that we can find points with small fixed-point error for operators satisfying a certain generalized commutativity property (equation (14)), but there is no obvious general analogue of extracting a “policy” from such a fixed point and arguing about the gain-preserving properties (in the sense of Corollary 2.2) of such a policy, so overall not all of our results have any analogue in this more general setting.
> 3. Question 4: The warm-start procedure is designed so that $T^{(t)}(0)$ is well-aligned with the direction $\rho^\star$ (in the sense that $T^{(t)}(0) \approx t \rho^\star$), so a different procedure could generally try to accomplish this in another fashion, but it is not immediately clear how to do so without knowing $\rho^\star$ a priori.
> 4. Question 5: It is unclear how to modify the proposed algorithms to utilize adaptive stepsizes, but this is an interesting topic of future research.

---

> > ### Comment · Reviewer_9RCM · 2025-08-05
> >
> > Having reviewed the authors' responses, I remain satisfied and will uphold my positive recommendation.

---

### Official Review · Reviewer_pddB · 2025-06-23

**Clarity:** 3
**Significance:** 2
**Originality:** 3
**Rating:** 4
**Confidence:** 4

**Summary:**

This paper presents new iterative algorithms to compute the gain in general finite MDPs and the optimal value in discounted MDPs.
The algorithms combine Picard iterates and Halpern iterates to alleviate the non-contraction of the Bellman operator.
In both cases, the error bounds are within small factors to lower bounds.

**Questions:**

Maybe a numerical section proving experiments on "generic" MDPs would show that the gap with previous approaches is not marginal?


The remark made in conclusion about the computability of Bellman operators is quite intriguing and deserves some explanations.
The reader does not understand what the authors have in mind.
For example, the case of compact action spaces is quite different because solutions of Bellman equations are not always optimal in that case.
This also rises the question of time and space complexity of the proposed algorithms.

**Ethical Concerns:**

["NO or VERY MINOR ethics concerns only"]

**Final Justification:**

I keep my positive score. I like the way the author use Cezaro type iterations but I still have concerns about using this result for ML purposes.

**Limitations:**

The main limitation is the incremental character of the paper. The discussion on previous work is precise and explains  the difference between both bounds. However, this difference can be marginal and can even disappear in some cases.


Another point of interest, that  is not discussed at all, is the use of alternative methods that do not use VI to compute the gain and value of the MDP.


An average or even smooth complexity analysis of fixed point methods in MDPs is also missing. That would have a more significant impact in practical uses.

The topic of this paper is borderline with respect to the main subject of NeurIPS (all aspects of Machine Learning).

**Paper Formatting Concerns:**

no concern about paper formatting.

**Quality:**

4

**Strengths And Weaknesses:**

The paper uses a well thought combination of classical (Picard) and Cezar-type (Halpern) iterates that seem to be innovative.
Maybe a discussion on this coud be added.


Both algorithms  (discounted  and undiscounted) have  marginally better errors bounds than existing results as they introduce a new complexity parameter T_{drop}
that is smaller than existing ones, although it is of the same order in the worst cases.

The definition of T_{drop} is qualitative and not quantitative: why not take the average gain of the actual difference (max with 0) instead of the indicator that looks less sharp?

---

> ### Author Rebuttal · Authors · 2025-07-30
>
> We thank you for your positive review. We would like to respond to some of your listed weaknesses and questions.
>
> 1. Third paragraph in Strengths and Weaknesses section regarding definition of $T_{drop}$: We agree that whether the indicator can be removed is an interesting question, and it would appear that the answer is negative for the following reason: We believe it is possible to show that $\mathbb{E}^\pi [\sum_{t=0}^\infty \rho^\star(S_t) - P_{S_t, A_t}\rho^\star ] = \mathbb{E}^\pi [\rho^\star(s_0) - \rho^\star(S_\infty)]$, where $S_\infty$ is some state in the (potentially random) closed recurrent class reached by the policy. This implies that we would always have $\mathbb{E}^\pi [\sum_{t=0}^\infty \rho^\star(S_t) - P_{S_t, A_t}\rho^\star] \leq \max_s \rho^\star(s) - \min_s \rho^\star(s)$, which is $\leq 1$ for our scaling (where $\|r\|_\infty \leq 1$), which is probably too small to serve as a complexity measure since it is bounded $O(1)$ independently of the properties of the MDP. (We believe that adding a max with $0$ would still not fix the overall issue.)
> 2. Second question: We are happy to expand upon our comments. One potential setup is that, instead of assuming access to the transition matrix $P$, and hence the ability to exactly compute matrix-vector products of the form $Ph$, we could instead investigate the development of algorithms which are only able to access $P$ stochastically via sampling: given $s,a$, we can only draw a sample $S’$ from the distribution given by the row vector $P_{sa}$; in this case then $h(S’)$ will be an unbiased estimate of $(Ph)(sa)$ (the $s,a$th coordinate of the vector $Ph$). Since computing $ (Ph)(sa) $ will generally take $O(S)$ time, if $S$ is large then this sampling approach might be able to form a sufficiently good estimate of $ (Ph)(sa) $ in a smaller amount of time (if samples can be obtained sufficiently quickly).
> 3. First paragraph of limitations: Beyond making quantitative improvements in algorithmic performance relative to prior work, we believe that overall we also develop a range of useful theoretical observations which expand the foundations of this topic. We also note that since value-iteration-style methods form the backbone of many approaches to reinforcement learning and have extensive prior work, even apparently small differences in approaches are still of significant interest and may lead to downstream impacts.
> 4. Second paragraph of limitations (re non-VI methods): We agree that this is an interesting topic, however, we are not aware of major studies of the complexity of alternative methods for solving general average-reward MDPs. Despite the very long history of VI methods for average-reward MDPs, non-asymptotic complexity guarantees have only been developed very recently.
> 5. Fourth paragraph of limitations: We feel that this paper is squarely within the stated call for papers for NeurIPS, fitting within the topic of "Reinforcement learning (e.g., decision and control, planning, hierarchical RL, robotics)", as our paper mainly focuses on solving MDPs (thus addressing the "planning" problem). Our paper is also related to the NeurIPS topic of Optimization. In particular, we present some results in a general language of fixed-point methods common to optimization researchers (ex. Theorem 4.2, Appendices H and I).

---

### Official Review · Reviewer_9Hca · 2025-07-02

**Clarity:** 2
**Significance:** 3
**Originality:** 3
**Rating:** 4
**Confidence:** 3

**Summary:**

They explicitly decompose the multichain average reward MDP problem into the “navigation/transient” subproblem (reaching the best recurrent class) and the “recurrent” subproblem (optimizing long-run reward within a class). Accelereated VI schemes are proposed and lower bounds are matched to show optimality of the theoretical convergence rate.

**Questions:**

- It would greatly enhance the paper's impact by constructing a motivating example in the introduction why multichain average reward is really a pressing subject to study instead of discounting and weakly communicating MDPs in an RL setting.
- Numerical examples for the proposed algorithm would also greatly help.

**Ethical Concerns:**

["NO or VERY MINOR ethics concerns only"]

**Final Justification:**

The authors clarified the points raised so I increased my score from 3 to 4.

**Limitations:**

yes

**Paper Formatting Concerns:**

None of any major formatting issues are noticed.

**Quality:**

3

**Strengths And Weaknesses:**

Strengths

- Halpern‑then‑Picard achieves the lower bounds in Banach spaces; immediately drives faster VI for multichain MDPs.
- Lemma 4.1 ties average‑reward error to discounted residual without exploding terms, enabling a practical $\gamma$‑tuning recipe.

Weaknesses

- Convergence relies on a Blackwell‑optimal policy that simultaneously maximizes both Bellman equations and on knowing n≫1/Δ; these caveats are scattered in footnotes/appendix.

- Notation is heavy - for example, nine new symbols appear in Section 2; key ideas (why Halpern warm‑start solves navigation) are buried on the other hand.

- The link to practical RL speed‑ups is weak. Bounds carry huge factors and require preset phase lengths based on unknown parameters, and no adaptive rule is proposed.

- Claims of “faster” remain theoretical; paper omits timing comparisons on even toy multichain MDPs.

---

> ### Author Rebuttal · Authors · 2025-07-30
>
> We would like to address several inaccuracies of this review.
>
> 1. Response to weaknesses bullet 1: Our algorithms for solving general average-reward MDPs do not depend on knowledge of Blackwell-optimal policies in any way. While some _guarantees_ in our theorems depend on a vector $h$ which satisfies both the modified and unmodified Bellman equations, such a vector $h$ is only used in the _analysis_ and is not needed to run the algorithms. Additionally, knowledge of $n > 1/\Delta$ is not required to run the algorithms. This condition is used in one location, Theorem 3.5, which strengthens the convergence rate for Algorithm 1 relative to Theorem 3.4 when this condition happens to hold. Overall, all our theorems explicitly state all assumptions required for them to apply to the algorithms, and there are no “scattered” caveats in the appendices/footnotes.
> 2. Response to weaknesses bullet 2: The paragraph beginning on line 248 is dedicated to discussing why the warm-start phase is helpful for solving the navigation problem. Section 2 features only two new definitions, our novel parameters $T^\pi_{drop}$ and $T_{drop}$.
> 3. Response to weaknesses bullet 3: As mentioned in our first bullet point, the only condition on phase lengths is $n > 1/\Delta$ appearing in Theorem 3.5, and this only strengthens the convergence rate of Algorithm 1 when this condition holds (relative to Theorem 3.4, which applies without this condition). Algorithm 1 does not require any unknown parameters to be run. While our results are stated for the setting where the desired number of iterations $n$ is decided upon beforehand, our convergence rates can trivially be extended to the “anytime” setting where $n$ is chosen later by running the algorithms with $n=2^1, 2^2, 2^3, …$. We feel as if the leading ($\Theta(1/n)$) terms in our results do not feature prohibitively large constants. Overall we focused on qualitative improvements relative to prior work.

---

> > ### Comment · Reviewer_9Hca · 2025-08-04
> >
> > Thank you for your responses, which clarify some of my previous confusions. I am willing to increase my score, but I could not locate your responses to my questions.

---

> ### Author Response · Authors · 2025-08-06
>
> We would like to address your questions:
>
> Q1: As multichain average-reward MDPs are the most general class of average-reward MDPs, developing algorithms for solving them is an impactful endeavor. Since many RL algorithms are inspired by value iteration methods, we believe that developing value iteration methods with superior nonasymptotic convergence guarantees has the potential to lead to future RL advances. Essentially any real problem where a catastrophic choice of policy can lead to irrecoverable states (for example, a self-driving car which crashes, or a faulty datacenter cooling policy which causes irreversible damage to the datacenter) can be understood as a general MDP. Another example not involving “catastrophic” actions is the investment/consumption example in [Puterman 1994, Example 9.0.1], where, at a high level, long-term wealth can only be accumulated by individuals with sufficient wages and/or initial wealth, creating multichain structure. General/multichain MDPs have been of interest to researchers for many decades and we refer to the references within [Puterman 1994, Chapter 9] for many more examples.
>
> Since it is probably impossible to develop sublinear regret algorithms for general average-reward MDPs in the standard online setting, this raises the importance of developing algorithms for the planning and offline settings, as they are thus the only ways of coping with these problem settings. We also note that our study of general MDPs has led to technical advances for both discounted (see Section 4) and weakly communicating MDPs (see Corollary 3.2, which applies to evaluating possibly suboptimal policies in weakly communicating MDPs, since even in weakly communicating MDPs not all policies are guaranteed to have constant gains).
>
>
>
> Q2: Our work is focused on theory and complexity bounds, and the most relevant prior work on nonasymptotic guarantees for multichain MDPs (i.e. [Lee and Ryu, 2024], also [Zurek and Chen, 2025b]) does not provide numerical experiments. Still, we are happy to provide an example which demonstrates some qualitative improvements demonstrated by our work.
>
> The following experiment is conducted on a MDP which consists of $T+1$ states, for a parameter $T$. State $0$ has only one action which gives reward $0$ (hence this state is absorbing and has $\rho^\star(0) = 0$.) The remaining $T$ states each have two actions, a “good” action and “bad” action. The good action leads to another of these $T$ states with probability $1$, such that if the good action is taken in all states, then they form a cycle of length $T$. The reward for the good action in each state is chosen randomly to be $0$ or $0.5$ with equal probability. In all these states $i = 1, .., T$ the bad action gives reward $1$ and leads to state $0$ with probability $1/T$, and remains in the state $i$ with probability $1-1/T$. Hence the optimal policy is to take the good action in all states besides state $0$ (leading to a gain of around $0.25$ for such states). For such MDPs we just consider the fixed-point error (FPE), since the problem of policy optimization is relatively straightforward for these instances, which are far from worst-case. Hence we plot our Algorithm 1 and the algorithm of [Lee and Ryu, 2024] (as the other algorithms mentioned in the paper are not designed to minimize fixed-point error of the average-reward Bellman operator). We also plot standard value iteration (VI). None of these algorithms require any tuning parameters. The following tables compare these two algorithms, where we set the environment parameter $T = 200$ and the total number of iterations ranges to $2000$:
>
> | n | 0 | 200 | 400 | 600 | 800 | 1000 | 1200 | 1400 | 1600 | 1800 | 2000 |
> | :-- | :-: | :-: | :-: | :-: | :-: | :-: | :-: | :-: | :-: | :-: | :-: |
> | Algorithm 1 FPE | 0.75 |  0.3828 |  0.1873 | 0.0788 | 0.0065 | 0.0053 | 0.0043 | 0.0038 | 0.0032 | 0.0029 | 0.0026 |
> |[Lee and Ryu 2024] FPE | 0.75  | 0.4855 | 0.3174 | 0.2164 | 0.1389 | 0.0816 | 0.0382 | 0.0109 | 0.0116 | 0.0092 | 0.0076 |
> | VI FPE | 0.75 |  0.1639 | 0.0519 | 0.0519 | 0.0519 | 0.0519 | 0.0519 | 0.0519 | 0.0519 | 0.0519 | 0.0519 |
>
>
> As we can see, our Algorithm 1 converges faster, in particular exhibiting rapid convergence around 800 iterations, which may be related to the faster rate predicted by Theorem 3.5 when the number of iterations is sufficiently large. We also note that because of the periodic structure of this MDP (due to the cycle), standard (average-reward) value iteration does not drive the fixed-point error to $0$. We will add this and additional experimental results in the final version.

---

> > ### Comment · Reviewer_9Hca · 2025-08-08
> >
> > Thanks for your response. I have no more questions.

---

### Official Review · Reviewer_RpJU · 2025-07-03

**Clarity:** 3
**Significance:** 4
**Originality:** 4
**Rating:** 5
**Confidence:** 3

**Summary:**

This paper presents a finite-time convergence analysis for value iteration in general (multichain) MDPs under the average-reward setting. Unlike traditional value iteration, which relies on the contraction property of the Bellman operator (typically present in discounted settings), the average-reward case lacks contraction, rendering standard analysis techniques inapplicable. While prior works have attempted to address this challenge using complexity parameters, their guarantees often degrade to worse than $\mathcal{O}(1/n)$ in the general MDP setting. This paper introduces a novel complexity measure, gain-dropping time, which is both theoretically analyzed and shown to yield sharper convergence bounds. In addition, the authors develop new average-to-discounted reductions and establish optimality results for fixed-point methods in general Banach spaces.

**Questions:**

* A proof sketch for key results (e.g., Theorems 3.4 and 4.5) would improve readability and help clarify the paper's novel analytical techniques.
* Lemma F.5 & F.6 shows that $T_{drop}$, which is asymptotically better than either bound in general. However, could $T_{drop}$ still be on the order of $in$ the worst case? Clarifying the tightness of this bound would be helpful.
* The concept of gain-dropping time is quite interesting. Is it possible to extend this notion to analyze sample complexity or regret bounds in the offline or online reinforcement learning settings?

**Ethical Concerns:**

["NO or VERY MINOR ethics concerns only"]

**Final Justification:**

I accept the authors’ explanation and will maintain my positive score for this paper.

**Limitations:**

The authors acknowledge that exact computation of Bellman operators may be infeasible in large-scale MDPs, and stochastic approximation may be necessary. Another limitation is that although the authors show that $T_{drop}$, no absolute constant upper bound is provided. As such, practical estimation or bounding of $T_{drop}$​ remains an open issue.

**Quality:**

3

**Strengths And Weaknesses:**

**Strengths**
* This paper introduces a novel complexity measure, gain-dropping time, to analyze the convergence rate of value iteration in multichain average-reward MDPs. Based on this measure, the authors derive convergence bounds that improve upon prior work.

* The motivation is clearly stated, and the limitations of existing approaches are thoroughly discussed.

* A new reduction technique is proposed that connects average-reward problems to discounted ones. This enables better bounds on the optimality gap under general MDP settings.


**Weaknesses**
* The paper is entirely theoretical. While the results are strong, empirical experiments demonstrating faster convergence compared to existing baselines would significantly strengthen the paper.

* The convergence guarantees rely on the assumption that the bias vector $h$ satisfies both the modified and unmodified Bellman equations. Although Theorem D.4 discusses special cases (e.g., when $h$ is constant), a more general constant bound on $h$ would enhance the practical interpretability of the results.

---

> ### Author Rebuttal · Authors · 2025-07-30
>
> We thank you for your positive review. We would like to respond to some of your listed weaknesses and questions.
>
> 1. Weakness 2: While we agree that requiring $h$ to satisfy both Bellman equations reduces its interpretability (relative to $h$ which satisfies only the unmodified Bellman equation, since in this case the bias function of a bias-optimal/Blackwell-optimal policy can simply be used), we note that prior works studying the convergence of value iteration for general average-reward MDPs [Lee and Ryu 2024, Puterman 1994] have also utilized $h$ satisfying these requirements. (In fact, [Lee and Ryu 2024, Puterman 1994] analyze under arguably an even less interpretable assumption, that $h$ satisfies the modified Bellman equation and also admits a policy in the simultaneous argmax of equations (2a) and (2b), in the sense of equations (7) and (8); one of our minor contributions is showing that such $h$ are exactly the $h$ which satisfy both the modified and unmodified Bellman equations (Lemma D.1).)
> 2. Question 1: Here we provide more details of the proofs of Theorems 3.4 and 4.5. Overall, we agree that including these details would benefit the paper, and we would use an additional camera-ready page to do so.
>
>     - Theorem 3.4: The high-level approach is to control both terms appearing in Corollary 2.2, which itself is a direct consequence of Lemma 2.1. As discussed in the paragraph beginning on line 241, Lemma 3.3 can be used to obtain an estimate $\hat{\rho}$ of the optimal gain $\rho^\star$. This enables the formation of the approximately shifted operator $\widehat{\mathcal{T}} := \mathcal{T} - \hat{\rho}$ which is used in Algorithm 1. At a high level, since standard convergence analyses of Halpern iteration would guarantee small fixed-point error if it were ran using the “perfectly” shifted operator $\mathcal{T} - \rho^\star$, our analysis then shows that instead using $\widehat{\mathcal{T}}$ does not cause too much error. As discussed in the paragraph beginning on line 248, this is only immediately sufficient to control the fixed-point error term from Corollary 2.2, but since the second phase of the algorithm is initialized with $x_n$ which is well-aligned with the direction $\rho^\star$ (and because our analysis demonstrates that this property is still true after the second phase terminates), the other term $\| P_\pi \rho^\star - \rho^\star \| $ appearing in Corollary 2.2 is also guaranteed to be small after termination.
>
>     - Theorem 4.5: The starting point here is to use our novel discounted reduction Lemma 4.1. Internally the proof of this lemma also makes use of Corollary 2.2 and connects its bound to quantities related to the discounted Bellman operator. As discussed in the paragraph beginning on line 287, Lemma 4.1 suggests that we must try to achieve small discounted fixed-point error while using a large value of the effective horizon $\frac{1}{1-\gamma}$. This objective leads us to the development of Algorithms 2 and 3, for reasons which are explained in those sections but in particular in the paragraphs beginning on lines 314 and 330. This culminates in Algorithm 3 and Theorem 4.4 describing its performance, which we can essentially directly combine with the reduction Lemma 4.1. One interesting key lemma is Lemma J.1, which relates the optimal discounted value function $V_\gamma^\star$ to $\rho^\star$ using the $T_{drop}$ parameter and is key for both the proof of Lemma 4.1 and the proof of Algorithm 3/Theorem 4.4. Lemma J.8 performs a similar function but instead relates $\mathcal{T}^{(n)}(\mathbf{0})$ to $\rho^\star$, and is also key to the idea of Algorithm 3 and Theorem 4.4.
> 3. Question 2: The bounds of Lemmas F.5 and F.6 are tight, as can be seen by the following example: Fix a parameter $T > 1$. Consider an MDP with three states (1,2,3). State 1 has only one action which is absorbing with reward +1. State 2 also has only one action which is absorbing and with reward +0. State 3 has two actions: the action 1 leads deterministically to state 1 and gives reward +1. Action 2 gives reward +1, and has probability $1-1/T$ of leading back to state 3, and probability $1/T$ of leading to state 2. Then it is straightforward to compute that $\rho^\star = [1 ~ 0 ~ 1]$ and the parameters $\mathsf{B}, \mathsf{T}_{drop}$, and $1/\Delta$ are all equal to $T$ for this instance.
> 4. Question 3: We believe that there is a high possibility that the gain-dropping time could be useful in simulator/offline RL settings, and it might be interesting to understand whether new algorithms could achieve sample complexity results in terms of this sharper parameter, rather than the bounded transient time parameter used in [Zurek and Chen 2025b]. It may also be useful in the online setting, but a main obstacle in the online setting is formulating a useful notion of regret for general average-reward MDPs, since generally an algorithm may become “trapped” in low gain regions of the MDP without any ability to escape.

---

> > ### Comment · Reviewer_RpJU · 2025-08-06
> >
> > I appreciate the authors' response. Most of my concerns have been addressed, and I hope the authors will consider incorporating this discussion into the paper.
> >
> > I have one additional question regarding my Q2. I apologize for the typo in my original review, what I intended to ask is: could $T_{\text{drop}}$ still be on the order of $n$ in the worst case? More specifically, since the upper bounds of both $B$ and $1/\Delta$ could be in terms of $n$, even though the authors show that $T_{\text{drop}} \leq B$ and $T_{\text{drop}} \leq 1/\Delta$, an analysis clarifying the relationship between $T_{\text{drop}}$ and $n$ would be helpful.

---

> > > ### Author Response · Authors · 2025-08-06
> > >
> > > We agree that adding the above discussion will benefit the paper and thank the reviewer for their comments and suggestions.
> > >
> > > Thank you for clarifying your question. While the parameters $T_{drop}, 1/\Delta, B$ depend only on properties of the MDP instance (i.e. $P$ and $r$), the value $n$ is the number of iterations that a practitioner chooses to run a given algorithm. Therefore actually none of the parameters $T_{drop}, 1/\Delta, B$ are necessarily bounded by $n$ and overall they are not related. Of course, $n$ needs to be sufficiently large relative to these parameters for the theorems to imply convergence, and so as you mention it is desirable to have $T_{drop}$ rather than the other two potentially much larger complexity parameters.
> > >
> > > We would also like to mention that beyond simply being smaller than $1/\Delta$ and $ B$ and thus giving faster convergence guarantees, we believe that our introduction of $T_{drop}$ represents a conceptual improvement in our understanding of multichain MDPs: As discussed in Section 2, the definition of $T_{drop}$ is closely related to the exact sensitivity analysis provided by Lemma 2.1 and clarifies the sources of hardness for general MDPs, whereas prior to our work the complexity parameters $1/\Delta$ and $ B$ were potentially very large even for many MDPs known to be simple (as discussed in the last paragraph of Section 2) and also incomparable with each other.

---

### Comment · Area_Chair_RWYe · 2025-08-04

The authors have provided responses to your reviews. Please first read and acknowledge it. If you need additional clarification or has additional questions for the authors, please also post them ASAP. Thanks.

---

### Decision · Program_Chairs · 2025-09-17

**Decision:**

Accept (poster)

**Comment:**

This submission studies faster fixed-point methods for solving multichain Markov Decision Processes (MDPs) under the average-reward criterion, where the Bellman operator is non-contractive and standard value-iteration analyses struggle. The authors propose iterative schemes that combine Halpern (Cesàro-type) and Picard updates, introduce a sharper complexity measure and develop average-to-discounted reductions and optimal fixed-point results in general Banach spaces. They provide non-asymptotic convergence guarantees that improve on prior bounds (e.g., Lee & Ryu; Zurek & Chen), with lower-bound matching up to small factors in the relevant settings.

Reviewers generally found the theoretical contributions solid and well-motivated. The new complexity parameter offers a cleaner characterization of the navigation/transient difficulty in multichain MDPs and is smaller than previous parameters (though worst-case orders can coincide). The algorithms (e.g., warm-start Halpern-then-Picard, shifted Halpern) are clearly specified with precise guarantees, and the discounted-to-average reduction avoids blow-ups that hampered earlier analyses.

At the same time, concerns remain. The work is entirely (or nearly entirely) theoretical; empirical validation is limited (the authors later provided a simple synthetic example illustrating fixed-point error decay, but broader experiments are still missing). Some assumptions raised questions about applicability, although the authors clarified that vectors satisfying both the modified and unmodified Bellman equations are used for analysis and statements of guarantees—not required to run the algorithms.

During rebuttal, the authors added proof sketches and clarifications (including tightness examples for parameter relationships) and addressed several misunderstandings. Taken together with the reviewers’ largely positive assessments (three borderline-accepts and one accept), I view this as a strong theoretical contribution that advances the state of the art on convergence guarantees for multichain average-reward MDPs. While richer experiments would strengthen the paper, the novelty and rigor here meet the bar.

Recommendation: Accept.